# Recovering Manifold Structure Using Ollivier-Ricci Curvature

**Tristan Luca Saidi**[1] , **Abigail Hickok**[2], **Andrew J. Blumberg**[3]

[1]Department of Computer Science, Columbia University
[2]Department of Statistics & Data Science, Wu Tsai Institute, Yale University
[3]Departments of Mathematics and Computer Science, Irving Institute for Cancer Dynamics,
  Columbia University

## Abstract

We introduce ORC-Manl, a new algorithm to prune spurious edges from nearest neighbor graphs using a criterion based on Ollivier-Ricci curvature and estimated metric distortion. Our motivation comes from manifold learning: we show that when the data generating the nearest-neighbor graph consists of noisy samples from a low-dimensional manifold, edges that shortcut through the ambient space have more negative Ollivier-Ricci curvature than edges that lie along the data manifold. We demonstrate that our method outperforms alternative pruning methods and that it significantly improves performance on many downstream geometric data analysis tasks that use nearest neighbor graphs as input. Specifically, we evaluate on manifold learning, persistent homology, dimension estimation, and others. We also show that ORC-Manl can be used to improve clustering and manifold learning of single-cell RNA sequencing data. Finally, we provide empirical convergence experiments that support our theoretical findings.

## 1 Introduction

The first step for almost all geometric data analysis tasks is to build a nearest neighbor graph. This reflects faith in the manifold hypothesis—the belief that the data actually lies on a low-dimensional submanifold of the ambient $\mathbb{R}^D$. In this setting, the nearest neighbor graph recovers the intrinsic geometry of the data manifold, using the observation that small ambient distances lie along the manifold whereas larger ones may not.

Unfortunately, building nearest neighbor graphs from noisy data typically results in inaccurate representation of the metric structure of the underlying manifold. In this paper, we study edges in nearest neighbor graphs that shortcut through the ambient space and bridge distant neighborhoods of the underlying manifold. Such "shortcut edges" distort inferred distances and negatively impact downstream algorithms that operate on the graphs.

We show that Ollivier-Ricci curvature (ORC), a measure of discrete curvature on graphs (Ollivier, 2007), can be used to effectively identify shortcut edges when the data consists of noisy samples from a low dimensional submanifold. We also show that graph distances can be used to support the identification of shortcut edges, allowing us to avoid accidentally catching "good" edges. Guided by these results, we describe an algorithm, Ollivier-Ricci Curvature-based Manifold Learning and recovery (ORC-Manl), to detect and prune shortcut edges.

ORC-Manl marks edges with extremely negative ORC as candidate shortcut edges. The algorithm then constructs a thresholded graph with the candidates removed. We then use the thresholded graph distance between the endpoints of all candidate edges to check—if the distance is large, the edge was likely shortcutting the manifold through the ambient space. If the distance is small, the edge is added back to the graph. We find that despite its simplicity, ORC-Manl is incredibly effective and provides tangible performance improvement for downstream geometric data analysis algorithms for a variety of synthetic and real datasets. Our code for ORC-Manl and all experiments are available on GitHub.[†]

---

[*]Correspondence: `tls2160@columbia.edu`, `abigail.hickok@yale.edu`, `andrew.blumberg@columbia.edu`

[†]Link to implementation: `https://github.com/TristanSaidi/orcml`

## 1.1 CONTRIBUTIONS

We introduce a general-purpose method, ORC-MANL, that uses discrete graph curvature to detect and prune unwanted connectivity in nearest-neighbor graphs. Our method is theoretically justified, and in practice significantly improves the performance of downstream tasks like manifold learning, persistent homology, and estimation of important geometric quantities like intrinsic dimension and curvature. Furthermore we find that ORC-MANL is effective on real world single-cell RNA sequencing data: ORC-MANL pruning reveals clusters in PBMC data in accordance with ground truth annotations and results in embeddings that better preserves communities of neuronal cells. Finally, we also include experiments to show that our theoretical convergence results are supported by empirical experimentation.

## 1.2 RELATED WORK

**Graph Pruning.** Several graph pruning approaches have been proposed in the literature. Some use density estimation as a heuristic for detecting unwanted edges and show results on noiseless toy datasets (Xia et al., 2008; Chao et al., 2007). Zemel & Carreira-Perpiñán (2004) proposed an approach that builds a minimum spanning tree of the original nearest neighbor graph; but this relies on the assumption that shortcutting edges are longer than good edges, a phenomenon that does not always hold. Another family of approaches attempt to adaptively tune the number-of-neighbors parameter $k$ of the nearest neighbor graph based on the local geometry of the data. Zhan et al. (2009); Elhenawy et al. (2019) adopt similar approaches, looking at the linearity of neighborhoods using PCA and pruning accordingly. These methods typically demonstrate a limited set of results on noiseless data and provide little theoretical justification.

**Ollivier-Ricci Curvature.** Ollivier-Ricci curvature (ORC) was proposed as a measure of curvature for finite metric spaces by Ollivier (2007), with follow-up results demonstrating theoretical and empirical convergence to the underlying manifold Ricci curvature under mild assumptions (Ollivier, 2009; van der Hoorn et al., 2021). In the network geometry literature, ORC based approaches have been effective for community detection, drawing connections to Ricci flow from Riemannian geometry (Sia et al., 2019; Ni et al., 2019). Sia et al. (2019) prune edges with extremely negative ORC and justify it using theory that argues that ORC detects *communities*. Our theoretical work instead justifies the use of ORC for recovering correct *manifold* structure. A multitude of papers have used ORC for clustering and modeling diffusion processes on graphs as well (Gosztolai & Arnaudon, 2021; Tian et al., 2023). This work has led to applications for graph neural networks, where ORC was used to prevent over-squashing and over-smoothing (Liu et al., 2023; Nguyen et al., 2023), and improving encodings of local graph structure (Fesser & Weber, 2024).

## 2 BACKGROUND AND DEFINITIONS

### 2.1 DIFFERENTIAL AND RIEMANNIAN GEOMETRY

**Manifolds.** A manifold is a generalization of the notion of a surface—it is a topological space that locally looks like Euclidean space. Concretely, a manifold $\mathcal{M}$ is an $m$-dimensional space such that for every point $x \in \mathcal{M}$ there is a neighborhood $U \subseteq \mathcal{M}$ such that $U$ is homeomorphic to $\mathbb{R}^m$. At every point $x \in \mathcal{M}$ one can attach a $m$-dimensional vector space called the *tangent space* (denoted $T_x\mathcal{M}$) that contains all directions in which a path in $\mathcal{M}$ can tangentially pass through $x$. In a similar manner, the *normal space* at $x$ (denoted $N_x\mathcal{M}$) is a vector space containing all directions normal to $\mathcal{M}$ at $x$. We will work with Riemannian manifolds, which are smooth manifolds endowed with a *Riemannian metric*. A Riemannian metric is an assignment of an inner product to each tangent space that varies smoothly with respect to $x \in \mathcal{M}$. This metric allows one to make statements about local similarity, angles, and distances. For a more detailed treatment of differential and Riemannian geometry, we direct readers to Prasolov (2022) and Lee (2018).

**Geodesics.** In this paper we are concerned with submanifolds $\mathcal{M}$ of $\mathbb{R}^D$ whose Riemannian metrics are induced by the ambient Euclidean metric. Recall that the length of a continuously differentiable path $\gamma : [a, b] \to \mathbb{R}^D$ is $L(\gamma) = \int_b^a \|\gamma'(t)\|_2 \, dt$. The *geodesic* distance between two points $x$ and $y$ in a submanifold $\mathcal{M}$ of $\mathbb{R}^D$ is simply the minimum length over all continuously differentiable paths

connecting $x$ and $y$. In this paper we consider the distance metric $d_{\mathcal{M}}(a, b)$ as the length of the geodesic path through $\mathcal{M}$ connecting $a \in \mathcal{M}$ to $b \in \mathcal{M}$.

**Tubular Neighborhoods.** The $\tau$-tubular neighborhood $\text{Tub}_{\tau}(\mathcal{M})$ of a submanifold $\mathcal{M}$ of $\mathbb{R}^D$ is the set $\{x \in \mathbb{R}^D \mid d(\mathcal{M}, x) \leq \tau\}$, where $d(\mathcal{M}, x)$ is the Euclidean distance between $x$ and the nearest point $x' \in \mathcal{M}$. Intuitively, $\text{Tub}_{\tau}(\mathcal{M})$ is a "fattened" submanifold of $\mathbb{R}^D$ that envelops $\mathcal{M}$. We use the tubular neighborhood as a model of the support of a noisy sampling distribution over $\mathcal{M}$ with a bounded level of isotropic noise.

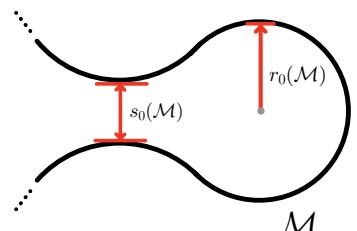

Figure 1: Visualization of the manifold parameters $r_0(\mathcal{M})$ and $s_0(\mathcal{M})$.

**Manifold Embedding Parameters.** We will define two manifold parameters that are central to the theoretical analysis that follows, adopted from Bernstein et al. (2001). The *minimum radius of curvature* is $r_0(\mathcal{M}) = \frac{1}{\max_{\gamma, t} \|\ddot{\gamma}(t)\|_2}$ where $\gamma : \mathbb{R}_+ \rightarrow \mathbb{R}^D$ is a time-parameterized unit-speed geodesic in $\mathcal{M}$. Intuitively, $r_0(\mathcal{M})$ indicates the extent to which manifold geodesics curl in the ambient Euclidean space. The *minimum branch separation* $s_0(\mathcal{M})$ is the largest positive number such that $\forall x, y \in \mathcal{M}, \|x - y\|_2 < s_0(\mathcal{M})$ implies $d_{\mathcal{M}}(x, y) \leq \pi r_0(\mathcal{M})$. While in general people use quantities like the *reach* and *injectivity radius* to describe manifold embeddings, these quantities are intimately related to the ones described; we choose to stick to the described parameters for compatibility with theorems invoked from prior work.

**Proposition 1.** *Suppose $\mathcal{M}$ is a compact submanifold of $\mathbb{R}^D$ without boundary and with a minimum radius of curvature $r_0(\mathcal{M})$. Then $\text{Tub}_{\tau}\mathcal{M}$ has a minimum radius of curvature of $r_0(\mathcal{M}) - \tau$.*

## 2.2 NEAREST NEIGHBOR GRAPHS AND UNWANTED CONNECTIVITY

Geometric machine learning approaches attempt to capture the underlying structure of $\mathcal{M}$ from noisy samples $\mathcal{X}$, typically beginning with the construction of a nearest neighbor graph. These graphs use connectivity rules of two flavors: $\epsilon$-radius, or $k$-nearest neighbor ($k$-NN) (Bernstein et al., 2001). The $\epsilon$-radius connectivity scheme asserts that for any two vertices $a$ and $b$, an edge exists between them if $\|a - b\|_2 \leq \epsilon$. The $k$-NN rule on the other hand asserts that the edge exists only if $b$ is one of the $k$ nearest neighbors of $a$, or vice versa. While the $\epsilon$-radius rule is more amenable to theoretical analysis, the $k$-NN rule is used more often in practice. Unless otherwise specified, an edge $(x, y)$ in a neighbor graph is assigned the weight $\|x - y\|_2$.

> **Definition 2.1.** *An edge $(x, y)$ in a nearest neighbor graph of noisy samples from $\mathcal{M}$ is a shortcut edge if*
> $$d_{\mathcal{M}}(\text{proj}_{\mathcal{M}} x, \text{proj}_{\mathcal{M}} y) > (\pi + 1)r_0(\mathcal{M})$$
> *where $\text{proj}_{\mathcal{M}}(\cdot)$ is the orthogonal projection onto $\mathcal{M}$ and $r_0(\mathcal{M})$ is the minimum radius of curvature of $\mathcal{M}$.*

One of the main corrupting phenomena in the construction of nearest neighbor graphs is the formation of *shortcut edges*, edges that bridge distant neighborhoods of the underlying manifold $\mathcal{M}$. Figure 2 depicts a simple 2-dimensional example of a shortcut edge, while Definition 2.1 provides a mathematical description of such edges. Note that in Section 3 we state the assumption that $\tau < r_0(\mathcal{M})$, which guarantees the uniqueness of the projection map. Intuitively, a shortcut edge is one where a large distance through $\mathcal{M}$ is required to traverse between the endpoints, relative to the Euclidean distance between the endpoints of the edge itself. While the definition has no explicit dependence on $\|x - y\|_2$, we make the assumption in Section 3 that the connectivity threshold $\epsilon$ (and thus $\|x - y\|_2$) is smaller than $r_0(\mathcal{M})$.

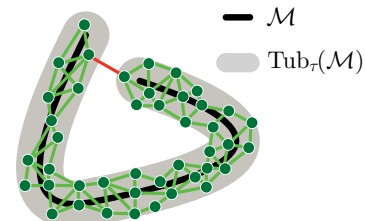

Figure 2: Visualization of a nearest neighbor graph build from noisy samples from $\mathcal{M}$. Desirable edges are shown in green, while the shortcut edge is shown in red.

## 2.3 OLLIVIER-RICCI CURVATURE

Ollivier-Ricci curvature (ORC) was proposed as a measure of curvature for discrete spaces (Ollivier, 2009) by leveraging the connection between optimal transport and Ricci curvature of Riemannian manifolds. While there have been many subtle variations of ORC, we will describe a modification to the most common one. Given an edge $(x, y)$ in a weighted graph $G$ with vertices $V$ and edges $E$, define the neighborhoods of $x$ and $y$ as $\mathcal{N}(x) := \{v \in V \mid (x, v) \in E, v \neq x\}$ and $\mathcal{N}(y) := \{v \in V \mid (y, v) \in E, v \neq y\}$. Further define $\mu_x$ and $\mu_y$ as uniform probability measures over $\mathcal{N}(x) \setminus \{y\}$ and $\mathcal{N}(y) \setminus \{x\}$, respectively. The ORC of the edge $(x, y)$, denoted $\kappa(x, y)$, is defined as

$$\kappa(x, y) := 1 - \frac{W(\mu_x, \mu_y)}{d_G(x, y)}$$

where $W(\mu_x, \mu_y)$ is the 1-Wasserstein distance between the measures $\mu_x$ and $\mu_y$ (regarded as measures on $V$), with respect to the weighted shortest-path metric $d_G(\cdot, \cdot)$. The weighted shortest-path metric $d_G(x, y)$ simply represents the total weight of a weight-minimizing path from $x$ to $y$ through $G$. The 1-Wasserstein distance is computed by solving the following optimal transport problem,

$$W(\mu_x, \mu_y) = \inf_{\gamma \in \Pi(\mu_x, \mu_y)} \sum_{(a,b) \in V \times V} d_G(a, b) \gamma(a, b)$$

where $\Pi(\mu_x, \mu_y)$ is the set of all measures on $V \times V$ with marginals $\mu_x$ and $\mu_y$. Intuitively, ORC quantifies the local structure of $G$: negative curvature implies that the edge is a "bottleneck", while positive curvature indicates the edge is present in a highly connected community.

In our setting, the vertices of the graph $G$ are points in $\mathbb{R}^D$, and edges connect points $(x, y)$ such that $\|x - y\|_2 \leq \epsilon$, where $\epsilon$ is a user-chosen connectivity threshold. A common choice is to weight the edges by the Euclidean distance $\|x - y\|_2$, but we make a slight modification to this formulation. In computing the ORC in the present paper we use unweighted edges; namely, all edges are snapped to a weight of 1. This is a key aspect of our method, as it forces the ORC to reflect only the local connectivity and makes it invariant to scale. This modification also restricts the ORC values to lie between $-2$ and $+1$, whereas with weighted graphs, the values are unbounded below. *For the rest of the paper, any reference to Ollivier-Ricci curvature is a reference to the formulation we have just described.* In all cases other than computing ORC, an edge $(x, y)$ is assigned the weight $\|x - y\|_2$, as mentioned in Section 2.2.

> **Proposition 2.** *For any edge $(x, y)$ in an unweighted graph, $-2 \leq \kappa(x, y) \leq 1$.*

## 3 METHOD AND THEORETICAL RESULTS

In this section we will describe ORC-MANL, a novel algorithm for pruning nearest neighbor graphs based on ORC and metric distortion. We will also describe the theoretical results that justify the algorithm construction, the proofs of which are in Appendix A.4.

ORC-MANL (Algorithm 1) takes as input noisy samples $\mathcal{X}$ from some underlying manifold, an accompanying nearest neighbor graph $G = (V, E)$ of the data, and tolerances $\delta, \lambda \in [0, 1]$. The method constructs a candidate set $C$ of edges that have curvature more negative than a threshold just larger than $-1$. We note that the exact expression for this threshold, $-1 + 4(1 - \delta)$, arises from Lemma A.1. The expression simply captures the notion that shortcut edges tend to have ORC less than some constant value near $-1$, where $\delta = 1$ results in a strict threshold of $-1$ and smaller $\delta$ values introduce slack. Importantly, not every edge in the candidate set is necessarily a shortcut; good edges can have extremely negative curvature as well. To combat this, the method proceeds by constructing a thresholded graph $G' = (V, E')$ where $E' = E \setminus C$. The *weighted* graph distance $d_{G'}(x, y)$ is checked for every edge $(x, y) \in C$. If this distance exceeds the threshold derived in Theorem 3.3, we can be confident that $(x, y)$ undesirably bridges distant neighborhoods of $\mathcal{M}$ (a visualization for which is shown in Figure 13). The edge is therefore removed from $G$. If the threshold is not exceeded, the edge is not removed. We would also like to emphasize that the threshold arising from Theorem 3.3 is unaffected by feature scale, as it is equivalent to a bound on $d_{G'}(x, y)/\epsilon$. Finally, for a time complexity analysis of the ORC-MANL algorithm we direct readers to Appendix A.2.

**Algorithm 1** ORC-MANL

**Require:** $G = (V, E)$ a nearest neighbor graph, $\lambda, \delta$
1: $C \leftarrow \{\}$
2: **for** $(x, y)$ in $E$ **do**                    ▷ candidate selection
3:     $\kappa(x, y) \leftarrow$ OllivierRicci$(G, (x, y))$
4:     **if** $\kappa(x, y) \leq -1 + 4(1 - \delta)$ **then**     ▷ Lemma A.1
5:         $C \leftarrow C \cup \{(x, y)\}$
6: $E' \leftarrow E \setminus C$
7: $G' \leftarrow (V, E')$
8: **for** $(x, y)$ in $C$ **do**                    ▷ shortcut detection
9:     $d_{G'}(x, y) \leftarrow$ ShortestPath$(G', x, y)$
10:    **if** $d_{G'}(x, y) > \frac{\pi(\pi+1)(1-\lambda)}{2\sqrt{24\lambda}}\epsilon$ **then**   ▷ Theorem 3.3
11:        $E \leftarrow E \setminus \{(x, y)\}$
12: **return** $(V, E)$

We provide three theoretical results that justify ORC-MANL. First, we show that when the data generating the nearest neighbor graph consists of noisy samples from an underlying manifold $\mathcal{M}$, the ORC of shortcutting edges tends to be very negative. Second, we show that as one samples more points, every vertex has an increasing number of *non*-shortcutting edges with very *positive* ORC. Last, we derive a bound on the graph distance (in the ORC thresholded graph $G'$) between any vertices connected by a shortcut edge in $G$.

We consider the setting where $\mathcal{M}$ is a compact $m$-dimensional smooth submanifold of $\mathbb{R}^D$ without boundary. Let $\text{Tub}_\tau(\mathcal{M})$ be the tubular neighborhood of $\mathcal{M}$, and assume $\mathcal{X} \subset \text{Tub}_\tau(\mathcal{M})$ consists of $n$ independent draws from the probability density function $\rho : \text{Tub}_\tau(\mathcal{M}) \to \mathbb{R}_+$,

$$\rho(z) = \begin{cases} \frac{1}{Z} e^{\frac{-\|z - \text{proj}_\mathcal{M} z\|_2^2}{2\sigma^2}}, & \|z - \text{proj}_\mathcal{M} z\|_2 \leq \tau \\ 0, & \text{o.w.}, \end{cases} \quad (1)$$

where $Z$ is a normalizing constant such that $\int_{\text{Tub}_\tau(\mathcal{M})} \rho(z)dV$ integrates to 1. We are given a constant $\lambda < 1$, which controls the threshold we use when checking the weighted-graph distances of candidate edges. For the rest of the paper, suppose that

1. (*Support criteria*): $2\tau < \epsilon$, $3\tau < s_0(\mathcal{M})$, $3\tau < r_0(\mathcal{M})$
2. (*$\epsilon$-radius criterion*): $\epsilon < \min\left\{\sqrt{(s_0(\mathcal{M}) - \tau)^2 - \tau^2}, \frac{2}{\pi}(r_0(\mathcal{M}) - \tau)\sqrt{24\lambda}, r_0(\mathcal{M})\right\}$,

where $s_0(\mathcal{M})$ is the minimum branch separation of $\mathcal{M}$ and $r_0(\mathcal{M})$ is the minimum radius of curvature of $\mathcal{M}$. Assumption 1 ensures that the orthogonal projection $\text{proj}_\mathcal{M} : \text{Tub}_\tau \mathcal{M} \to \mathcal{M}$ is unique, while 2 allows us to use Bernstein et al. (2001) to make statements about geodesic distances.

Our first theorem establishes that the ORC for shortcut edges converges to negative values in the limit of vanishing noise $\sigma$. We note that so long as $\tau > 0$ and $\sigma > 0$, shortcut edges occur with nonzero probability. Observe that, in combination with Proposition 2, this result indicates that the ORC of shortcut edges tends to concentrate in the bottom third of the possible range of values.

**Theorem 3.1** (Ollivier-Ricci Curvature of Shortcut Edges). *Suppose that $\mathcal{X}_i$ is a point cloud sampled from $\rho$ with parameters $\sigma_i$ and $\tau_i$ and $G_i$ is its nearest-neighbor graph. Also suppose that $\mathcal{M}$ satisfies the conditions above, and $\sigma_i \to 0^+$ and $\tau_i \to 0^+$ as $i \to \infty$. Then as $i \to \infty$, we have $\kappa(x, y) \leq -1$ for all shortcut edges $(x, y)$ in $G_i$ with probability approaching 1.*

To be confident that ORC can be effective at identifying potentially shortcutting edges, we would also like to be sure that there exists a sufficient number of good (non-shortcut) edges with more positive ORC. This motivates Theorem 3.2.

**Theorem 3.2** (Ollivier-Ricci Curvature of Non-Shortcut Edges). *Let $k$ be a positive integer. With high probability as the number of points $n \to \infty$, every point has at least $k$ neighbors that it is connected to by non-shortcut edges with ORC $+1$.*

Proofs of Theorem 3.1 and Theorem 3.2 can be found in Appendix A.4.1. These results allow us to create a candidate set of shortcut edges by looking at the ORC of edges in $G$. We then expect that the graph $G' = (V, E')$, where $E'$ has all extremely negative-curvature edges removed, has no

shortcut edges and a large number of good edges. It should therefore have a metric structure that is more aligned with that of the underlying manifold $\mathcal{M}$. With that said, we also expect the candidate set to contain some non-shortcut edges, which we do not want to remove from $G$. In the last part of our algorithm, we filter our candidate set to identify edges that are most likely to be shortcuts. This motivates our third theorem, which establishes that for vertices that were previously connected in $G$ by a shortcut edge, their weighted graph distance in $G'$ is large relative to $\epsilon$.

> **Theorem 3.3** (Filtered Graph Distance). *Suppose that $\mathcal{X}_i$ is a point cloud sampled from $\rho$ with parameters $\sigma_i$ and $\tau_i$ and $G_i = (V_i, E_i)$ is its nearest neighbor graph. Also suppose that $\mathcal{M}$ satisfies the conditions above and $\sigma_i \to 0^+$ and $\tau_i \to 0^+$ as $i \to \infty$. Define the subgraph $G'_i = (V_i, E'_i)$ where*
>
> $$E'_i = \left\{ (x_i, y_i) \in E_i \,\middle|\, \kappa(x_i, y_i) > -1 \right\}.$$
>
> *Then as $i \to \infty$ we have*
>
> $$d_{G'}(x, y) > \beta \frac{\pi(\pi + 1)(1 - \lambda)}{2\sqrt{24\lambda}} \epsilon$$
>
> *for all shortcut edges in $G_i$ with probability approaching 1, where $\beta \in [0, 1]$ (eq. (34)) is a random variable whose distribution is dependent on $\mathcal{M}$ and $\tau_i$.*

> **Remark 1.** *The random variable $\beta$ is inversely related to the number and lengths of edges that shortcut through $\text{Tub}_\tau \mathcal{M}$ but do not satisfy the definition of a shortcut edge in $\mathcal{M}$. We expect that $\beta$ concentrates close to 1, and experimentally we find that $\beta = 1$ works well.*

Theorem 3.3 arises from two results: (1) one can bound geodesic distances through $\text{Tub}_\tau \mathcal{M}$ with geodesic distances of paths through $\mathcal{M}$ with similar endpoints, and (2) under reasonable conditions one can relate the graph distances through a nearest neighbor graph built from data sampled from $\text{Tub}_\tau \mathcal{M}$ to the geodesic distance through $\text{Tub}_\tau \mathcal{M}$.

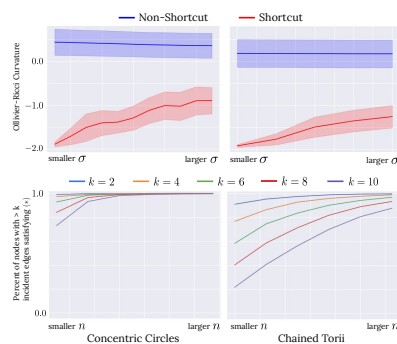

Figure 3 shows experimental support for Theorem 3.1 and Theorem 3.2 on two synthetic manifolds averaged across 10 seeds. In accordance with Theorem 3.1 we find that as the noise parameter $\sigma$ falls, the ORC of shortcutting edges falls commensurately. We see that it rapidly drops below $-1$ and asymptotes to the most negative possible value, $-2$. In accordance with Theorem 3.2 we find that as we sample more points each vertex has an increasing number of non-shortcut incident edges with positive ORC. For more empirical support of the theory we provide an illustrative example in Figure 25 in Appendix A.5.8. We also discuss the behavior of ORC-MANL as a function of underlying manifold curvature in Appendix A.5.7.

Figure 3: Empirical convergence results. The bottom row plots the percent of nodes with at least $k$ incident edges that (∗) *are non-shortcutting and have positive ORC.*

## 4 EXPERIMENTS

Building nearest neighbor graphs as a data pre-processing step is ubiquitous in geometric data analysis. Therefore our method is well suited for evaluation on a broad range of algorithms that operate on nearest neighbor graphs.

Our experiments are divided into two sections. Firstly, we evaluate ORC-MANL on a variety of synthetic manifolds for a broad array of benchmark geometric data analysis and machine learning tasks. We also compare pruning accuracy to several benchmark methods presented previously in the literature. We then show that ORC-MANL pruning reveals clusters aligned with ground truth annotation for single-cell RNA sequencing (scRNAseq) data of peripheral blood mononuclear cells (PBMCs). We also find that ORC-MANL pruning improves downstream manifold learning embeddings of scRNAseq data of anterolateral motor cortex (ALM) brain cells in mice. Furthermore, we include experiments in Appendix A.5.3 that suggest that ORC-MANL is effective on MNIST (Deng, 2012) and KMNIST (Clanuwat et al., 2018), and we add ablations for nearest neighbor graph parameter $k$ and ORC-MANL parameters $\delta, \lambda$ in Appendix A.5.5.

## 4.1 RESULTS: SYNTHETIC DATA

For our synthetic manifolds, we have curated a list of 1 and 2-dimensional manifolds embedded in $\mathbb{R}^2$ and $\mathbb{R}^3$, respectively, that exhibit varying intrinsic and extrinsic curvature. The 1-dimensional manifolds include concentric circles, a mixture of Gaussians, twin moons, an S curve, the 1-dimensional swiss roll, a Cassini oval (Cassini, 1693) and concentric parabolas. The 2-dimensional manifolds include chained tori, concentric hyperboloids, an adjacent hyperboloid and paraboloid, adjacent paraboloids and the 2-dimensional swiss roll. Note that we include additional experiments indicating the improved performance of spectral clustering with ORC-MANL pruning in Appendix A.5.4. We also provide more experimental details regarding sampling and nearest neighbor graph construction in Appendix A.3.

### 4.1.1 PRUNING

Our first experiment quantifies the ability of ORC-MANL to prune (and therefore classify) edges of nearest neighbor graphs. We report classification accuracy for 1-dimensional manifolds in Table 1 and 2-dimensional manifolds in Table 4. In the tables, we also include performance for several baseline graph pruning approaches. Algorithm descriptions for each baseline are provided in Appendix A.3.1. Accompanying visualizations for ORC-MANL pruning results are shown in Figure 4.

Table 1: Pruning performance of ORC-MANL vs. baselines. For each entry, the top and bottom rows indicate the percentage of "good" edges and shortcut edges removed, respectively.

| | Concentric Tori | Concentric Hyperboloids | Hyperboloid and Paraboloid | Paraboloids | Swiss Roll |
|---|---|---|---|---|---|
| ORC-MANL (ours) | $\mathbf{0.0} \pm 0.0$ | $\mathbf{0.0} \pm 0.0$ | $\mathbf{0.0} \pm 0.0$ | $\mathbf{0.0} \pm 0.0$ | $\mathbf{0.0} \pm 0.0$ |
| | $\mathbf{100.0} \pm 0.0$ | $99.1 \pm 2.6$ | $89.4 \pm 21.3$ | $\mathbf{100.0} \pm 0.0$ | $\mathbf{100.0} \pm 0.0$ |
| ORC ONLY | $15.7 \pm 0.3$ | $13.1 \pm 0.2$ | $18.1 \pm 0.3$ | $18.0 \pm 0.3$ | $14.8 \pm 0.2$ |
| | $\mathbf{100.0} \pm 0.0$ | $\mathbf{100.0} \pm 0.0$ | $\mathbf{98.9} \pm 3.3$ | $\mathbf{100.0} \pm 0.0$ | $\mathbf{100.0} \pm 0.0$ |
| BISECTION (Xia et al., 2008) | $2.8 \pm 0.0$ | $0.4 \pm 0.1$ | $0.4 \pm 0.1$ | $0.5 \pm 0.0$ | $0.4 \pm 0.1$ |
| | $8.8 \pm 3.5$ | $20.0 \pm 4.2$ | $61.3 \pm 21.3$ | $10.3 \pm 12.2$ | $32.8 \pm 9.4$ |
| MST (Zemel & Carreira-Perpiñán, 2004; Chao et al., 2007) | $29.2 \pm 0.3$ | $2.3 \pm 0.2$ | $20.4 \pm 0.5$ | $24.3 \pm 0.4$ | $1.8 \pm 0.3$ |
| | $89.6 \pm 3.7$ | $91.6 \pm 1.3$ | $58.5 \pm 10.8$ | $66.2 \pm 17.2$ | $\mathbf{100.0} \pm 0.0$ |
| DENSITY (Chao et al., 2006) | $30.4 \pm 1.6$ | $22.9 \pm 0.9$ | $32.9 \pm 1.1$ | $63.6 \pm 1.3$ | $7.8 \pm 0.7$ |
| | $16.8 \pm 8.0$ | $1.0 \pm 1.3$ | $0.6 \pm 1.9$ | $8.9 \pm 7.4$ | $29.5 \pm 8.0$ |
| DISTANCE | $22.0 \pm 0.4$ | $9.2 \pm 0.5$ | $18.9 \pm 0.4$ | $34.7 \pm 0.5$ | $2.3 \pm 0.2$ |
| | $88.8 \pm 5.6$ | $61.1 \pm 8.7$ | $73.8 \pm 18.2$ | $26.5 \pm 13.7$ | $40.0 \pm 9.8$ |

We find that ORC-MANL vastly outperforms all baselines on a majority of the synthetic manifolds. Our approach removes all shortcut edges in all but two examples, and removes no shortcut edges across all examples. Unsurprisingly, we find that ORC ONLY exhibits identical performance on shortcut edge removal, but typically removes around 15% of good edges as well. BISECTION performs poorly on all manifolds except the moons and the S curve, likely stemming from the fact that shortcut edges for those examples travel longer distances and pass through lower density regions than shortcut edges for other synthetic manifolds.

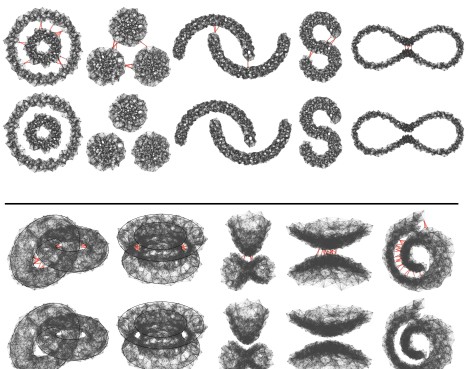

Figure 4: Visualization of nearest neighbor graphs before and after pruning with ORC-MANL. Shortcut edges are highlighted red.

This is also reflected in stronger performance by DISTANCE on the moons and the S curve. MST unsurprisingly performs well on examples where the underlying manifold $\mathcal{M}$ has a single connected component (like the swiss roll and S curve) but consistently struggles with most other examples. Finally, DENSITY and DISTANCE exhibit erratic performance, at times being competitive with ORC-MANL, and at times performing poorly. We attribute this to the variability in the extent to which shortcut edges (1) pass through low density regions, and (2) have longer length relative to other graph edges. Furthermore, these effects can be compounded by poor kernel density estimates for the DENSITY baseline.

We would like to underscore the fact that algorithm parameters for ORC-MANL are fixed at $\delta = 0.8$ and $\lambda = 0.01$ across *all* manifolds and trials (and $\delta$

for ORC ONLY is similarly fixed at 0.8) in Table 1. The baseline methods MST, DENSITY and DISTANCE however required manual threshold tuning for *each* manifold in the evaluation set listed. Despite this additional labor induced by parameter tuning, we find that ORC-MANL outperforms these other methods. All manifold and algorithm parameters are listed in Table 2 and Table 3.

### 4.1.2 MANIFOLD LEARNING

We turn to evaluate ORC-MANL on manifold learning, a family of algorithms concerned with finding low dimensional coordinates on the data submanifold. Almost all manifold learning algorithms begin by constructing a nearest neighbor graph, making ORC-MANL a perfect pre-processing step.

We compare embeddings of nearest neighbor graphs with and without ORC-MANL preprocessing for data sampled from a noisy 2-dimensional swiss roll embedded in $\mathbb{R}^3$ with and without a hole. We test on Isomap (Tenenbaum et al., 2000), Locally Linear Embeddings (LLE), (Belkin & Niyogi, 2003), UMAP (McInnes et al., 2018a) and t-SNE (Van der Maaten & Hinton, 2008) shown in Figure 5. We leave out Laplacian Eigenmaps (Belkin & Niyogi, 2003) as we find that it fails to preserve both underlying dimensions of the noisy swiss roll, independent of the presence of shortcut edges. To the best of our understanding, this likely arises from the *Repeated Eigendirection Problem* (REP) associated with eigenfunction based methods (Dsilva et al., 2018). For UMAP and t-SNE we use graph distances (induced by the pruned and unpruned graphs respectively) as algorithm inputs.

We observe that across both manifolds and all methods, ORC-MANL pruning typically improves the embedding. For Isomap, this improvement is the most pronounced; without pruning, Isomap fails to unroll either swiss roll. We attribute this to distorted geodesic distance estimates (which is the first step of the Isomap algorithm) arising from shortcut edges. We also find significant improvement in the quality of the embeddings for LLE. For both manifolds, LLE without ORC-MANL struggles to preserve the primary dimension of the underlying manifold.

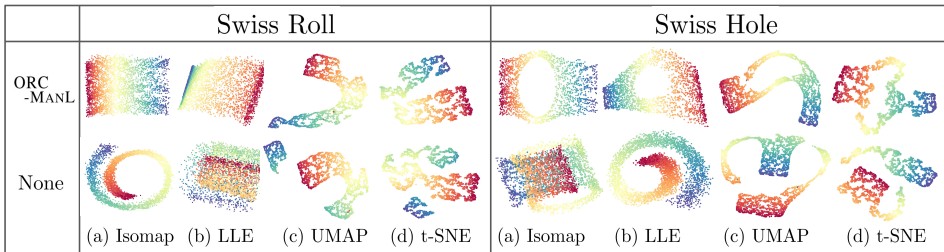

Figure 5: Embeddings of the noisy 3-dimensional swiss roll (left) and noisy 3-dimensional swiss hole (right) produced by several popular manifold learning algorithms. The UMAP embeddings shown were run on noisier samples than those for the other embedding algorithms.

We find that UMAP benefits from ORC-MANL pruning as well, though the difference with and without pruning emerges only when the noise $\tau$ is extremely large. Finally, t-SNE embeddings for both datasets appear to be better when coupled with ORC-MANL, as it prevents discontinuities that arise in the unpruned embeddings. Unfortunately across trials we do not see consistent benefit from pruning, though it is never detrimental. For completeness, embeddings for three different trials are shown in Figure 20 in Appendix A.5.2.

### 4.1.3 PERSISTENT HOMOLOGY

Next, we evaluate our method on persistent homology. Persistent homology begins by constructing a nested sequence $K_{r_0} \subseteq K_{r_1} \subseteq \ldots K_{r_n} \ldots$ of simplicial complexes, also referred to as a *filtered complex*. A common example is the Vietoris-Rips (VR) filtered complex. For data points $\mathcal{X}$ with metric $d$, the VR simplicial complex $K_r$ has a simplex for every subset of points with pairwise distances $\leq r$ (Ghrist, 2014). For our experiments, we construct the VR filtered complex where the metric is the weighted graph distance. One can summarize the result of persistent homology with a persistence diagram, simply a multiset of points in $\overline{\mathbb{R}}_>^2$. Each point $(r_{\text{birth}}, r_{\text{death}})$ in the persistence diagram records the filtration parameter at which a homology class is born and dies respectively.

We present results for persistent homology applied to nearest neighbor graphs with and without ORC-MANL preprocessing in Figure 6. We show results on (1) the noisy concentric circles dataset,

and (2) the noisy chained tori manifolds. For each manifold, we present the persistence diagram of an oversampled, noiseless point cloud of the manifold as a proxy for ground truth. The persistence diagrams are shown for qualitative evaluation, but we also include dissimilarity scores measured as the Wasserstein distance to the noiseless persistence diagram for quantitative evaluation (the details for which are provided in Appendix A.3.2).

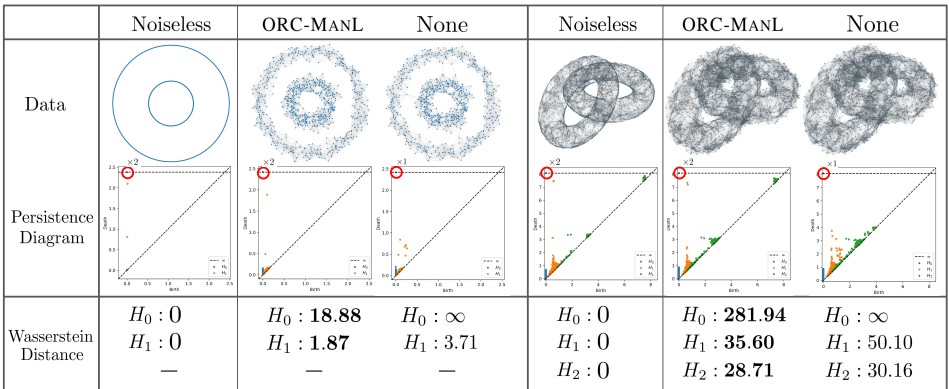

|  | Noiseless | ORC-MANL | None | Noiseless | ORC-MANL | None |
|---|---|---|---|---|---|---|
| Data | | | | | | |
| Persistence Diagram | | | | | | |
| Wasserstein Distance | $H_0 : 0$ $H_1 : 0$ — | $H_0 : \mathbf{18.88}$ $H_1 : \mathbf{1.87}$ — | $H_0 : \infty$ $H_1 : 3.71$ — | $H_0 : 0$ $H_1 : 0$ $H_2 : 0$ | $H_0 : \mathbf{281.94}$ $H_1 : \mathbf{35.60}$ $H_2 : \mathbf{28.71}$ | $H_0 : \infty$ $H_1 : 50.10$ $H_2 : 30.16$ |

Figure 6: Persistent homology applied to the concentric-circles (left) and the chained-tori (right). The bottom row indicates the Wasserstein distance to the noiseless persistence diagram.

We observe that for both datasets, the persistence diagrams computed on ORC-MANL pruned nearest neighbor graphs are qualitatively and quantitatively more similar to the noiseless persistence diagrams than their unpruned counterparts. We emphasize that for both synthetic manifolds, the pruned diagrams correctly capture the number of *essential classes* (homology classes that persist forever), while non-pruned diagrams do not. We observe infinite Wasserstein distances for $H_0$ between the unpruned and noiseless persistence diagrams, but only finite distances for the ORC-MANL pruned diagrams. For other homology dimensions, we also see that ORC-MANL pruned diagrams consistently exhibit smaller distances to the noiseless than non-pruned diagrams.

### 4.1.4 GEOMETRIC DESCRIPTORS

Finally, we test the ability of ORC-MANL to preserve geometric descriptors of the underlying manifolds. We estimate intrinsic dimension and scalar curvature for the (1) swiss roll and (2) adjacent spheres datasets. We compare our estimates to the ground-truth values. We use the maximum-likelihood estimation approach to estimating intrinsic dimension from Levina & Bickel (2004),

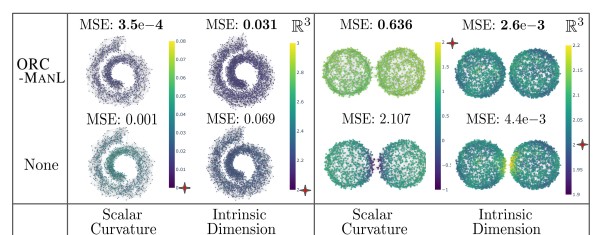

Figure 7: Scalar curvature and intrinsic dimension estimation. The star indicates the ground truth.

and we use the algorithm from Hickok & Blumberg (2023) to estimate scalar curvature. The details for both algorithms are detailed in Appendix A.3.3 and Appendix A.3.4, respectively.

We report intrinsic dimension and scalar curvature estimation results on the noisy swiss roll and noisy adjacent spheres dataset in Figure 7. We find that ORC-MANL pruning provides tangible improvement over non-pruned estimates, especially for scalar curvature. For intrinsic dimension estimation we see improved point-wise accuracy in regions with a high number of shortcut edges (for example, the area between the adjacent spheres); in regions with a smaller presence of shortcut edges, we find that the unpruned graphs produce accurate results.

### 4.2 RESULTS: REAL DATA

We evaluate the efficacy of ORC-MANL on real-world data by using it to analyze cell-type annotated scRNAseq data of (1) anterolateral motor cortex (ALM) brain cells in mice available from the Allen Institute (Abdelaal et al., 2019), and (2) scRNAseq data of peripheral blood mononuclear

cells (PBMC) available from 10XGenomics. In accordance with previous work, for both datasets we extract the 2000 most variable genes, followed by PCA (Pearson, 1901) to obtain a 50-dimensional representation of the original data.

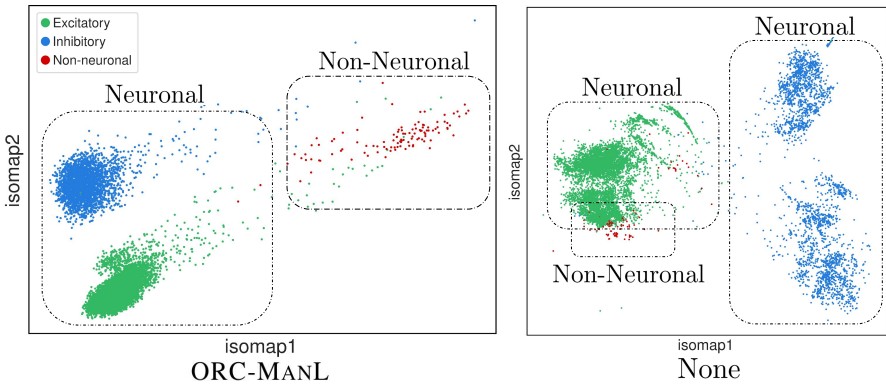

Figure 8: Isomap (Tenenbaum et al., 2000) embedding of scRNAseq data from anterolateral motor cortex (ALM) brain cells in mice with and without ORC-MANL pruning.

For the brain cells we use Isomap (Tenenbaum et al., 2000) to embed ORC-MANL pruned and unpruned nearest neighbor graphs. Figure 8 shows the embeddings with base truth annotations. We find that pruning with ORC-MANL qualitatively improves the Isomap embedding of the data, as the distinction between neuronal cells (labeled "Inhibitory" and "Excitatory") and non-neuronal cells becomes significantly more pronounced after pruning. For completeness, we include UMAP and t-SNE embeddings of this dataset in the appendix in Figure 23; we observe poor performance as measured by neuronal community preservation both with and without pruning, suggesting issues with the embedding algorithms themselves as opposed to pruning.

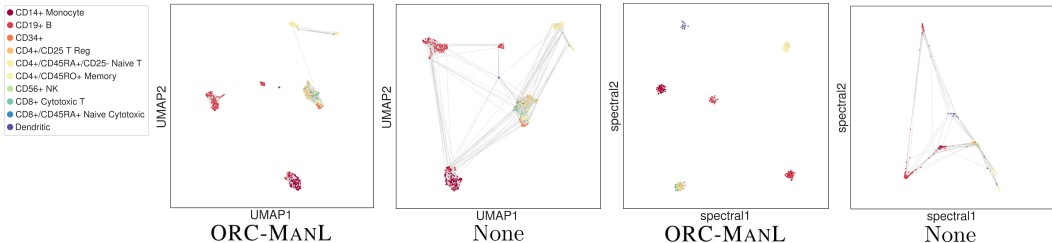

Figure 9: UMAP and spectral embeddings of PBMC scRNAseq data with and without ORC-MANL preprocessing, with edges of the pruned and unpruned graphs visualized in grey. Gaussian noise was added to the spectral embeddings with ORC-MANL pruning for visibility, as connected components get mapped to a single point.

Now we turn to the PBMC dataset, where we show UMAP and spectral embeddings in Figure 9. We find that ORC-MANL pruning leads to connected components that largely align with base truth annotations; this is reflected in the spectral embedding, which maps each component to a unique location in the embedding space. Furthermore, edges that formerly bridged seemingly distinct clusters in the embeddings were removed, resulting in clearer cluster structure in the pruned graphs.

## 5 CONCLUSION

In this work we present ORC-MANL, a theoretically justified and empirically validated approach for nearest neighbor graph pruning. We rigorously demonstrate that shortcut edges necessarily exhibit particularly negative ORC for graphs sampled from manifolds with a bounded level of isotropic noise. Our method also validates the removal of any edge by using a theoretically derived bound on graph distances implied by the geometry of shortcut edges. We qualitatively and quantitatively demonstrate that ORC-MANL is beneficial to a variety of downstream geometric data analysis tasks on synthetic and real data. We also compare our method to baselines from the literature in its ability to correctly prune shortcut edges and avoid non-shortcut edges.

## ACKNOWLEDGEMENTS

The authors would like to thank Yining Liu for her invaluable advice on the real data experiments. Furthermore, they would like to thank Gilad Turok, Gabriel Guo, and Nakul Verma for their insightful thoughts and discussions about the manuscript and the work. Abigail Hickok was supported by NSF grant DMS-2303402. Andrew J. Blumberg was supported in part by NFS grant DMS-2311338 and ONR grant N00014-22-1-2679.

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

# A  APPENDIX

## A.1  REPRODUCIBILITY STATEMENT

**Theoretical Results:** We detail global assumptions for all theoretical results in Section 3. We also make a concerted effort to clarify all proof-specific assumptions in the Theorem, Lemma and Proposition statements.

**Experimental Results:** To ensure the reproducibility of our experimental results we release the source code for ORC-MANL and the scripts for reproducing all experiments.[*] Included in the experiment scripts are the parameters used; for clarity we describe key parameter choices throughout the body of the paper, and provide extra details in Appendix A.3.

## A.2  ORC-MANL TIME COMPLEXITY

ORC-MANL consists of two sequential stages. In the first stage the ORC of each edge is evaluated. According to Fesser & Weber (2024) such an operation has complexity $\mathcal{O}(|E|d_{\max}^3)$, where $d_{\max}$ indicates the highest node degree in the graph. Assuming $k$-NN connectivity (which is commonplace in practice), we have $d_{\max} = \mathcal{O}(k)$ and $|E| = \mathcal{O}(|V|k)$. This means the first stage of the algorithm has complexity $\mathcal{O}(|V|k^4)$. The second stage of the ORC-MANL algorithm requires one to find the shortest path distance $d_{G'}(x, y)$ between the endpoints of all candidate edges $(x, y) \in C$ in the ORC thresholded graph $G'$. With a naive implementation of Dijkstra's algorithm the time complexity is $\mathcal{O}(|C||V|^2)$, but efficient implementations admit a time complexity of $\mathcal{O}(|C||V|\log|V|)$. Finally since $C \subset E$ and $|E| = \mathcal{O}(|V|k)$, the time complexity of the second stage of ORC-MANL scales as $\mathcal{O}(k|V|^2 \log|V|)$.

The total time complexity of the ORC-MANL algorithm therefore scales as $\mathcal{O}(|V|k^4) + \mathcal{O}(k|V|^2 \log|V|)$. Seeing as many geometric data analysis algorithms (e.g., PCA) require eigendecompositions that scale as $\mathcal{O}(|V|^3)$, we consider the runtime of ORC-MANL to be reasonable. For empirical evidence to support this claim, we provide experiments that compare the wall-clock time of ORC-MANL and UMAP for the Swiss Roll in Figure 10. For this experiment, both algorithms were run on a 16-core machine with 187 GB of RAM. We use the UMAP implementation from McInnes et al. (2018b) and the ORC implementation from Ni et al. (2019), both of which use

---

[*]https://github.com/TristanSaidi/orcml

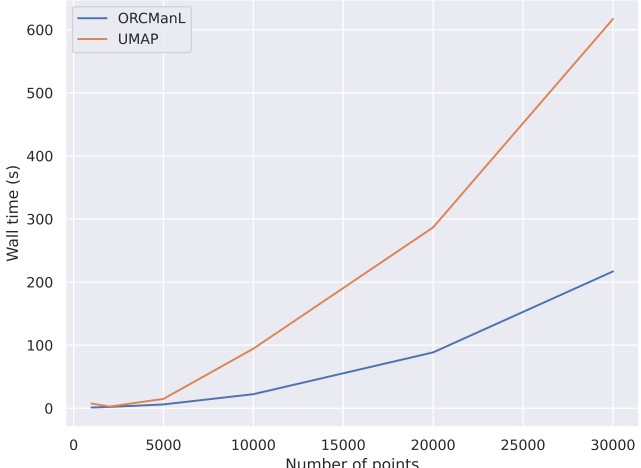

Figure 10: Wall clock time of the ORC-MANL algorithm and the UMAP algorithm for the noisy Swiss Roll dataset of varying size.

multiprocessing. We see that in this range ORC-MANL runs substantially faster than UMAP and appears to have slower asymptotic growth.

## A.3 EXPERIMENTAL DETAILS

For all experiments in Section 4.1, datapoints are sampled according to $\rho$ defined in eq. (1) for each synthetic manifold $\mathcal{M}$. To achieve this, we sample uniformly from the underlying $m$-dimensional manifold $\mathcal{M}$ (unless otherwise specified) driven by the $m$-dimensional volume form defined by the Euclidean metric. We then use rejection sampling to obtain isotropic Gaussian noise $\xi \sim \mathcal{N}(0, \mathbf{I})$ such that $\|\xi\|_2 \leq \tau$. For all experiments detailed in Section 4.1, we use $k$-NN connectivity; while our theoretical analysis assumes $\epsilon$-radius connectivity, $\epsilon$-radius connectivity is more challenging to tune in practice. Since $k$-NN graphs are more commonplace in the applied literature, we opt to present results on $k$-NN graphs (unless otherwise specified).

### A.3.1 PRUNING

Here we detail each of the pruning baselines that appear in Table 1.

1. ORC ONLY: This baseline returns a graph $G'$ where edges with curvature less than $-1 + 4(1 - \delta)$ are removed, but no additional validation step is undertaken. Namely, Algorithm 1 is terminated at line 7 and the graph $G'$ is returned.

2. BISECTION: This method was proposed by Xia et al. (2008) to address topological instability associated with the Isomap algorithm (Tenenbaum et al., 2000). For each edge $(x, y)$, it performs a local search around the midpoint $l = (x + y)/2$. Namely, it searches in a bounding box $\prod_{i=1}^{D}[l_i - \epsilon', l_i + \epsilon']$ for a neighboring point, where $\epsilon'$ is the average distance of $x$ and $y$ to their $k_{\text{bis}}$ nearest neighbors. If a point exists in the bounding box, the edge is removed.

3. MST: This method was proposed by Zemel & Carreira-Perpiñán (2004) and simplified by Chao et al. (2006). In essence, this leverages the insight that shortcutting edges empirically exhibit longer edge length than their non-shortcutting counterparts (though neither paper presented any theoretical justification for such claims). The method builds a Minimum-Spanning-Tree (MST) $T$ of the original graph $G$ with Kruskal's algorithm (Kruskal, 1956). The edges in this MST are removed to obtain $G' = (V, E \setminus T)$, from which one obtains another MST $T'$. Finally, $T$ and $T'$ are combined to get $G''$. If for any edge $(x, y)$ in $G$ the

      graph shortest path distance in $G''$ is larger than some manually set threshold $d_{\text{MST}}$, then it is removed.

4. DENSITY: While similar in theme to BISECTION, DENSITY uses explicit kernel density estimation inspired by Chao et al. (2007). For this baseline, we estimate the density at the midpoint of all edges, and remove edges that exhibit especially low density (based on some manually tuned threshold $\rho_{\text{min}}$).

5. DISTANCE: This baseline simply removes the longest edges based on some manually tuned threshold $d_{\text{dist}}$.

We note that for manifolds with a single connected component, we estimate edge labels (shortcut versus non-shortcut) by checking whether the ratio of the manifold distance to the Euclidean distance is large for a denser noiseless sample. For reproducibility we include noise parameters used for manifold sampling in Table 2. We also include algorithm parameters for the baselines in Table 3. As mentioned earlier in the body of the paper, all ORC-MANL experiments use $\delta = 0.8$ and $\lambda = 0.01$. Similarly, all ORC ONLY experiments use $\delta = 0.8$ as well.

Table 2: Manifold noise parameters $\tau$ and $\sigma$ for each manifold used in the pruning evaluation (Figure 4).

| $\mathcal{M}$ | $\tau$ | $\sigma$ |
|---|---|---|
| Concentric Circles | 0.28 | 0.09 |
| Mix. of Gaussians | 0.45 | 0.18 |
| Moons | 0.19 | 0.20 |
| S curve | 0.52 | 0.28 |
| Cassini | 0.135 | 0.05 |
| Tori | 0.75 | 0.4 |
| Hyperboloids | 0.25 | 0.20 |
| Hyp. and Parab. | 0.48 | 0.40 |
| Paraboloids | 0.70 | 0.60 |
| Swiss Roll | 2.25 | 6.25 |

Table 3: Algorithm parameters for each baseline and each manifold used in the pruning evaluation (Figure 4).

| $\mathcal{M}$ | $k_{\text{bis}}$ | $d_{\text{MST}}$ | $\rho_{\text{min}}$ | $d_{\text{dist}}$ |
|---|---|---|---|---|
| Concentric Circles | 10 | 0.5 | 0.125 | 0.15 |
| Mix. of Gaussians | 10 | 0.3 | 0.15 | 0.15 |
| Moons | 10 | 0.5 | 0.175 | 0.15 |
| S curve | 10 | 0.5 | 0.05 | 0.15 |
| Cassini | 10 | 0.5 | 0.20 | 0.09 |
| Tori | 10 | 1.5 | 0.0007 | 1.0 |
| Hyperboloids | 10 | 1.5 | 0.015 | 0.4 |
| Hyp. and Parab. | 10 | 0.75 | 0.015 | 0.4 |
| Paraboloids | 10 | 1.0 | 0.01 | 0.5 |
| Swiss Roll | 10 | 10 | 0.00005 | 3.0 |

### A.3.2 PERSISTENCE DISTANCES

We compute distances between persistence diagrams using the approach presented in Lacombe et al. (2018). The $p$-Wasserstein distance is defined as

$$d_p(D_1, D_2) = \left( \min_{\zeta \in \Gamma(D_1, D_2)} \sum_{(x,y) \in \zeta} \|x - y\|_p^p + \sum_{s \in D_1 \cup D_2 \setminus \zeta} \|s - \pi_\Delta(s)\|_p^p \right)^{\frac{1}{p}}$$

where $D_1$ and $D_2$ are persistence diagrams represented as a set of points in $\bar{\mathbb{R}}_>^2$, $\Gamma(D_1, D_2)$ is the set of all partial matchings between points in $D_1$ and points in $D_2$, and $\pi_\Delta(s)$ denotes the orthogonal

projection of an unmatched point to the diagonal. We use The GUDHI Project (2020) to compute persistence distances with $p = \infty$.

### A.3.3 INTRINSIC DIMENSION ESTIMATION

We use the maximum-likelihood based approach for estimating intrinsic dimension proposed in Levina & Bickel (2004). The method assumes the data points are generated from a homogeneous Poisson point process on a manifold. From there, they derive the maximum likelihood estimator for the intrinsic dimension $m$, with

$$\hat{m}_k(x) = \left[ \frac{1}{k-1} \sum_{j=1}^{k-1} \log \frac{T_k(x)}{T_j(x)} \right]^{-1}$$

where $T_j(x)$ represents the distance of $x$ to its $j$-th nearest neighbor and $k$ is a user-chosen parameter. In our experiments, we report the mean-squared-error (MSE) between pointwise intrinsic dimension estimates and the base truth value for the underlying manifold. While it has been pointed out that the global intrinsic dimension estimate derived in Levina & Bickel (2004) is biased, the pointwise estimators are, in fact, unbiased. For our experiments we extract nearest neighbors and nearest neighbor distances using the graph metric. We also use $k = 200$ and $n = 4000$ (where $n$ is the number of points in the sample) for the experiments in Figure 7.

### A.3.4 SCALAR CURVATURE ESTIMATION

Scalar curvature assigns a scalar quantity to each point of a Riemannian manifold. It is intimately related to the notion of sectional curvature through the equation $S(x) = \sum_{i \neq j} \mathrm{Sec}(e_i, e_j)$, where $\mathrm{Sec}(\cdot)$ is the sectional curvature and $e_1, \ldots, e_d$ form an orthonormal basis of the manifold's tangent space at $x$. Note that for surfaces (manifolds with intrinsic dimension 2) the scalar curvature is simply twice the Gaussian curvature. We use the algorithm from Hickok & Blumberg (2023) to estimate $S(x)$. This method leverages the following result, which relates scalar curvature of an $m$ dimensional manifold $\mathcal{M}$ to the volume of geodesic balls relative to Euclidean geodesic balls. It states,

$$\frac{\mathrm{Vol}\big( B_r(x) \big) \subset \mathcal{M}}{\mathrm{Vol}\big( B_r(0) \big) \subset \mathbb{R}^m} = 1 - \frac{S(x)}{6(d+2)} r^2 + \mathcal{O}(r^3)$$

for sufficiently small $r$. The method first builds a nearest neighbor graph and uses the induced graph metric to estimate geodesic ball volumes at a range of scales. From there a point-wise estimate $\hat{S}$ of scalar curvature can be produced via regression over the volume ratios over some range $[0, r_{\max}]$. In our experiments in Figure 7 we use $n = 4000$ points and $r_{\max} = 50$ for the swiss roll and $r_{\max} = 5$ for the adjacent spheres.

### A.4 PROOFS

#### A.4.1 THEOREMS 3.1, 3.2 AND 3.3

In this first section we will detail the proofs of the theorem statements put forth in Section 3. For clarity, recall the assumptions: we consider the setting where $\mathcal{M}$ is a compact $m$-dimensional smooth submanifold of $\mathbb{R}^D$ without boundary. Let $\mathrm{Tub}_\tau(\mathcal{M})$ be the tubular neighborhood of $\mathcal{M}$, and assume $\mathcal{X} \subset \mathrm{Tub}_\tau(\mathcal{M})$ consists of $n$ independent draws from the probability density function $\rho : \mathrm{Tub}_\tau(\mathcal{M}) \to \mathbb{R}_+$,

$$\rho(z) = \begin{cases} \dfrac{1}{Z} e^{\frac{-\|z - \mathrm{proj}_{\mathcal{M}} z\|_2^2}{2\sigma^2}} & \|z - \mathrm{proj}_{\mathcal{M}} z\|_2 \leq \tau \\ 0 & \text{o.w.} \end{cases}$$

where $Z$ is a normalizing constant such that $\int_{\mathrm{Tub}_\tau(\mathcal{M})} \rho(z) dV$ integrates to 1. We are given the constant $\lambda < 1$. For the rest of the paper, suppose

1. (*Support criteria*): $2\tau < \epsilon$, $3\tau < s_0(\mathcal{M})$, $3\tau < r_0(\mathcal{M})$

2. ($\epsilon$-*radius criterion*): $\epsilon < \min\left\{\sqrt{(s_0(\mathcal{M}) - \tau)^2 - \tau^2}, \frac{2}{\pi}(r_0(\mathcal{M}) - \tau)\sqrt{24\lambda}, r_0(\mathcal{M})\right\}$

where $s_0(\mathcal{M})$ is the minimum branch separation of $\mathcal{M}$ and $r_0(\mathcal{M})$ is the minimum radius of curvature of $\mathcal{M}$. We can now begin to dissect the argument for Theorem 3.1.

**Theorem 3.1** (Ollivier-Ricci Curvature of Shortcut Edges). *Suppose that $\mathcal{X}_i$ is a point cloud sampled from $\rho$ with parameters $\sigma_i$ and $\tau_i$ and $G_i$ is its nearest-neighbor graph. Also suppose that $\mathcal{M}$ satisfies the conditions above, and $\sigma_i \to 0^+$ and $\tau_i \to 0^+$ as $i \to \infty$. Then as $i \to \infty$, we have $\kappa(x, y) \leq -1$ for all shortcut edges $(x, y)$ in $G_i$ with probability approaching 1.*

This theorem establishes that in the limit of vanishing noise, the ORC of shortcut edges is necessarily upper bounded by $-1$. The theorem stitches together Lemmas A.3, A.4 and A.1. Here is how we will approach it.

Simply put, one can show that all shortcut edges are necessarily close to a normal direction to the manifold at both endpoints; if this was not the case, the edge would likely intersect the manifold earlier and thus no longer bridge extremely distant neighborhoods. This concept is formalized and proven in Lemma A.3.

Given that shortcut edges are close to normal, one can show that most of the measure of epsilon balls around the endpoints concentrates far from the opposing endpoint (assuming the pdf $\rho$ described above). It follows that the neighborhoods of $x$ and $y$ will tend to concentrate far from each other. An example visualization of this phenomenon is provided in Figure 11.

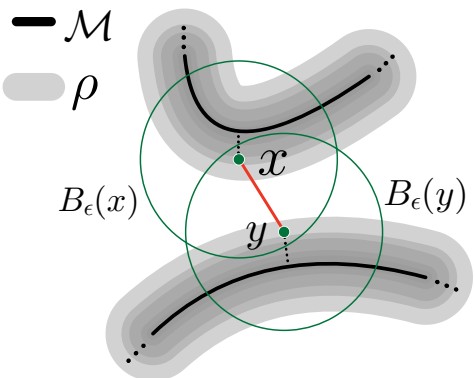

Figure 11: Visualizations of the $\epsilon$-balls at the endpoints of a shortcut edge $(x, y)$.

This divergence between the neighborhoods of $x$ and $y$ leads to a large Wasserstein distance between the neighborhoods, implying more negative ORC. This fact is formalized and shown in Lemma A.1. With the high-level roadmap in place, we will proceed by proving Theorem 3.1.

*Proof of Theorem 3.1.* Consider a sequence of point clouds $\{\mathcal{X}_i\}_{i=1}^{\infty}$ with $n$ points each, where each point cloud was sampled from $\rho$ (defined in eq. (1)) with parameters $\sigma_i$ and $\tau_i$. Also suppose $\sigma_i \to 0$ and $\tau_i \to 0$ as $i \to \infty$. Consider the $i$-th pointcloud in this sequence, and let $x_i$ and $y_i$ be any two points connected by a shortcut edge (if they exist).

Since $(x_i, y_i)$ is a shortcut edge, we can apply Lemma A.3 to find the vector $v_{x_i} \in N_{\text{proj}_{\mathcal{M}} x_i}\mathcal{M}$ such that the angle $\phi_{x_i}$ between $v_{x_i}$ and $(y_i - x_i)$ is less than

$$\Phi(\tau_i) = \arccos\left(\frac{(r_0(\mathcal{M}) - \tau_i)(s_0(\mathcal{M}) - \tau_i) - 1.27 r_0(\mathcal{M})\tau_i}{r_0(\mathcal{M})\sqrt{(s_0(\mathcal{M}) - \tau_i)^2 - \tau_i^2}}\right) < \pi/2 \qquad (2)$$

and the angle $\theta_{x_i}$ between $(x_i - \text{proj}_{\mathcal{M}} x_i)$ and $v_{x_i}$ is less than $\pi/2$. In an identical manner, A.3 can be applied to find analogous quantities $v_{y_i}$, $\theta_{y_i}$ and $\phi_{y_i}$. The fact that $\phi_{x_i}$ and $\phi_{y_i}$ are necessarily bounded by 2 formalizes the notion that shortcut edges are necessarily oriented close to normal with respect to the manifold at each endpoint.

Now we would like to say something about the degree to which the neighborhoods of $x_i$ and $y_i$ are likely to overlap. To do so, we define the sets $U_{x_i} \subset B_\epsilon(x_i)$ and $U_{y_i} \subset B_\epsilon(y_i)$ such that for all $a, b \in U_{x_i} \times U_{y_i}$, $\|a - b\|_2 > \epsilon$. One particular instantiation of these sets is shown in Figure 12. For the analysis that follows we will consider this instantiation, namely the case where the boundary of $U_{x_i}$ is defined by a hyperplane orthogonal to $(y_i - x_i)$ (and $U_{y_i}$ is defined analogously). In this instance, the sets $U_{x_i}$ and $U_{y_i}$ are simply hyperspherical caps.

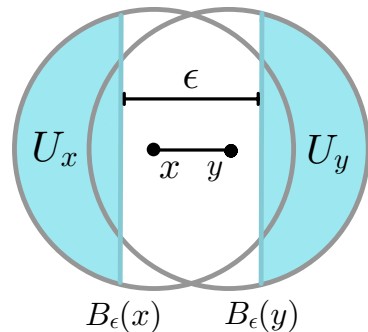

Figure 12: Visualization of the sets $U_x$ and $U_y$ for an edge $(x, y)$.

Now lets also define $p_{\sigma_i}(x_i) := \mathbb{P}[\, a \in U_{x_i} \,|\, a \in B_\epsilon(x_i)]$ and $p_{\sigma_i}(y_i) := \mathbb{P}[\, b \in U_{y_i} \,|\, b \in B_\epsilon(y_i)]$. We can use Lemma A.4 to say

$$\lim_{\sigma_i \to 0^+} p_{\sigma_i}(x_i) \geq \frac{\int_{f_{x_i}(\phi_{x_i})}^{\sqrt{\epsilon^2 - \|x_i - \mathrm{proj}_\mathcal{M} x_i\|_2^2}} \mathrm{Vol}(B_{r_{x_i}(z)}^{m-1}(0))dz}{\mathrm{Vol}(B_{R_{x_i}}^m(0))}$$

and

$$\lim_{\sigma_i \to 0^+} p_{\sigma_i}(y_i) \geq \frac{\int_{f_{y_i}(\phi_{y_i})}^{\sqrt{\epsilon^2 - \|y_i - \mathrm{proj}_\mathcal{M} y_i\|_2^2}} \mathrm{Vol}(B_{r_{y_i}(z)}^{m-1}(0))dz}{\mathrm{Vol}(B_{R_{y_i}}^m(0))}$$

where

$$f_{x_i}(\phi_{x_i}) = \cot(\phi_{x_i})\left( \sec(\phi_{x_i})\left( \frac{\epsilon - \|x_i - y_i\|_2}{2} \right) - \|x_i - \mathrm{proj}_\mathcal{M} x_i\|_2 \cos(\theta_{x_i}) \right)$$

and

$$f_{y_i}(\phi_{y_i}) = \cot(\phi_{y_i})\left( \sec(\phi_{y_i})\left( \frac{\epsilon - \|x_i - y_i\|_2}{2} \right) - \|y_i - \mathrm{proj}_\mathcal{M} y_i\|_2 \cos(\theta_{y_i}) \right).$$

Note that $R_{x_i} = \sqrt{\epsilon^2 - \|x_i - \mathrm{proj}_\mathcal{M} x_i\|_2^2}$ and

$$r_{x_i}(z) = \begin{cases} \sqrt{\epsilon^2 - \|x_i - \mathrm{proj}_\mathcal{M} x_i\|_2^2 - z^2} & |z| \leq \sqrt{\epsilon^2 - \|x_i - \mathrm{proj}_\mathcal{M} x_i\|_2^2} \\ 0 & \text{otherwise} \end{cases}$$

are defined in Lemma A.4, with analogous quantities defined for $y_i$. Observe that the further the edge $(x_i, y_i)$ is from normal with respect to either endpoint (as measured by $\phi_{x_i}$ and $\phi_{y_i}$), the smaller $p_{\sigma_i}(x_i)$ and $p_{\sigma_i}(y_i)$ are respectively. This follows from the fact that $f_{x_i}$ and $f_{y_i}$ are increasing functions of $\phi_{x_i}$ and $\phi_{y_i}$ respectively. This can be confirmed by a quick visual inspection of Figure 11 - the further from normal $(x - y)$ is, the less probability density exists in $U_x$ and $U_y$.

Now we will show that $f_{x_i}$ and $f_{y_i}$ are increasing in $\phi_{x_i}$ and $\phi_{y_i}$ when in the interval $(0, \pi/2]$ (which is always the case as shown in A.3). The derivative of $\cot(c)(a \sec(c) - b)$ is $(b - a\cos(c))/\sin^2(c)$. Note that the derivative is positive on $(0, \pi/2]$ when $a \leq b$. From eq. (37) in Lemma A.3 we know that

$$\|x_i - y_i\|_2 \geq \sqrt{(s_0(\mathcal{M}) - \tau_i)^2 - \|x_i - \mathrm{proj}_\mathcal{M} x_i\|_2^2 \sin^2(\theta_i)} - \|x_i - \mathrm{proj}_\mathcal{M} x_i\|_2 \cos(\theta_i).$$

The right-hand side is bounded from below by $\epsilon - \|x_i - \text{proj}_{\mathcal{M}} x_i\|_2 \cos(\theta_i)$ due to assumption 2 and the fact that $\tau_i \geq \|x_i - \text{proj}_{\mathcal{M}} x_i\|_2$. Thus $b = \|x_i - \text{proj}_{\mathcal{M}} x_i\|_2 \cos(\theta_i) > \epsilon - \|x_i - y_i\|_2 = 2a$, rendering the derivative positive. Since $\Phi(\tau_i)$ is larger than all possible $\phi_{x_i}$ and $\phi_{y_i}$ we can say $f_{x_i}(\Phi(\tau_i)) \geq f_{x_i}(\phi_{x_i})$ and $f_{y_i}(\Phi(\tau_i)) \geq f_{y_i}(\phi_{y_i})$, allowing us to conclude that

$$\lim_{\sigma_i \to 0^+} p_{\sigma_i}(x_i) \geq \frac{\int_{f_{x_i}(\Phi(\tau_i))}^{\sqrt{\epsilon^2 - \|x_i - \text{proj}_{\mathcal{M}} x_i\|_2^2}} \text{Vol}(B_{r_{x_i}(z)}^{m-1}(0)) dz}{\text{Vol}(B_{R_{x_i}}^m(0))}$$

where an analogous bound holds for $\lim_{\sigma_i \to 0^+} p_{\sigma_i}(y_i)$. Taking the limit as $\tau_i \to 0^+$ yields

$$\lim_{\tau_i \to 0^+} \lim_{\sigma_i \to 0^+} p_{\sigma_i}(x_i) \tag{3}$$

$$\geq \lim_{\tau_i \to 0^+} \frac{\int_{f_{x_i}(\Phi(\tau_i))}^{\sqrt{\epsilon^2 - \|x_i - \text{proj}_{\mathcal{M}} x_i\|_2^2}} \text{Vol}(B_{r_{x_i}(z)}^{m-1}(0)) dz}{\text{Vol}(B_{R_{x_i}}^m(0))} \tag{4}$$

$$= \lim_{\tau_i \to 0^+} \frac{\int_{-\infty}^{\infty} \text{Vol}(B_{r_{x_i}(z)}^{m-1}(0)) \cdot \mathbb{1}\left[z \in \left[f_{x_i}(\Phi(\tau_i)), \sqrt{\epsilon^2 - \|x_i - \text{proj}_{\mathcal{M}} x_i\|_2^2}\right]\right] dz}{\text{Vol}(B_{R_{x_i}}^m(0))}$$

$$\tag{5}$$

$$= \frac{\int_{f_{x_i}(\lim_{\tau_i \to 0^+} \Phi(\tau_i))}^{\sqrt{\epsilon^2 - \|x_i - \text{proj}_{\mathcal{M}} x_i\|_2^2}} \text{Vol}(B_{r_{x_i}(z)}^{m-1}(0)) dz}{\text{Vol}(B_{R_{x_i}}^m(0))}. \tag{6}$$

In the second line we can use the dominated convergence theorem to pull $\lim_{\tau_i \to 0^+}$ into the limits of the integral, since the integrand is bounded above by the integrable function $\text{Vol}(B_\epsilon^{m-1}(0)) \cdot \mathbb{1}[z \in [-\epsilon, \epsilon]]$ and bounded below by $0 \ \forall \tau_i > 0$. Now we can further evaluate

$$\lim_{\tau_i \to 0^+} \Phi(\tau_i) = \Phi(0)$$

$$= \arccos\left(\frac{r_0(\mathcal{M})s_0(\mathcal{M})}{r_0(\mathcal{M})s_0(\mathcal{M})}\right)$$

$$= 0.$$

This implies $\lim_{\tau_i \to 0^+} f_{x_i}(\Phi(\tau_i)) = -\infty$ since $f_{x_i} \to -\infty$ as its argument approaches 0. Note $f_{x_i} \to -\infty$ as its argument approaches 0 because as $c \to 0^+$, $a \sec(c) - b \cos(c)$ converges to $a - b$ (which we have established is strictly negative) and $\cot(c)$ converges to $+\infty$. Now, since $r_{x_i}(z)$ in the integrand of 6 is 0 for values less than $-\sqrt{\epsilon^2 - \|x_i - \text{proj}_{\mathcal{M}} x_i\|_2^2}$, we can say that

$$\lim_{\tau_i \to 0^+} \lim_{\sigma_i \to 0^+} p_{\sigma_i}(x_i) \geq \frac{\int_{-\sqrt{\epsilon^2 - \|x_i - \text{proj}_{\mathcal{M}} x_i\|_2^2}}^{\sqrt{\epsilon^2 - \|x_i - \text{proj}_{\mathcal{M}} x_i\|_2^2}} \text{Vol}(B_{r_{x_i}(z)}^{m-1}(0)) dz}{\text{Vol}(B_{R_{x_i}}^m(0))} \tag{7}$$

$$= \frac{\text{Vol}(B_{R_{x_i}}^m(0))}{\text{Vol}(B_{R_{x_i}}^m(0))} \tag{8}$$

$$= 1. \tag{9}$$

Note that the same argument can be used to prove the same equality on $p_{\sigma_i}(y_i)$ in the limit. Now we would like to use these results to make a statement about the number of neighbors that end up falling in either $U_{x_i}$ or $U_{y_i}$ for $n$ i.i.d samples from $\rho$ with parameters $\sigma_i$ and $\tau_i$ (eq. (1)). Define the sets $S_{x_i} = \{a \mid a \in \mathcal{N}(x_i) \cap U_{x_i}\}$ and $S_{y_i} = \{b \mid b \in \mathcal{N}(y_i) \cap U_{y_i}\}$. Also let $\mathcal{N}_{x_i} = |\mathcal{N}(x_i) \setminus y_i|$ and $\mathcal{N}_{y_i} = |\mathcal{N}(y_i) \setminus x_i|$. We can say,

$$\mathbb{P}\left[|S_{x_i}| = \mathcal{N}_{x_i}\right] = p_{\sigma_i}(x_i)^{\mathcal{N}_{x_i}} \geq p_{\sigma_i}(x_i)^n \tag{10}$$

since $\mathcal{N}_{x_i} \leq n$. Through application of 9 we can conclude

$$\lim_{i \to \infty} \mathbb{P}\left[|S_{x_i}| = \mathcal{N}_{x_i}\right] = 1. \tag{11}$$

With the same argument, we can say

$$\mathbb{P}\Big[|S_{y_i}| = \mathcal{N}_{y_i}\Big] \geq p_{\sigma_i}(y_i)^n \tag{12}$$

and thus,

$$\lim_{i \to \infty} \mathbb{P}\Big[|S_{y_i}| = \mathcal{N}_{y_i}\Big] = 1. \tag{13}$$

Now we want to use 10 and 12 to bound $\mathbb{P}\big[|S_{x_i}| = \mathcal{N}_{x_i}, |S_{y_i}| = \mathcal{N}_{y_i}\big]$. Define $\alpha_{x_i} = \mathbb{1}[|S_{x_i}| = \mathcal{N}_{x_i}]$ and $\alpha_{y_i} = \mathbb{1}[|S_{y_i}| = \mathcal{N}_{y_i}]$. We can apply a union bound to say

$$\begin{aligned}
\mathbb{P}\big[\alpha_{x_i} = 1, \alpha_{y_i} = 1\big] &\geq 1 - \sum_{\alpha_{x_i} \neq 1 \,\vee\, \alpha_{y_i} \neq 1} \mathbb{P}\big[\alpha_{x_i}, \alpha_{y_i}\big] \\
&\geq 1 - \sum_{\alpha_{x_i} \neq 1 \,\vee\, \alpha_{y_i} \neq 1} \min\big\{\mathbb{P}[\alpha_{x_i}], \mathbb{P}[\alpha_{y_i}]\big\} \\
&= 1 - \min\Big\{\mathbb{P}[\alpha_{x_i} = 1], \, 1 - \mathbb{P}[\alpha_{y_i} = 1]\Big\} \\
&\quad - \min\Big\{1 - \mathbb{P}[\alpha_{x_i} = 1], \, \mathbb{P}[\alpha_{y_i} = 1]\Big\} \\
&\quad - \min\Big\{1 - \mathbb{P}[\alpha_{x_i} = 1], \, 1 - \mathbb{P}[\alpha_{y_i} = 1]\Big\}.
\end{aligned}$$

Taking the limit and applying 11 and 13 yields

$$\lim_{i \to \infty} \mathbb{P}\big[\alpha_x = 1, \alpha_y = 1\big] = 1.$$

This result crystallizes the notion that, in the limit of vanishing noise, the neighborhoods of the endpoints of shortcut edges tend to concentrate very far from each other. To restate formally, we have shown that $\mathbb{P}[|S_{x_i}| = \mathcal{N}_{x_i}, |S_{y_i}| = \mathcal{N}_{y_i}]$ approaches 1 in the limit.

Now we will apply Lemma A.1 to control the ORC of the edge $(x_i, y_i)$. The Lemma proves that $|S_{y_i}| = \delta_{y_i} \mathcal{N}_{y_i}$ and $|S_{x_i}| = \delta_{x_i} \mathcal{N}_{x_i}$ implies $\kappa(x_i, y_i) \leq -1 + 2(2 - (\delta_{x_i} + \delta_{y_i}))$. Combining everything using $\delta_{x_i} = \delta_{y_i} = 1$,

$$\mathbb{P}\Big[\kappa(x_i, y_i) \leq -1\Big] \geq \mathbb{P}\Big[\alpha_{x_i} = 1, \alpha_{y_i} = 1\Big]$$

and taking the limit,

$$\lim_{i \to \infty} \mathbb{P}\Big[\kappa(x_i, y_i) \leq -1\Big] = 1. \tag{14}$$

Since we have a finite number of edges and thus a finite number of shortcut edges, we can apply a union bound to say that Equation (14) holds for all edges as $i \to \infty$.

$\square$

Thus we have a result indicating the convergence of the ORC of shortcut edges to at most $-1$ in the limit of vanishing noise. To be confident that ORC-MANL is truly effective, we would like to be able to say that under some conditions the ORC of *non*-shortcut edges tends to be less negative, or even positive. This motivates Theorem 3.2.

> **Theorem 3.2** (Ollivier-Ricci Curvature of Non-Shortcut Edges). *Let $k$ be a positive integer. With high probability as the number of points $n \to \infty$, every point has at least $k$ neighbors that it is connected to by non-shortcut edges with ORC $+1$.*

This theorem states that, as the number of points increases, every point has at least $k$ non-shortcut incident edges with ORC arbitrarily close to $+1$ with high probability. We will outline a high-level overview of the argument to prove this theorem before delving into the formal proof itself.

First, one should observe that the probability that any point has at least some finite number $k$ of neighbors (independent of $n$) within a ball of radius $\delta$ is increasing in $n$. An exact bound for this probability results directly from Proposition 4. One can then construct a sequence $n_i \to \infty$ and $\delta_i \to 0$ such that as $i$ goes to infinity, each point has at least $k$ neighbors within distance $\delta_i$ with high probability.

From here, we want to consider the behavior of the ORC of edges connecting points at some distance $\delta_i$ as $i$ tends to infinity (and thus as $\delta_i$ tends to 0). Proposition 5 and Lemma A.2 in combination allow us to show that as the distance between two points tends to 0, the ORC of the edge connecting them tends to $+1$ with high probability. From all of this it follows that as $n$ tends to infinity, each point has many arbitrarily close (and thus non-shortcutting) neighbors connected by an edge with ORC $+1$ with probability approaching 1. This wraps up the proof sketch.

*Proof of Theorem 3.2.* First we will invoke Proposition 4 to say that for all $x \in \text{Tub}_\tau \mathcal{M}$,

$$\mathbb{P}_{z \sim \rho}\Big[\|z - x\|_2 \le \delta\Big] \ge \frac{\delta^D \text{Vol}\big(B_1^D(0)\big)}{2Z} e^{-\frac{\tau^2}{2\sigma^2}}$$
$$= C\delta^D$$

This result indicates that for any point $x$ in the tubular neighborhood, there is always a nonzero probability that there exists a neighbor at distance at most $\delta$, for $\delta > 0$. Now choose sequences $\{n_i\}_{i=1}^\infty$ and $\{\delta_i\}_{i=1}^\infty$ where $n_i = i$ and $\delta_i = (Ci)^{-1/(2D)}$. Observe that $n_i \to \infty$, $\delta_i \to 0$ and $n_i C \delta_i^D \to \infty$.

Now we would like to evaluate the probability of the existence of at least $k$ neighbors within radius $\delta_i$ for $n_i$ i.i.d. samples from $\rho$. We will use $N_{\delta_i}$ to denote the random variable representing number of neighbors within $\delta_i$ to some point $x$ given $n_i$ i.i.d. draws from $\rho$. Observe that $N_{\delta_i}$ is distributed according to $\text{Bin}(n_i, C\delta_i^D)$. Thus, we can apply a Chernoff bound to say, for $0 < \gamma < 1$

$$\mathbb{P}[N_{\delta_i} \le (1-\gamma)n_i C\delta_i^D] \le \exp\left(-\frac{\gamma^2 n_i C\delta_i^D}{2}\right)$$

and thus,

$$\mathbb{P}[N_{\delta_i} > (1-\gamma)n_i C\delta_i^D] > 1 - \exp\left(-\frac{\gamma^2 n_i C\delta_i^D}{2}\right)$$

(Tsun, 2020). Replacing $\gamma$ with $1 - k/(n_i C\delta_i^D)$ we obtain

$$\mathbb{P}[N_{\delta_i} > k] > 1 - \exp\left(-\frac{n_i C\delta_i^D (1 - k/(n_i C\delta_i^D))^2}{2}\right) \tag{15}$$

and taking the limit as $i \to \infty$,

$$\lim_{i \to \infty} \mathbb{P}[N_{\delta_i} > k] = 1 \tag{16}$$

since $n_i C\delta_i^D \to \infty$.

We have established that if the radius $\delta_i$ shrinks sufficiently slowly, the probability that any point has more than $k$ neighbors within this decreasing radius approaches 1 as $n_i$ tends to infinity. Now we would like to use this result to make a statement about the ORC of edges connecting a point and its neighbors within distance $\delta_i$. To do so, consider a point $x$ and distances $\{\delta_i\}_{i=1}^\infty$ such that $\delta_i \to 0$. We can consider $x$ as a static point in a sequence of nested point clouds, where each point cloud $\mathcal{X}_i$ is a union of the previous point cloud $\mathcal{X}_{i-1}$ and one more point sampled from $\rho$. Also for each $i$ consider the set of neighbors $\{y_i^j \mid y_i^j \in \mathcal{X}_i, y_i^j \ne x\}_j$ such that $\|x - y_i^j\|_2 \le \delta_i$; note that the earlier results indicate that $|\{y_i^j\}_j| > k$ with probability $\mathbb{P}[N_{\delta_i} > k]$, as described by 15.

Let $\mathcal{N}_{x,y_i^j} = |\mathcal{N}(x) \cup \mathcal{N}(y_i^j) \setminus \{x, y_i^j\}|$, which denotes the set of all neighbors of $x$ and $y_i^j$ (excluding themselves). Now define $S_{x,y_i^j}$ to be the neighbors of $x$ and $y_i^j$ that lay in $B_\epsilon(x) \cap B_\epsilon(y_i^j)$. Finally, let $p_{x,y_i^j} = \mathbb{P}_{a \sim \rho}\big[a \in B_\epsilon(x) \cap B_\epsilon(y_i^j) \,\big|\, a \in B_\epsilon(x) \cup B_\epsilon(y_i^j)\big]$. Observe that $|S_{x,y_i^j}|$ is a random variable distributed according to $\text{Bin}(\mathcal{N}_{x,y_i^j}, p_{x,y_i^j})$ when $\mathcal{N}_{x,y_i^j}$ is fixed. Also note that $\mathcal{N}_{x,y_i^j}$ itself is a random variable with a binomial distribution with $n_i = i$ trials. Observe that it exhibits success probability $p \le 2C''\epsilon^D := C'\epsilon^D$, which follows from a slight modification to the result of Proposition 4. Thus, $\mathbb{E}[\mathcal{N}_{x,y_i^j}] \ge iC'\epsilon^D$. Armed with this knowledge, define two more sequences $k_{i*} = \sqrt{i}$ and $\gamma_i = i^{-1/8}$. We will return to these definitions shortly. First, we will apply a Chernoff bound to say

$$\mathbb{P}\Big[|S_{x,y_i^j}| \le (1-\gamma_i)\mathcal{N}_{x,y_i^j} p_{x,y_i^j} \,\Big|\, \mathcal{N}_{x,y_i^j} = k\Big] \le \exp\left(-\frac{\gamma_i^2 k p_{x,y_i^j}}{2}\right)$$

and thus,

$$\mathbb{P}\Big[|S_{x,y_i^j}| \le (1-\gamma_i)\mathcal{N}_{x,y_i^j} p_{x,y_i^j}\Big] \le \sum_{k=0}^{i} \exp\left(-\frac{\gamma_i^2 k p_{x,y_i^j}}{2}\right) \mathbb{P}\Big[\mathcal{N}_{x,y_i^j} = k\Big] \tag{17}$$

$$= \sum_{k \le k_{i*}} \underbrace{\exp\left(-\frac{\gamma_i^2 k p_{x,y_i^j}}{2}\right)}_{\le 1} \mathbb{P}\Big[\mathcal{N}_{x,y_i^j} = k\Big]$$

$$+ \sum_{k > k_{i*}} \underbrace{\exp\left(-\frac{\gamma_i^2 k p_{x,y_i^j}}{2}\right)}_{\le \exp\left(-(\gamma_i^2 k_{i*} p_{x,y_i^j})/2\right)} \underbrace{\mathbb{P}\Big[\mathcal{N}_{x,y_i^j} = k\Big]}_{\le 1}$$

$$\tag{18}$$

$$\le \mathbb{P}\Big[\mathcal{N}_{x,y_i^j} \le k_{i*}\Big] + \exp\left(-\frac{\gamma_i^2 k_{i*} p_{x,y_i^j}}{2}\right). \tag{19}$$

We can apply one more Chernoff bound to bound the first term, resulting in

$$\mathbb{P}\Big[|S_{x,y_i^j}| \le (1-\gamma_i)\mathcal{N}_{x,y_i^j} p_{x,y_i^j}\Big] \le \exp\left(-\frac{\mathbb{E}[\mathcal{N}_{x,y_i^j}](1 - k_{i*}/\mathbb{E}[\mathcal{N}_{x,y_i^j}])^2}{2}\right)$$

$$+ \exp\left(-\frac{\gamma_i^2 k_{i*} p_{x,y_i^j}}{2}\right). \tag{20}$$

We can apply Proposition 5 to say as $i \to \infty$, $p_{x,y_i^j} \to 1$. Now since $\gamma_i \to 0$, $\mathbb{E}[\mathcal{N}_{x,y_i^j}] \to \infty$, $k_{i*}/\mathbb{E}[\mathcal{N}_{x,y_i^j}] \to 0$ and $\gamma_i^2 k_{i*} \to \infty$, we have

$$\lim_{i \to \infty} \mathbb{P}\Big[|S_{x,y_i^j}| < \mathcal{N}_{x,y_i^j}\Big] = 0 \tag{21}$$

and thus, since $|S_{x,y_i^j}| \le \mathcal{N}_{x,y_i^j}$

$$\lim_{i \to \infty} \mathbb{P}\Big[|S_{x,y_i^j}| = \mathcal{N}_{x,y_i^j}\Big] = 1.$$

Now we can invoke Lemma A.2 and Proposition 2 to say that $|S_{x,y_i^j}| = \mathcal{N}_{x,y_i^j}$ implies $\kappa(x, y_i^j) = 1$. Putting it all together, we see that

$$\lim_{i \to \infty} \mathbb{P}\Big[\kappa(x, y_i^j) = 1\Big] = 1. \tag{22}$$

Now let $p_i^j = \mathbb{P}[\kappa(x, y_i^j) = 1]$. Now we want to use this to bound the probability that, for a fixed $i$, a subset of size $k$ of all $N_{\delta_i}$ neighbors have ORC of 1. To do so, let's denote $\alpha_i^j = \mathbb{1}[\kappa(x, y_i^j) = 1]$. Instead of considering all combinations of subsets of size $k$, we will just consider a single one to create a lower bound. Namely, we will cherry pick the first $k$ indices (unless $N_{\delta_i} < k$). Now observe that we can apply a union bound to obtain

$$\mathbb{P}\Big[\kappa(x, y_i^j) = 1 \,\forall\, j \in [\min\{k, N_{\delta_i}\}]\Big]$$

$$\ge 1 - \sum_{\left\{\{\alpha_i^j\}_{j=1}^i \,\big|\, \sum_j \alpha_i^j < \min\{k, N_{\delta_i}\}\right\}} \mathbb{P}\Big[\alpha_i^1, \dots, \alpha_i^{\min\{k, N_{\delta_i}\}}\Big] \tag{23}$$

where the second term on the right-hand side sums over all settings of $\{\alpha_i^j\}$ such that there exists at least one $\alpha_i^j$ taking on the value 0. Now we can use the fact that $\mathbb{P}[X_1, \dots, X_n] \le \min\{\mathbb{P}[X_1], \dots, \mathbb{P}[X_n]\}$ to say

$$\mathbb{P}\Big[\kappa(x, y_i^j) = 1 \,\forall\, j \in [\min\{k, N_{\delta_i}\}]\Big] \ge$$

$$1 - \sum_{\left\{\{\alpha_i^j\}_{j=1}^i \,\big|\, \sum_j \alpha_i^j < \min\{k, N_{\delta_i}\}\right\}} \min_{j \in [\min\{k, N_{\delta_i}\}]} \mathbb{P}\Big[\alpha_i^j\Big]. \tag{24}$$

Now we will take the limit as $i \to \infty$ resulting in

$$\lim_{i \to \infty} \mathbb{P}\Big[\kappa(x, y_i^j) = 1 \, \forall \, j \in [\min\{k, N_{\delta_i}\}]\Big]$$

$$\geq 1 - \lim_{i \to \infty} \sum_{\big\{\{\alpha_i^j\}_{j=1}^i \,\big|\, \sum_j \alpha_i^j < \min\{k, N_{\delta_i}\}\big\}} \min_{j \in [\min\{k, N_{\delta_i}\}]} \mathbb{P}\big[\alpha_i^j\big]. \quad (25)$$

Since $\lim_{i \to \infty} \mathbb{P}[N_{\delta_i} > k] = 1$ and since the summation is over a finite number of terms,

$$\lim_{i \to \infty} \mathbb{P}\Big[\kappa(x, y_i^j) = 1 \, \forall \, j \in [k]\Big] \geq 1 - \sum_{\big\{\{\alpha_i^j\}_{j=1}^i \,\big|\, \sum_j \alpha_i^j < k\big\}} \lim_{i \to \infty} \min_{j \in [\min\{k, N_{\delta_i}\}]} \mathbb{P}\big[\alpha_i^j\big]. \quad (26)$$

Note $\mathbb{P}[\alpha_i^j = 1] = p_i^j$ and $\mathbb{P}[\alpha_i^j = 0] = 1 - p_i^j$. Observe that in the limit as $i$ goes to infinity, these two expressions converge to 1 and 0 respectively due to eq. (22). Thus the limit can be pulled inside of the $\min$ function. Since we know that for each element of the sum there necessarily exists an $\alpha_i^j = 0$, one of the arguments of the $\min$ function must be 0. Thus,

$$\lim_{i \to \infty} \mathbb{P}\Big[\kappa(x, y_i^j) = 1 \, \forall \, j \in [k]\Big] = 1. \quad (27)$$

Note that for sufficiently large $i$, $x$ and $y_i^j$ cannot be shortcut edges as $\|x - y_i^j\|_2 < s_0(\mathcal{M})$ which, in turn, implies $d_{\mathcal{M}}(\text{proj}_{\mathcal{M}} x, y_i^j) \leq \pi r_0(\mathcal{M})$. Finally, we can succinctly state the conclusion by saying that if $n$ grows sufficiently fast, every point has at least $k$ non-shortcut neighbors such that the ORC is 1 with high probability. This completes the proof.

$\square$

Together, Theorems 3.1 and 3.2 establish that in the limit of vanishing noise and infinite points shortcut edges can be detected with complete accuracy. In practice though, we may see too much noise or too few samples to rely on ORC alone. Therefore it is of interest to us to add another validation step. We do so by looking at shortest path graph distances in a modified version of the original graph, one which has all especially negative curvature edges removed; we call this graph the 'thresholded' graph. Theorem 3.3 establishes that graph shortest paths between endpoints of what were shortcut edges in the original graph necessarily have a large graph distance in the thresholded graph. Figure 13 provides an intuitive visualization of this phenomenon.

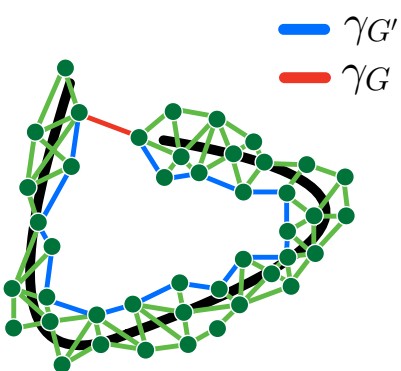

Figure 13: Visualization of graph shortest paths between the endpoints of a shortcut edge $(x, y)$. The path $\gamma_G$ denotes the path through $G$ (which is trivially a single hop), while $\gamma_{G'}$ denotes the path through the ORC thresholded graph, $G'$.

**Theorem 3.3** (Filtered Graph Distance). *Suppose that $\mathcal{X}_i$ is a point cloud sampled from $\rho$ with parameters $\sigma_i$ and $\tau_i$ and $G_i = (V_i, E_i)$ is its nearest neighbor graph. Also suppose that $\mathcal{M}$ satisfies the conditions above and $\sigma_i \to 0^+$ and $\tau_i \to 0^+$ as $i \to \infty$. Define the subgraph $G_i' = (V_i, E_i')$ where*

$$E_i' = \Big\{(x_i, y_i) \in E_i \,\Big|\, \kappa(x_i, y_i) > -1\Big\}.$$

*Then as $i \to \infty$ we have*

$$d_{G'}(x, y) > \beta \frac{\pi(\pi + 1)(1 - \lambda)}{2\sqrt{24\lambda}} \epsilon$$

*for all shortcut edges in $G_i$ with probability approaching 1, where $\beta \in [0, 1]$ (eq. (34)) is a random variable whose distribution is dependent on $\mathcal{M}$ and $\tau_i$.*

The proof of the theorem adopts the following strategy. Geodesic distances through the manifold $\mathcal{M}$ can be bounded as a function of geodesic distances through the tubular neighborhood $\text{Tub}_\tau \mathcal{M}$ using Proposition 3. We can then pull from Bernstein et al. (2001), which relates graph geodesics to manifold geodesics, to bound graph distances through $G'$ as a function of manifold distances $\mathcal{M}$. The theorem can be concisely stated as showing that in the limit of vanishing noise, graph distances in $G'$ between endpoints of (formerly) shortcut edges are necessarily very large.

*Proof of Theorem 3.3.* Again suppose $\{\mathcal{X}_i\}_{i=1}^\infty$ is a sequence of point clouds of size $n$ sampled from $\rho$ defined in eq. (1) with parameters $\sigma_i$ and $\tau_i$ with $\sigma_i \to 0^+$ and $\tau_i \to 0^+$ as $i \to \infty$. Also suppose $G_i = (V_i, E_i)$ is the nearest neighbor graph built with parameter $\epsilon$ from the point cloud $\mathcal{X}_i$. Consider the $i$-th pointcloud and nearest neighbor graph in this sequence, and let $x_i$ and $y_i$ be any two points connected by a shortcut edge (if they exist). Since $(x_i, y_i)$ is a shortcut edge in $G_i$ we have,

$$d_{\mathcal{M}}(\text{proj}_{\mathcal{M}} x_i, \text{proj}_{\mathcal{M}} y_i) > (\pi + 1)r_0(\mathcal{M}).$$

From Proposition 3, we know this implies

$$d_{\text{Tub}}(x_i, y_i) \geq \frac{r_0(\mathcal{M}) - \tau_i}{r_0(\mathcal{M})}(\pi + 1)r_0(\mathcal{M}) \tag{28}$$

$$= (r_0(\mathcal{M}) - \tau_i)(\pi + 1). \tag{29}$$

Now consider the ORC thresholded graph $G_i' = (V_i, E_i')$, with edges

$$E_i' = \left\{ (v, v') \in E_i \,\middle|\, \kappa(v, v') > -1 \right\}.$$

We can apply Theorem 3.1 to say that with high probability $G_i'$ has no shortcut edges. With that in mind, select a length minimizing path $a_0^i, a_1^i, \cdots, a_p^i$ connecting $x_i$ to $y_i$ through the graph $G_i'$. With high probability we have,

$$d_{\mathcal{M}}(\text{proj}_{\mathcal{M}} a_i^i, \text{proj}_{\mathcal{M}} a_{i+1}^i) \leq (\pi + 1)r_0(\mathcal{M}).$$

It follows that $d_{\text{Tub}}(a_i^i, a_{i+1}^i) \leq (\pi + 1)r_0(\mathcal{M}) + 2\tau_i$, as $\mathcal{M} \subset \text{Tub}_{\tau_i}\mathcal{M}$; if this wasn't true, the shortest path through the tubular neighborhood would instead take the path traversed by $\mathcal{M}$ through $\text{proj}_{\mathcal{M}} a_i^i, \text{proj}_{\mathcal{M}} a_{i+1}^i$. Now we can say,

$$d_{\text{Tub}}(a_i^i, a_{i+1}^i) \leq (\pi + 1)r_0(\mathcal{M}) + 2\tau_i.$$

Consider the $p'$ of the $p - 1$ hops along this path where

$$d_{\text{Tub}}(a_i^i, a_{i+1}^i) \leq \pi(r_0(\mathcal{M}) - \tau_i).$$

Note that from Proposition 1 we know that the minimum radius of curvature of geodesics in the tubular neighborhood is $r_0(\mathcal{M}) - \tau_i$. Now we can invoke the first-order weakening of Lemma 3 from Bernstein et al. (2001) with $r_0 = r_0(\mathcal{M}) - \tau_i$ to say

$$\frac{2}{\pi} d_{\text{Tub}}(a_i^i, a_{i+1}^i) \leq \|a_i^i - a_{i+1}^i\|_2$$

$$\leq \frac{2}{\pi}(r_0(\mathcal{M}) - \tau_i)\sqrt{24\lambda}$$

where the second line follows from assumption 2. Solving for $\lambda$ we get

$$\lambda \geq \frac{1}{24}\left(\frac{d_{\text{Tub}}(a_i^i, a_{i+1}^i)}{r_0(\mathcal{M}) - \tau_i}\right)^2$$

and therefore

$$1 - \lambda \leq 1 - \frac{1}{24}\left(\frac{d_{\text{Tub}}(a_i^i, a_{i+1}^i)}{r_0(\mathcal{M}) - \tau_i}\right)^2.$$

Finally we can apply the weakened form of the same Lemma from Bernstein et al. (2001) to say

$$(1 - \lambda)d_{\text{Tub}}(a_i^i, a_{i+1}^i) \leq \left(1 - \frac{1}{24}\left(\frac{d_{\text{Tub}}(a_i^i, a_{i+1}^i)}{r_0(\mathcal{M}) - \tau_i}\right)^2\right)d_{\text{Tub}}(a_i^i, a_{i+1}^i)$$

$$\leq \|a_i^i - a_{i+1}^i\|_2.$$

Now consider the other $(p - 1) - p'$ hops, where we know

$$d_{\text{Tub}}(a_i^i, a_{i+1}^i) > \pi(r_0(\mathcal{M}) - \tau_i).$$

We can apply the definition of minimum branch separation to say that $\|a_i^i - a_{i+1}^i\|_2 \geq s_0(\text{Tub}_{\tau_i}\mathcal{M})$. In general deriving an expression for the right-hand side in terms of $r_0(\mathcal{M})$ and $s_0(\mathcal{M})$ is impossible without more information about the embedding of $\mathcal{M}$. Thus, we will proceed without breaking down $s_0(\text{Tub}_{\tau_i}\mathcal{M})$. We have,

$$\|a_i^i - a_{i+1}^i\|_2 \geq s_0(\text{Tub}_{\tau_i}\mathcal{M}) \tag{30}$$

$$\geq \frac{s_0(\text{Tub}_{\tau_i}\mathcal{M})}{(\pi + 1)r_0(\mathcal{M}) + 2\tau_i}d_{\text{Tub}_i}(a_i^i, a_{i+1}^i). \tag{31}$$

Now we can combine the bounds on $\|a_i^i - a_{i+1}^i\|_2$ to say

$$d_{G_i'}(x_i, y_i) > (1 - \lambda)\sum_S d_{\text{Tub}}(a_i^i, a_{i+1}^i) + \sum_L \frac{s_0(\text{Tub}_{\tau_i}\mathcal{M})}{(\pi + 1)r_0(\mathcal{M}) + 2\tau_i}d_{\text{Tub}_i}(a_i^i, a_{i+1}^i) \tag{32}$$

$$> (1 - \lambda)\sum_S d_{\text{Tub}}(a_i^i, a_{i+1}^i) \tag{33}$$

where $S$ is the set of all hops with tubular distance at most $\pi(r_0(\mathcal{M}) - \tau_i)$, and $L$ is the set of all remaining hops. Defining

$$\beta = \frac{\sum_S d_{\text{Tub}}(a_i^i, a_{i+1}^i)}{d_{\text{Tub}}(x_i, y_i)} \tag{34}$$

allows us to say

$$d_{G_i'}(x_i, y_i) > \beta(1 - \lambda)d_{\text{Tub}}(x_i, y_i)$$

$$\geq \beta(1 - \lambda)(r_0(\mathcal{M}) - \tau_i)(\pi + 1)$$

and seeing as we have stipulated that $\epsilon < \frac{2}{\pi}(r_0(\mathcal{M}) - \tau)\sqrt{24\lambda}$ in assumption 2, we can say

$$d_{G_i'}(x_i, y_i) > \beta\frac{\pi(\pi + 1)(1 - \lambda)}{2\sqrt{24da}}\epsilon \tag{35}$$

with high probability. $\qquad\square$

### A.4.2 LEMMATA

In this section, we will prove the lemmas used by Theorems 3.1, 3.2 and 3.3. Lemmas A.1 and A.2 simply bound (from above and below, respectively) the ORC of an edge based on the locations of each endpoints neighbors. Lemma A.1 considers the scenario where some subset of the neighborhoods of $x$ and $y$ are positioned in the sets $U_x$ and $U_y$, respectively (where $U_x$ and $U_y$ are defined according to Figure 12). The large distance between $U_x$ and $y$ (and vice versa) allows one to lower bound the Wasserstein distance between $\mu_x$ and $\mu_y$, and thus upper bound the ORC of $(x, y)$.

**Lemma A.1.** *Let $(x, y)$ be an edge in a nearest neighbor graph built from potentially noisy samples of $\mathcal{M}$, suppose $U_x$ and $U_y$ are defined as in 12. Define $S_x = \{a \,|\, a \in \mathcal{N}(x) \cap U_x\}$ and $S_y = \{b \,|\, b \in \mathcal{N}(y) \cap U_y\}$, and suppose that*

$$\delta_x = \frac{|S_x|}{|\mathcal{N}(x) \setminus \{y\}|}, \qquad \delta_y = \frac{|S_y|}{|\mathcal{N}(y) \setminus \{x\}|}.$$

*Then,*

$$\kappa(x,y) \leq -1 + 2\Big(2 - (\delta_x + \delta_y)\Big).$$

*Proof.* We are interested in bounding the ORC of the edge $(x,y)$ from above. Recall that we have defined ORC to use the *unweighted* graph metric in Section 2.3. To bound the ORC we need to bound the Wasserstein distance between $\mu_x$ and $\mu_y$, where $\mu_x$ is a uniform measure on the set $\mathcal{N}(x) \setminus \{y\} = \{v \in V \,|\, d_G(x,v) \leq \epsilon, v \neq y, v \neq x\}$ and $\mu_y$ is a uniform measure on the set $\mathcal{N}(y) \setminus \{x\} = \{v \in V \,|\, d_G(y,v) \leq \epsilon, v \neq x, v \neq y\}$.

Let's define $\hat{\mu}_x$ and $\hat{\mu}_y$ as uniform probability measures over $S_x$ and $S_y$ respectively. We can bound the Wasserstein distance between $\mu_x$ and $\mu_y$ as

$$W(\mu_x, \mu_y) \geq W(\hat{\mu}_x, \hat{\mu}_y) - W(\hat{\mu}_x, \mu_x) - W(\hat{\mu}_y, \mu_y).$$

Since $\|a - b\|_2 > \epsilon$ for all $a \in S_x$ and for all $b \in S_y$, we know that $d_G(a,b) \geq 2$ for all $a \in S_x$ and for all $b \in S_y$. Thus, a lower bound on the first term follows,

$$W(\hat{\mu}_x, \hat{\mu}_y) \geq 2.$$

Now we would like to bound $W(\hat{\mu}_x, \mu_x)$ from above. There is $1/|\mathcal{N}(x) \setminus \{y\}|$ mass on each node $a \in \text{supp}(\mu_x)$, while there is $1/\delta_x|\mathcal{N}(x) \setminus \{y\}|$ mass on each $a' \in \text{supp}(\hat{\mu}_x)$. We can define a feasible transport plan that transports all excess mass on $a' \in \text{supp}(\hat{\mu}_x)$ to the nodes $\text{supp}(\mu_x) \setminus \text{supp}(\hat{\mu}_x)$. Since the Wasserstein distance minimizes over all possible transport plans, the Wasserstein cost for this transport plan will upper bound the true distance.

The excess mass on any $a' \in \text{supp}(\hat{\mu}_x)$ is exactly

$$\frac{1}{\delta_x|\mathcal{N}(x) \setminus \{y\}|} - \frac{1}{|\mathcal{N}(x) \setminus \{y\}|}$$

which means the total mass that needs to be transported is

$$\delta_x|\mathcal{N}(x) \setminus \{y\}|\left(\frac{1}{\delta_x|\mathcal{N}(x) \setminus \{y\}|} - \frac{1}{|\mathcal{N}(x) \setminus \{y\}|}\right) = 1 - \delta_x$$

We also know that from any $a' \in \text{supp}(\hat{\mu}_x)$ to any $a \in \text{supp}(\mu_x)$ there exists a length 2 path through the node $x$. Therefore, $d_G(a, a') \leq 2$. We can then conclude

$$W(\hat{\mu}_x, \mu_x) \leq 2\big(1 - \delta_x\big).$$

With the same argument, the following bound can also be derived,

$$W(\hat{\mu}_y, \mu_y) \leq 2\big(1 - \delta_y\big).$$

Putting it all together,

$$W(\mu_x, \mu_y) \geq 2 - 2\big(2 - (\delta_x + \delta_y)\big).$$

Solving for the ORC,

$$\kappa(x,y) \leq 1 - \frac{2 - 2\big(2 - (\delta_x + \delta_y)\big)}{1}$$
$$= -1 + 2\Big(2 - (\delta_x + \delta_y)\Big).$$

$\square$

Now we can adopt a similar argument to tackle Lemma A.2. Now we are interested in *lower* bounding the ORC based on the neighborhood structure. In this scenario, we replace the sets $U_x$ and $U_y$ with a $B_\epsilon(x) \cap B_\epsilon(y)$. Each neighbor of an endpoint $(x)$ that is positioned in this set is necessarily a neighbor of the other $(y)$. Therefore if many nodes are present in this set, then many of the neighbors of $x$ and $y$ are shared. An upper bound on the Wasserstein distance between $\mu_x$ and $\mu_y$ follows, which implies a lower bound on the ORC of the edge $(x,y)$.

**Lemma A.2.** *Let $(x, y)$ be an edge in a nearest neighbor graph built from potentially noisy samples of $\mathcal{M}$, and define $S_{x,y} = \{a \mid a \in \mathcal{N}(x) \cap \mathcal{N}(x)\}$. Suppose that*

$$\delta_x = \frac{|S_{x,y}|}{|\mathcal{N}(x) \setminus \{y\}|}, \qquad \delta_y = \frac{|S_{x,y}|}{|\mathcal{N}(y) \setminus \{x\}|}.$$

*Then,*

$$\kappa(x, y) \geq 1 - 2\Big(2 - (\delta_x + \delta_y)\Big).$$

*Proof.* We will adopt a very similar argument as that of the proof for Lemma A.1. We are interested in bounding the ORC of the edge $(x, y)$ from below. Doing so involves bounding the Wasserstein distance between $\mu_x$ and $\mu_y$, where $\mu_x$ is a uniform measure on the set $\mathcal{N}(x) \setminus \{y\} = \{v \in V \mid d_G(x, v) \leq \epsilon, v \neq y, v \neq x\}$ and $\mu_y$ is a uniform measure on the set $\mathcal{N}(y) \setminus \{x\} = \{v \in V \mid d_G(y, v) \leq \epsilon, v \neq x, v \neq y\}$. Again recall that we have defined ORC to use the *unweighted* graph metric in Section 2.3.

Define $\hat{\mu}_{x,y}$ to be the uniform probability measure over $S_{x,y}$. We can bound the Wasserstein distance between $\mu_x$ and $\mu_y$ as

$$W(\mu_x, \mu_y) \leq W(\mu_x, \hat{\mu}_{x,y}) + W(\hat{\mu}_{x,y}, \mu_y).$$

Now we would like to bound $W(\mu_x, \hat{\mu}_{x,y})$ from above. There is $1/|\mathcal{N}(x) \setminus \{y\}|$ mass on each node $a \in \text{supp}(\mu_x)$, while there is $1/\delta_x|\mathcal{N}(x) \setminus \{y\}|$ mass on each $a' \in \text{supp}(\hat{\mu}_{x,y})$. We can define a feasible transport plan that transports all excess mass on $a' \in \text{supp}(\hat{\mu}_{x,y})$ to the nodes $\text{supp}(\mu_x) \setminus \text{supp}(\hat{\mu}_{x,y})$. Since the Wasserstein distance minimizes over all possible transport plans, the Wasserstein cost for this transport plan will upper bound $W(\mu_x, \hat{\mu}_{x,y})$.

The excess mass on any $a' \in \text{supp}(\hat{\mu}_x)$ is exactly

$$\frac{1}{\delta_x |\mathcal{N}(x) \setminus \{y\}|} - \frac{1}{|\mathcal{N}(x) \setminus \{y\}|}$$

which means the total mass that needs to be transported is

$$\delta_x |\mathcal{N}(x) \setminus \{y\}| \left( \frac{1}{\delta_x |\mathcal{N}(x) \setminus \{y\}|} - \frac{1}{|\mathcal{N}(x) \setminus \{y\}|} \right) = 1 - \delta_x$$

We also know that from any $a' \in \text{supp}(\hat{\mu}_{x,y})$ to any $a \in \text{supp}(\mu_x)$ there exists a length 2 path through the node $x$. Therefore, $d_G(a, b) \leq 2$. We can then conclude

$$W(\mu_x, \hat{\mu}_{x,y}) \leq 2\big(1 - \delta_x\big).$$

With the same argument, the following bound can also be derived,

$$W(\hat{\mu}_{x,y}, \mu_y) \leq 2\big(1 - \delta_y\big).$$

Putting it all together,

$$W(\mu_x, \mu_y) \leq 2\big(2 - (\delta_x + \delta_y)\big).$$

Solving for the ORC,

$$\kappa(x, y) \geq 1 - \frac{2\big(2 - (\delta_x + \delta_y)\big)}{1}$$
$$= 1 - 2\big(2 - (\delta_x + \delta_y)\big).$$

$\square$

Moving on, Lemma A.3 and Lemma A.4 constitute a majority of the theoretical work of this paper. Together, they show that most of the measure in the $\epsilon$-ball of a point $x$ connected by a shortcut edge

to $y$ tends to concentrate far from $y$. For context, Theorem 3.1 uses this fact in conjunction with Lemma A.1 to show that the ORC of shortcut edges is necessarily very negative.

To set the stage, Lemma A.3 establishes two things. First, and most importantly, it shows that a shortcut edge is necessarily oriented close to a normal direction $v$ to the manifold at its endpoints. This concept is rather intuitive; if shortcut edges were closer to tangential directions instead, then these edges would intersect with the manifold prematurely, no longer bridging distant neighborhoods and contradicting their status as shortcuts. The second property the lemma establishes is the fact that the residual of the projection (onto $\mathcal{M}$) of an endpoint $x$ of a shortcut edge must be within $\pi/2$ radians of the aforementioned normal direction $v$. Put simply, this means that the endpoints of a shortcut edge are necessarily displaced off the manifold in the direction of or orthogonal to the normal direction that the edge itself spans. This simply arises from the assumptions about the size of $\epsilon$ relative to the manifold embedding parameters $r_0(\mathcal{M})$ and $s_0(\mathcal{M})$. We show this second property holds as it simplifies some of the downstream calculations.

**Lemma A.3.** *If an edge $(x, y)$ in a nearest neighbor graph built from potentially noisy samples of $\mathcal{M}$ is a shortcut edge then*

1. *there exists a unit vector $v \in N_{\mathrm{proj}_{\mathcal{M}} x}\mathcal{M}$ (the normal space of $\mathcal{M}$ at $\mathrm{proj}_{\mathcal{M}} x$) such that the angle $\phi$ between $v$ and $(y - x)$ is smaller than*

$$\arccos\left(\frac{(r_0(\mathcal{M}) - \tau)(s_0(\mathcal{M}) - \tau) - 1.27 r_0(\mathcal{M})\tau}{r_0(\mathcal{M})\sqrt{(s_0(\mathcal{M}) - \tau)^2 - \tau^2}}\right) < \pi/2. \quad (36)$$

2. *the angle $\theta$ between $v$ and $(x - \mathrm{proj}_{\mathcal{M}} x)$ is at most $\pi/2$.*

*Proof.* Observe that the vector $(y - x)$ can be written as $\alpha v_T + \beta v$, where $v_T$ is a unit vector $\in T_{\mathrm{proj}_{\mathcal{M}} x}\mathcal{M}$, while $v$ is a unit vector $\in N_{\mathrm{proj}_x \mathcal{M}}\mathcal{M}$. This vector $v$ is in fact the unit vector in $N_{\mathrm{proj}_x \mathcal{M}}\mathcal{M}$ that minimizes the angle $\phi$ to $(y - x)$. Motivated by this, Figure 14 visualizes the 3 dimensional subspace spanned by the vectors $v_T, v$ and $(x - \mathrm{proj}_{\mathcal{M}} x)$. This visualization lays the conceptual foundation that the rest of the proof rests on.

First, we will show (2) holds, as it is helpful in showing (1). To do so, we will show that the geometry of shortcutting edges implies a bound on $\|x - y\|_2$ as a function of $\theta$. We will then show that this bound violates the assumptions on the size of $\epsilon$ for $\theta \geq \pi/2$.

Since $(x, y)$ is a shortcut edge, $d_{\mathcal{M}}(\mathrm{proj}_{\mathcal{M}} x, \mathrm{proj}_{\mathcal{M}} y) > (\pi + 1)r_0(\mathcal{M}) > \pi r_0(\mathcal{M})$. By the definition of minimum branch separation, we know then $\|\mathrm{proj}_{\mathcal{M}} x - \mathrm{proj}_{\mathcal{M}} y\|_2 \geq s_0(\mathcal{M})$. Now consider the triangle defined by the endpoints $\mathrm{proj}_{\mathcal{M}} x, \mathrm{proj}_{\mathcal{M}} y$ and $y$. We can apply the triangle inequality to say

$$\|\mathrm{proj}_{\mathcal{M}} x - y\|_2 \geq \|\mathrm{proj}_{\mathcal{M}} x - \mathrm{proj}_{\mathcal{M}} y\|_2 - \|\mathrm{proj}_{\mathcal{M}} y - y\|_2$$
$$\geq s_0(\mathcal{M}) - \tau$$

since $\tau \geq \|\mathrm{proj}_{\mathcal{M}} y - y\|_2$. Now consider the triangle defined by the endpoints $y, \mathrm{proj}_{\mathcal{M}} x$ and $y'$, where $y'$ is the orthogonal projection of $y$ onto the subspace spanned by $v$ and $(y - x)$ centered at $\mathrm{proj}_{\mathcal{M}} x$ (shown in Figure 14). Note that

$$\|y' - \mathrm{proj}_{\mathcal{M}} x\|_2 \geq \sqrt{\|\mathrm{proj}_{\mathcal{M}} x - y\|_2^2 - \|x - \mathrm{proj}_{\mathcal{M}} x\|_2^2 \sin^2(\theta)}$$

$$\geq \sqrt{(s_0(\mathcal{M}) - \tau)^2 - \|x - \mathrm{proj}_{\mathcal{M}} x\|_2^2 \sin^2(\theta)}.$$

Finally, consider the triangle defined by the endpoints $\mathrm{proj}_{\mathcal{M}} x, y'$ and $x'$, where $x'$ is the orthogonal projection of $x$ onto $v$. Observe that $\|x' - \mathrm{proj}_{\mathcal{M}} x\|_2 = \|x - \mathrm{proj}_{\mathcal{M}} x\|_2 \cos(\theta)$ and $\|x' - y'\|_2 = \|x - y\|_2$. The second equivalence stems from the fact that the vectors $v$ and $v_T$ were chosen specifically so that they can be linearly combined to get $(y - x)$; it follows that $x' - y'$ is the same as $x - y$. With that in mind, we can apply the triangle inequality again to say

$$\|x - y\|_2 \geq \sqrt{(s_0(\mathcal{M}) - \tau)^2 - \|x - \mathrm{proj}_{\mathcal{M}} x\|_2^2 \sin^2(\theta)} - \|x - \mathrm{proj}_{\mathcal{M}} x\|_2 \cos(\theta). \quad (37)$$

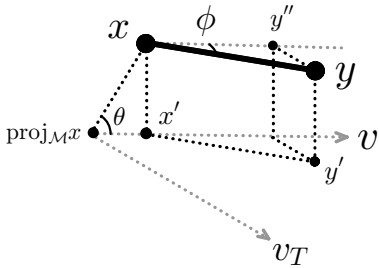

Figure 14: A visualization of the plane spanned by the vectors $(y - x)$ and $v$, and related quantities.

Property (2) follows from the prescribed upper bound, $\|x - y\|_2 \leq \sqrt{(s_0(\mathcal{M}) - \tau)^2 - \tau^2}$ from assumption 2. Thus,

$$\sqrt{(s_0(\mathcal{M}) - \tau)^2 - \tau^2} \geq \sqrt{(s_0(\mathcal{M}) - \tau)^2 - \|x - \mathrm{proj}_{\mathcal{M}} x\|_2^2 \sin^2(\theta)} - \|x - \mathrm{proj}_{\mathcal{M}} x\|_2 \cos(\theta).$$

Solving for $\theta$,

$$(s_0(\mathcal{M}) - \tau)^2 - \tau^2 + 2\|x - \mathrm{proj}_{\mathcal{M}} x\|_2 \cos(\theta) \sqrt{(s_0(\mathcal{M}) - \tau)^2 - \tau^2} + \|x - \mathrm{proj}_{\mathcal{M}} x\|_2^2 \cos^2(\theta)$$
$$\geq (s_0(\mathcal{M}) - \tau)^2 - \|x - \mathrm{proj}_{\mathcal{M}} x\|_2^2 \sin^2(\theta)$$

$$(s_0(\mathcal{M}) - \tau)^2 - \tau^2 + 2\|x - \mathrm{proj}_{\mathcal{M}} x\|_2 \cos(\theta) \sqrt{(s_0(\mathcal{M}) - \tau)^2 - \tau^2}$$
$$\geq (s_0(\mathcal{M}) - \tau)^2 - \|x - \mathrm{proj}_{\mathcal{M}} x\|_2^2.$$

Rearranging a little more,

$$2\|x - \mathrm{proj}_{\mathcal{M}} x\|_2 \cos(\theta) \sqrt{(s_0(\mathcal{M}) - \tau)^2 - \tau^2} \geq \tau^2 - \|x - \mathrm{proj}_{\mathcal{M}} x\|_2^2$$

$$\cos(\theta) \geq \frac{\tau^2 - \|x - \mathrm{proj}_{\mathcal{M}} x\|_2^2}{2\|x - \mathrm{proj}_{\mathcal{M}} x\|_2 \sqrt{(s_0(\mathcal{M}) - \tau)^2 - \tau^2}}$$

thus,

$$\theta \leq \arccos\left(\frac{\tau^2 - \|x - \mathrm{proj}_{\mathcal{M}} x\|_2^2}{2\|x - \mathrm{proj}_{\mathcal{M}} x\|_2 \sqrt{(s_0(\mathcal{M}) - \tau)^2 - \tau^2}}\right)$$
$$\leq \arccos(0)$$
$$= \pi/2$$

Since $\|x - \mathrm{proj}_{\mathcal{M}} x\|_2 \leq \tau$. This concludes the proof of (2).

Now we will show (1). At a high level, we will show that if the angle $\phi$ is *not* smaller than the listed threshold, $y$ must have a small Euclidean distance to a small neighborhood around $x$; for points in this neighborhood, the manifold distance to the projection of $y$ is large. This leads to a contradiction stemming from a violation of the minimum branch separation of $\mathcal{M}$.

We will first show that $\phi < \pi/2$ with a simple instantiation of the argument above. We will then proceed to refine the result to establish that $\phi$ is actually upper bounded by 36, a complicated function of manifold parameters.

**Regime 1:** $\phi \in [\pi/2, \pi]$. First we will show that for any $\phi$ in this range, we have a violation of minimum branch separation due to proximity between $y$ and $\mathrm{proj}_{\mathcal{M}} x$. First we will derive $\|\mathrm{proj}_{\mathcal{M}} x - y''\|_2$ (where $y''$ is the projection of $y$ onto the plane spanned by $v$ and $x - \mathrm{proj}_{\mathcal{M}} x$, shown in Figure 15). Consider the triangle with endpoints $\mathrm{proj}_{\mathcal{M}} x, y''$ and the projection of $y''$ onto the $v, v_T$ plane. We can use the triangle inequality to get

$$\|\mathrm{proj}_{\mathcal{M}} x - y''\|_2$$
$$= \sqrt{\left(\|x - \mathrm{proj}_{\mathcal{M}} x\|_2 \cos(\theta) + \|x - y\|_2 \cos(\phi)\right)^2 + \|x - \mathrm{proj}_{\mathcal{M}} x\|_2^2 \sin^2(\theta)}$$
$$= \sqrt{\|x - \mathrm{proj}_{\mathcal{M}} x\|_2^2 + 2\|x - \mathrm{proj}_{\mathcal{M}} x\|_2 \|x - y\|_2 \cos(\theta) \cos(\phi) + \|x - y\|_2^2 \cos^2(\phi)}.$$

Now we can solve for $\| \operatorname{proj}_{\mathcal{M}} x - y\|_2$ by looking at the triangle with endpoints $\operatorname{proj}_{\mathcal{M}} x, y$ and $y''$ as follows,

$$\| \operatorname{proj}_{\mathcal{M}} x - y\|_2$$
$$= \sqrt{\| \operatorname{proj}_{\mathcal{M}} x - y''\|_2^2 + \|x - y\|_2^2 \sin^2(\phi)}$$
$$= \sqrt{\|x - \operatorname{proj}_{\mathcal{M}} x\|_2^2 + 2\|x - \operatorname{proj}_{\mathcal{M}} x\|_2 \|x - y\|_2 \cos(\theta)\cos(\phi) + \|x - y\|_2^2}.$$

Note that $\| \operatorname{proj}_{\mathcal{M}} x - y\|_2$ is decreasing in $\phi$ for $\phi \in (0, \pi)$. Rearranging and solving for $\phi$ yields,

$$\phi = \arccos\left( \frac{\| \operatorname{proj}_{\mathcal{M}} x - y\|_2^2 - \|x - \operatorname{proj}_{\mathcal{M}} x\|_2^2 - \|x - y\|_2^2}{2\|x - \operatorname{proj}_{\mathcal{M}} x\|_2 \|x - y\|_2 \cos(\theta)} \right).$$

Now let

$$f(L) = \arccos\left( \frac{L^2 - \|x - \operatorname{proj}_{\mathcal{M}} x\|_2^2 - \|x - y\|_2^2}{2\|x - \operatorname{proj}_{\mathcal{M}} x\|_2 \|x - y\|_2 \cos(\theta)} \right).$$

We will evaluate $f(L)$ for $L = s_0(\mathcal{M}) - \tau$. We will then show shortly that if $L$ is less than $s_0(\mathcal{M}) - \tau$, we have a contradiction. Since $\| \operatorname{proj}_{\mathcal{M}} x - y\|_2$ is decreasing in $\phi$, we know that for all $\phi > f(L)$, $\| \operatorname{proj}_{\mathcal{M}} x - y\|_2 < s_0(\mathcal{M}) - \tau$. Evaluating $f$ at $s_0(\mathcal{M}) - \tau$,

$$f(s_0(\mathcal{M}) - \tau) = \arccos\left( \frac{(s_0(\mathcal{M}) - \tau)^2 - \|x - \operatorname{proj}_{\mathcal{M}} x\|_2^2 - \|x - y\|_2^2}{2\|x - \operatorname{proj}_{\mathcal{M}} x\|_2 \|x - y\|_2 \cos(\theta)} \right).$$

Now let's upper bound $f(s_0(\mathcal{M}) - \tau)$ as a means to simplify. Since $\arccos$ is decreasing on $(-1, 1)$

$$f(s_0(\mathcal{M}) - \tau) = \arccos\left( \frac{(s_0(\mathcal{M}) - \tau)^2 - \|x - \operatorname{proj}_{\mathcal{M}} x\|_2^2 - \|x - y\|_2^2}{2\|x - \operatorname{proj}_{\mathcal{M}} x\|_2 \|x - y\|_2 \cos(\theta)} \right)$$
$$< \arccos\left( \frac{(s_0(\mathcal{M}) - \tau)^2 - \tau^2 - \left(\sqrt{(s_0(\mathcal{M}) - \tau)^2 - \tau^2}\right)^2}{2\|x - \operatorname{proj}_{\mathcal{M}} x\|_2 \|x - y\|_2 \cos(\theta)} \right) \quad \text{Assumption 2}$$
$$= \arccos\left( \frac{(s_0(\mathcal{M}) - \tau)^2 - \tau^2 - (s_0(\mathcal{M}) - \tau)^2 + \tau^2}{2\|x - \operatorname{proj}_{\mathcal{M}} x\|_2 \|x - y\|_2 \cos(\theta)} \right)$$
$$= \arccos(0)$$
$$= \pi/2.$$

Since $f(s_0(\mathcal{M}) - \tau) < \pi/2$, we can say that for all $\phi \geq \pi/2 > f(s_0(\mathcal{M}) - \tau)$, $\| \operatorname{proj}_{\mathcal{M}} x - y\|_2 < s_0(\mathcal{M}) - \tau$. Observe that, since $(x, y)$ is a shortcut edge $d_{\mathcal{M}}(\operatorname{proj}_{\mathcal{M}} x, \operatorname{proj}_{\mathcal{M}} y) > (\pi + 1)r_0(\mathcal{M}) > \pi r_0(\mathcal{M})$. From the definition of minimum branch separation, we know that this implies $\| \operatorname{proj}_{\mathcal{M}} x - \operatorname{proj}_{\mathcal{M}} y\|_2 \geq s_0(\mathcal{M})$. Now let's apply the triangle inequality,

$$\| \operatorname{proj}_{\mathcal{M}} x - y\|_2 \geq \| \operatorname{proj}_{\mathcal{M}} x - \operatorname{proj}_{\mathcal{M}} y\|_2 - \| \operatorname{proj}_{\mathcal{M}} y - y\|_2$$
$$\geq s_0(\mathcal{M}) - \tau.$$

Thus, for $\phi \geq \pi/2$ we have a contradiction stemming from a violation of minimum branch separation. We therefore know that $\phi$ cannot exist in the interval $[\pi/2, \pi]$.

**Regime 2:** $\phi \in [0, \pi/2)$. While the previous result gives us a bound on $\phi$, we can sharpen it further. Since we are dealing with manifolds without boundary, we can use the exponential map $\exp_{\operatorname{proj}_{\mathcal{M}} x}$ to send the tangent vector $v_T$ to a geodesic arc of $\mathcal{M}$ denoted $\gamma$. Again we will show that if $\phi$ is sufficiently large, then the distance from $y$ to $\gamma$ violates the minimum branch separation in the same manner as we saw in Regime 1.

Consider the particular scenario where $\gamma$ curls away from $y$ maximally. We will approximate $\gamma$ near $\operatorname{proj}_{\mathcal{M}} x$ with its osculating circle $C$: a circle contained in the plane spanned by $v_T$ and $(y - \operatorname{proj}_{\mathcal{M}} x)$ with the smallest possible radius of curvature, $r_0(\mathcal{M})$. This particular instantiation is shown in Figure 15. Note that $C$ approximates a geodesic arc passing through $\operatorname{proj}_{\mathcal{M}} x$ with initial velocity $v_T$ and curls away from $y$ maximally. Thus the distance between all possible geodesic arcs

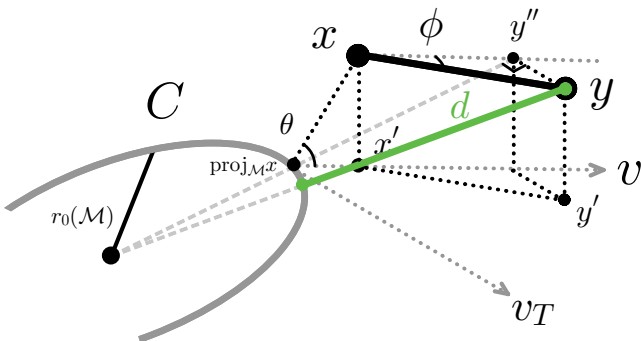

Figure 15: A visualization of the plane spanned by the vectors $(y - x)$ and $v$, and related quantities.

passing through $\text{proj}_{\mathcal{M}} x$ with velocity vector $v_T$ is approximately bounded by the distance between $C$ and $y$.

Now we want to derive an expression for the distance between $C$ and $y$; this distance will be denoted $d$. We will start by deriving $\| \text{proj}_{\mathcal{M}} x - y'' \|_2$ (where $y''$ is the projection of $y$ onto the plane spanned by $v$ and $x - \text{proj}_{\mathcal{M}} x$, shown in Figure 15). Consider the triangle defined by $\text{proj}_{\mathcal{M}} x, y''$ and the projection of $y''$ onto the $v, v_T$ plane and apply the Pythagorean theorem to say

$$\| \text{proj}_{\mathcal{M}} x - y'' \|_2 \tag{38}$$

$$= \sqrt{\left( \| x - \text{proj}_{\mathcal{M}} x \|_2 \cos(\theta) + \| x - y \|_2 \cos(\phi) \right)^2 + \| x - \text{proj}_{\mathcal{M}} x \|_2^2 \sin^2(\theta)} \tag{39}$$

$$= \sqrt{\| x - \text{proj}_{\mathcal{M}} x \|_2^2 + 2 \| x - \text{proj}_{\mathcal{M}} x \|_2 \| x - y \|_2 \cos(\theta) \cos(\phi) + \| x - y \|_2^2 \cos^2(\phi)}. \tag{40}$$

Now consider the triangle defined by the center of the osculating circle $C$, $y''$ and $y$. We can solve for $d + r_0(\mathcal{M})$ as,

$$d + r_0(\mathcal{M}) = \sqrt{\left( \| \text{proj}_{\mathcal{M}} x - y'' \|_2 + r_0(\mathcal{M}) \right)^2 + \| x - y \|_2^2 \sin^2(\phi)}.$$

Squaring both sides and plugging in eq. (40) yields the following for the right hand side,

$$\left( \| \text{proj}_{\mathcal{M}} x - y'' \|_2 + r_0(\mathcal{M}) \right)^2 + \| x - y \|_2^2 \sin^2(\phi) \tag{41}$$

$$= \| \text{proj}_{\mathcal{M}} x - y'' \|_2^2 + 2 r_0(\mathcal{M}) \| \text{proj}_{\mathcal{M}} x - y'' \|_2 + r_0(\mathcal{M})^2 + \| x - y \|_2^2 \sin^2(\phi) \tag{42}$$

$$= \| x - \text{proj}_{\mathcal{M}} x \|_2^2 + 2 \| x - \text{proj}_{\mathcal{M}} x \|_2 \| x - y \|_2 \cos(\theta) \cos(\phi) + \| x - y \|_2^2 \cos^2(\phi)$$
$$+ 2 r_0(\mathcal{M})$$
$$\cdot \sqrt{\| x - \text{proj}_{\mathcal{M}} x \|_2^2 + 2 \| x - \text{proj}_{\mathcal{M}} x \|_2 \| x - y \|_2 \cos(\theta) \cos(\phi) + \| x - y \|_2^2 \cos^2(\phi)} \tag{43}$$
$$+ r_0(\mathcal{M})^2 + \| x - y \|_2^2 \cos^2(\phi)$$

$$= \| x - \text{proj}_{\mathcal{M}} x \|_2^2 + 2 \| x - \text{proj}_{\mathcal{M}} x \|_2 \| x - y \|_2 \cos(\theta) \cos(\phi) + \| x - y \|_2^2 + r_0(\mathcal{M})^2$$
$$+ 2 r_0(\mathcal{M}) \sqrt{\| x - \text{proj}_{\mathcal{M}} x \|_2^2 + 2 \| x - \text{proj}_{\mathcal{M}} x \|_2 \| x - y \|_2 \cos(\theta) \cos(\phi) + \| x - y \|_2^2 \cos^2(\phi)}. \tag{44}$$

Observe that $\left( d + r_0(\mathcal{M}) \right)^2$ is decreasing in $\phi$ on the interval $\phi \in (0, \pi/2)$ since $\cos(\theta) \geq 0$, $\cos(\phi)$ is decreasing on $(0, \pi)$ and $\cos^2(\phi)$ is decreasing on $(0, \pi/2)$. To establish the contradiction we want to find the $\phi' \in (0, \pi/2)$ such that $d = s_0(\mathcal{M}) - \tau$; from that, we know that for any $\phi > \phi'$,

$d < s_0(\mathcal{M}) - \tau$. Rearranging 44 slightly yields

$$\underbrace{\Big(d + r_0(\mathcal{M})\Big)^2 - \|x - y\|_2^2 - r_0(\mathcal{M})^2}_{c_1}$$

$$= \underbrace{\|x - \mathrm{proj}_{\mathcal{M}}\, x\|_2^2}_{c_2} + \underbrace{2\|x - \mathrm{proj}_{\mathcal{M}}\, x\|_2 \|x - y\|_2 \cos(\theta)}_{c_3} \cos(\phi)$$

$$+ \underbrace{2r_0(\mathcal{M})}_{c_4} \sqrt{\underbrace{\|x - \mathrm{proj}_{\mathcal{M}}\, x\|_2^2}_{c_2} + \underbrace{2\|x - \mathrm{proj}_{\mathcal{M}}\, x\|_2 \|x - y\|_2 \cos(\theta)}_{c_3} \cos(\phi) + \underbrace{\|x - y\|_2^2}_{c_5} \cos^2(\phi)}.$$

Using these intermediate terms and defining $z = \cos(\phi)$, we see

$$c_1 = c_2 + c_3 z + c_4 \sqrt{c_2 + c_3 z + c_5 z^2}.$$

Rearranging to get a quadratic polynomial gives

$$\Big((c_1 - c_2) - c_3 z\Big)^2 = c_4^2 \Big(c_2 + c_3 z + c_5 z^2\Big)$$

$$(c_1 - c_2)^2 - 2c_3(c_1 - c_2)z + c_3^2 z^2 = c_4^2 c_2 + c_4^2 c_3 z + c_4^2 c_5 z^2$$

$$-\Big((c_1 - c_2)^2 - c_4^2 c_2\Big) + \Big(2c_3(c_1 - c_2) + c_4^2 c_3\Big)z - \Big(c_3^2 - c_4^2 c_5\Big)z^2 = 0.$$

Applying the quadratic formula,

$$z = \frac{-\Big(2c_3(c_1 - c_2) + c_4^2 c_3\Big) \pm \sqrt{\Big(2c_3(c_1 - c_2) + c_4^2 c_3\Big)^2 + 4\Big(c_4^2 c_5 - c_3^2\Big)\Big((c_1 - c_2)^2 - c_4^2 c_2\Big)}}{2\Big(c_4^2 c_5 - c_3^2\Big)}$$

(45)

Evaluating the denominator gives

$$2\Big(c_4^2 c_5 - c_3^2\Big) = 2\Big(-4\|x - \mathrm{proj}_{\mathcal{M}}\, x\|_2^2 \|x - y\|_2^2 \cos^2(\theta) + 4r_0(\mathcal{M})^2 \|x - y\|_2^2\Big) \qquad (46)$$

$$= 8\Big(-\|x - \mathrm{proj}_{\mathcal{M}}\, x\|_2^2 \|x - y\|_2^2 \cos^2(\theta) + r_0(\mathcal{M})^2 \|x - y\|_2^2\Big) \qquad (47)$$

$$= 8\|x - y\|_2^2 \Big(r_0(\mathcal{M})^2 - \|x - \mathrm{proj}_{\mathcal{M}}\, x\|_2^2 \cos^2(\theta)\Big). \qquad (48)$$

Evaluating the first term of the numerator,

$$2c_3(c_1 - c_2)$$

$$= 4\|x - \mathrm{proj}_{\mathcal{M}}\, x\|_2 \|x - y\|_2 \cos(\theta)$$

$$\cdot \Big((d + r_0(\mathcal{M}))^2 - \|x - y\|_2^2 - r_0(\mathcal{M})^2 - \|x - \mathrm{proj}_{\mathcal{M}}\, x\|_2^2\Big)$$

$$= 4\|x - \mathrm{proj}_{\mathcal{M}}\, x\|_2 \|x - y\|_2 \cos(\theta)\Big(d^2 + 2dr_0(\mathcal{M}) - \|x - y\|_2^2 - \|x - \mathrm{proj}_{\mathcal{M}}\, x\|_2^2\Big),$$

evaluating the second term of the numerator,

$$c_4^2 c_3 = 8r_0(\mathcal{M})^2 \|x - \mathrm{proj}_{\mathcal{M}}\, x\|_2 \|x - y\|_2 \cos(\theta)$$

and combining,

$$2c_3(c_1 - c_2) + c_4^2 c_3 = 4\|x - \mathrm{proj}_{\mathcal{M}}\, x\|_2 \|x - y\|_2 \cos(\theta)\Big(d^2 + 2dr_0(\mathcal{M})$$

$$- \|x - y\|_2^2 - \|x - \mathrm{proj}_{\mathcal{M}}\, x\|_2^2 + 2r_0(\mathcal{M})^2\Big) \quad (49)$$

Now we will evaluate terms inside of the square root in (45). Note that the first term is (49) squared, while the second term includes two times (48). The remaining part of the second term is

$$(c_1 - c_2)^2 - c_4^2 c_2 = \Big(d^2 + 2dr_0(\mathcal{M}) - \|x - y\|_2^2 - \|x - \mathrm{proj}_{\mathcal{M}}\, x\|_2^2\Big)^2$$

$$- 4r_0(\mathcal{M})^2 \|x - \mathrm{proj}_{\mathcal{M}}\, x\|_2^2. \quad (50)$$

Combining everything inside the square root yields,

$$16\|x - \text{proj}_{\mathcal{M}} x\|_2^2 \|x - y\|_2^2 \cos^2(\theta)$$
$$\cdot \left[ \left( d^2 + 2dr_0(\mathcal{M}) - \|x - y\|_2^2 - \|x - \text{proj}_{\mathcal{M}} x\|_2^2 \right) + 2r_0(\mathcal{M})^2 \right]^2$$
$$- \left[ 16\|x - y\|_2^2 \|x - \text{proj}_{\mathcal{M}} x\|^2 \cos^2(\theta) - 16\|x - y\|_2^2 r_0(\mathcal{M})^2 \right]$$
$$\cdot \left[ \left( d^2 + 2dr_0(\mathcal{M}) - \|x - y\|_2^2 - \|x - \text{proj}_{\mathcal{M}} x\|_2^2 \right)^2 - 4r_0(\mathcal{M})^2 \|x - \text{proj}_{\mathcal{M}} x\|_2^2 \right]$$
$$= 16\|x - \text{proj}_{\mathcal{M}} x\|_2^2 \|x - y\|_2^2 \cos^2(\theta) \left( d^2 + 2dr_0(\mathcal{M}) - \|x - y\|_2^2 - \|x - \text{proj}_{\mathcal{M}} x\|_2^2 \right)^2$$
$$+ 64\|x - \text{proj}_{\mathcal{M}} x\|_2^2 \|x - y\|_2^2 \cos^2(\theta) r_0(\mathcal{M})^2 \left( d^2 + 2dr_0(\mathcal{M}) - \|x - y\|_2^2 - \|x - \text{proj}_{\mathcal{M}} x\|_2^2 \right)$$
$$+ 64 r_0(\mathcal{M})^4 \|x - \text{proj}_{\mathcal{M}} x\|_2^2 \|x - y\|_2^2 \cos^2(\theta)$$
$$- 16\|x - \text{proj}_{\mathcal{M}} x\|_2^2 \|x - y\|_2^2 \cos^2(\theta) \left( d^2 + 2dr_0(\mathcal{M}) - \|x - y\|_2^2 - \|x - \text{proj}_{\mathcal{M}} x\|_2^2 \right)^2$$
$$+ 64\|x - y\|_2^2 \|x - \text{proj}_{\mathcal{M}} x\|_2^4 \cos^2(\theta) r_0(\mathcal{M})^2$$
$$+ 16\|x - y\|_2^2 r_0(\mathcal{M})^2 \left( d^2 + 2dr_0(\mathcal{M}) - \|x - y\|_2^2 - \|x - \text{proj}_{\mathcal{M}} x\|_2^2 \right)^2$$
$$- 64\|x - y\|_2^2 r_0(\mathcal{M})^4 \|x - \text{proj}_{\mathcal{M}} x\|_2^2.$$

Continuing to simplify yields

$$16\|x - y\|_2^2 r_0(\mathcal{M})^2 \left[ \left( d^2 + 2dr_0(\mathcal{M}) - \|x - y\|_2^2 - \|x - \text{proj}_{\mathcal{M}} x\|_2^2 \right)^2 \right.$$
$$+ 4\|x - \text{proj}_{\mathcal{M}} x\|_2^2 \cos^2(\theta) \left( d^2 + 2dr_0(\mathcal{M}) - \|x - y\|_2^2 - \|x - \text{proj}_{\mathcal{M}} x\|_2^2 \right)$$
$$+ 4 r_0(\mathcal{M})^2 \|x - \text{proj}_{\mathcal{M}} x\|_2^2 \cos^2(\theta)$$
$$- 4 r_0(\mathcal{M})^2 \|x - \text{proj}_{\mathcal{M}} x\|_2^2$$
$$\left. + 4\|x - \text{proj}_{\mathcal{M}} x\|_2^4 \cos^2(\theta) \right]. \quad (51)$$

Now we will complete the square in the large bracket of 51,

$$\left( d^2 + 2dr_0(\mathcal{M}) - \|x - y\|_2^2 - \|x - \text{proj}_{\mathcal{M}} x\|_2^2 \right)^2$$
$$+ 4\|x - \text{proj}_{\mathcal{M}} x\|_2^2 \cos^2(\theta) \left( d^2 + 2dr_0(\mathcal{M}) - \|x - y\|_2^2 - \|x - \text{proj}_{\mathcal{M}} x\|_2^2 \right)$$
$$+ 4\cos^4(\theta)\|x - \text{proj}_{\mathcal{M}} x\|_2^4 - 4\cos^4(\theta)\|x - \text{proj}_{\mathcal{M}} x\|_2^4 \quad (52)$$
$$+ 4 r_0(\mathcal{M})^2 \|x - \text{proj}_{\mathcal{M}} x\|_2^2 \cos^2(\theta)$$
$$- 4 r_0(\mathcal{M})^2 \|x - \text{proj}_{\mathcal{M}} x\|_2^2$$
$$+ 4\|x - \text{proj}_{\mathcal{M}} x\|_2^4 \cos^2(\theta).$$

Now the first three terms and last four terms can be factored,

$$\left( \left( d^2 + 2dr_0(\mathcal{M}) - \|x - y\|_2^2 - \|x - \text{proj}_{\mathcal{M}} x\|_2^2 \right) + 2\cos^2(\theta)\|x - \text{proj}_{\mathcal{M}} x\|_2^2 \right)^2$$
$$- 4\cos^4(\theta)\|x - \text{proj}_{\mathcal{M}} x\|_2^4$$
$$+ 4 r_0(\mathcal{M})^2 \|x - \text{proj}_{\mathcal{M}} x\|_2^2 \cos^2(\theta) \quad (53)$$
$$- 4 r_0(\mathcal{M})^2 \|x - \text{proj}_{\mathcal{M}} x\|_2^2$$
$$+ 4\|x - \text{proj}_{\mathcal{M}} x\|_2^4 \cos^2(\theta).$$

Further simplifying,

$$
\left( \left( \left( d^2 + 2dr_0(\mathcal{M}) - \|x - y\|_2^2 - \|x - \text{proj}_{\mathcal{M}} x\|_2^2 \right) + 2\cos^2(\theta)\|x - \text{proj}_{\mathcal{M}} x\|_2^2 \right)^2 \right.
$$
$$
- 4\|x - \text{proj}_{\mathcal{M}} x\|_2^2
$$
$$
\cdot \left( \|x - \text{proj}_{\mathcal{M}} x\|_2^2 \cos^4(\theta) + r_0(\mathcal{M})^2 \sin^2(\theta) - \|x - \text{proj}_{\mathcal{M}} x\|_2^2 \cos^2(\theta) \right) \tag{54}
$$
$$
= \left( \left( d^2 + 2dr_0(\mathcal{M}) - \|x - y\|_2^2 - \|x - \text{proj}_{\mathcal{M}} x\|_2^2 \right) + 2\cos^2(\theta)\|x - \text{proj}_{\mathcal{M}} x\|_2^2 \right)^2
$$
$$
- 4\|x - \text{proj}_{\mathcal{M}} x\|_2^2 \left( r_0(\mathcal{M})^2 \sin^2(\theta) - \|x - \text{proj}_{\mathcal{M}} x\|_2^2 \cos^2(\theta) \sin^2(\theta) \right) \tag{55}
$$
$$
= \underbrace{ \left( \left( d^2 + 2dr_0(\mathcal{M}) - \|x - y\|_2^2 - \|x - \text{proj}_{\mathcal{M}} x\|_2^2 \right) + 2\cos^2(\theta)\|x - \text{proj}_{\mathcal{M}} x\|_2^2 \right)^2 }_{:= a^2}
$$
$$
\underbrace{ - 4\|x - \text{proj}_{\mathcal{M}} x\|_2^2 \sin^2(\theta) \left( r_0(\mathcal{M})^2 - \|x - \text{proj}_{\mathcal{M}} x\|_2^2 \cos^2(\theta) \right) }_{:= b^2} \tag{56}
$$

where the two intermediate steps leverage the fact that $1 - \cos^2(\theta) = \sin^2(\theta)$. The equation 45 has two roots, and one of the two will result in a term on the right-hand side of 45 that is positive (since the denominator is necessarily positive); we will consider this root. Recall that $d$ is decreasing in $\phi$ on the interval $\phi \in (0, \pi/2)$. We want to find a $\phi'$ such that for all $\phi > \phi'$, $d < s_0(\mathcal{M}) - \tau$. It suffices then to find an upper bound for $\phi'$, which implies finding a lower bound on $\cos(\phi')$.

Let's denote 56 with $B$, and recall that it represents the expression in the large bracket of 51. Also recall that 51 is a simplified version of the term in the radical of 45. We can express the square root of 51 with $4\|x - y\|_2 r_0(\mathcal{M})\sqrt{B}$. Observe that $\sqrt{B}$ can be written as $\sqrt{a^2 - b^2}$ where $a, b > 0$. Since $\sqrt{a^2 - b^2} \geq a - b$ for $a \geq b > 0$, we can say

$$
4\|x - y\|_2 r_0(\mathcal{M})\sqrt{B}
$$
$$
\geq 4\|x - y\|_2 r_0(\mathcal{M}) \left( \left( d^2 + 2dr_0(\mathcal{M}) - \|x - y\|_2^2 - \|x - \text{proj}_{\mathcal{M}} x\|_2^2 \right) \right.
$$
$$
+ 2\cos^2(\theta)\|x - \text{proj}_{\mathcal{M}} x\|_2^2
$$
$$
\left. - 2\|x - \text{proj}_{\mathcal{M}} x\|_2 \sin(\theta)\sqrt{r_0(\mathcal{M})^2 - \|x - \text{proj}_{\mathcal{M}} x\|_2^2 \cos^2(\theta)} \right). \tag{57}
$$

Note that the following steps will demonstrate $a - b \geq 0$, justifying the previous step when evaluating at $d = s_0(\mathcal{M}) - \tau$. Note that

$$
d^2 - \|x - y\|_2^2 - \|x - \text{proj}_{\mathcal{M}} x\|_2^2 = (s_0(\mathcal{M}) - \tau)^2 - \|x - y\|_2^2 - \|x - \text{proj}_{\mathcal{M}} x\|_2^2 \tag{58}
$$
$$
\geq (s_0(\mathcal{M}) - \tau)^2 - (\sqrt{(s_0(\mathcal{M}) - \tau)^2 - \tau^2})^2 - \tau^2 \tag{59}
$$
$$
= 0. \tag{60}
$$

Thus,

$$
a - b \geq 2(s_0(\mathcal{M}) - \tau)r_0(\mathcal{M}) + 2\cos^2(\theta)\|x - \text{proj}_{\mathcal{M}} x\|_2^2
$$
$$
- 2\|x - \text{proj}_{\mathcal{M}} x\|_2 \sin(\theta)\sqrt{r_0(\mathcal{M})^2 - \|x - \text{proj}_{\mathcal{M}} x\|_2^2 \cos^2(\theta)} \right) \tag{61}
$$
$$
\geq 2(s_0(\mathcal{M}) - \tau)r_0(\mathcal{M}) - 2\tau r_0(\mathcal{M}). \tag{62}
$$

Note that the last term is necessarily positive since we have stipulated that $s_0(\mathcal{M}) > 3\tau$. In fact, $a \geq 2b$, a fact that we can use to sharpen the bound $\sqrt{a^2 - b^2} \geq a - b$. When $a \geq 2b$, $\sqrt{a^2 - b^2} \geq$

$a - 0.27b$, which can be verified easily by plugging in and bounding. Thus,

$$
\begin{aligned}
4\|x - y\|_2 r_0(\mathcal{M})&\sqrt{B} \\
&\geq 4\|x - y\|_2 r_0(\mathcal{M})\bigg(\Big(d^2 + 2dr_0(\mathcal{M}) - \|x - y\|_2^2 - \|x - \mathrm{proj}_{\mathcal{M}}\, x\|_2^2\Big) \\
&\quad + 2\cos^2(\theta)\|x - \mathrm{proj}_{\mathcal{M}}\, x\|_2^2 \\
&\quad - 0.54\|x - \mathrm{proj}_{\mathcal{M}}\, x\|_2 \sin(\theta)\sqrt{r_0(\mathcal{M})^2 - \|x - \mathrm{proj}_{\mathcal{M}}\, x\|_2^2 \cos^2(\theta)}\bigg).
\end{aligned}
\tag{63}
$$

Now we will evaluate a bound on one of the roots of 45 by combining 63, 49 and 48,

$$
\begin{aligned}
z \geq \; & \frac{\Big(r_0(\mathcal{M}) - \|x - \mathrm{proj}_{\mathcal{M}}\, x\|_2 \cos(\theta)\Big)\Big(d^2 + 2dr_0(\mathcal{M}) - \|x - y\|_2^2 - \|x - \mathrm{proj}_{\mathcal{M}}\, x\|_2^2\Big)}{2\|x - y\|_2\Big(r_0(\mathcal{M})^2 - \|x - \mathrm{proj}_{\mathcal{M}}\, x\|_2^2 \cos^2(\theta)\Big)} \\
& + \frac{-2r_0(\mathcal{M})\|x - \mathrm{proj}_{\mathcal{M}}\, x\|_2}{2\|x - y\|_2\Big(r_0(\mathcal{M})^2 - \|x - \mathrm{proj}_{\mathcal{M}}\, x\|_2^2 \cos^2(\theta)\Big)} \\
& \cdot \Big(r_0(\mathcal{M}) \cos(\theta) - \|x - \mathrm{proj}_{\mathcal{M}}\, x\|_2 \cos^2(\theta) + 0.27\sin(\theta)\sqrt{r_0(\mathcal{M})^2 - \|x - \mathrm{proj}_{\mathcal{M}}\, x\|_2^2 \cos^2(\theta)}\Big).
\end{aligned}
\tag{64}
$$

Bounding the numerator of the second term from below (since the denominator is nonnegative),

$$
\begin{aligned}
-2r_0(\mathcal{M})\|x - \mathrm{proj}_{\mathcal{M}}\, x\|_2 \Big(& r_0(\mathcal{M}) \cos(\theta) - \|x - \mathrm{proj}_{\mathcal{M}}\, x\|_2 \cos^2(\theta) \\
& + 0.27\sin(\theta)\sqrt{r_0(\mathcal{M})^2 - \|x - \mathrm{proj}_{\mathcal{M}}\, x\|_2^2 \cos^2(\theta)}\Big)
\end{aligned}
\tag{65}
$$

$$
\geq -2r_0(\mathcal{M})\tau\Big(r_0(\mathcal{M}) + 0.27r_0(\mathcal{M})\Big)
$$

$$
= -2.54r_0(\mathcal{M})^2\tau.
\tag{66}
$$

Now we will bound the numerator of the first term from below. We can apply the lower bound from 60 with $d$ evaluated at $s_0(\mathcal{M}) - \tau$ to get

$$
\begin{aligned}
\Big(r_0(\mathcal{M}) - &\|x - \mathrm{proj}_{\mathcal{M}}\, x\|_2 \cos(\theta)\Big) \\
&\Big((s_0(\mathcal{M}) - \tau)^2 + 2dr_0(\mathcal{M}) - \|x - y\|_2^2 - \|x - \mathrm{proj}_{\mathcal{M}}\, x\|_2^2\Big)
\end{aligned}
\tag{67}
$$

$$
\geq (r_0(\mathcal{M}) - \tau)(2(s_0(\mathcal{M}) - \tau)r_0(\mathcal{M}))
\tag{68}
$$

$$
= 2(r_0(\mathcal{M}) - \tau)(s_0(\mathcal{M}) - \tau)r_0(\mathcal{M}).
\tag{69}
$$

Putting it all together, we get

$$
z \geq \frac{2(r_0(\mathcal{M}) - \tau)(s_0(\mathcal{M}) - \tau)r_0(\mathcal{M}) - 2.54r_0(\mathcal{M})^2\tau}{2\|x - y\|_2\Big(r_0(\mathcal{M})^2 - \|x - \mathrm{proj}_{\mathcal{M}}\, x\|_2^2 \cos^2(\theta)\Big)}
\tag{70}
$$

$$
= \frac{(r_0(\mathcal{M}) - \tau)(s_0(\mathcal{M}) - \tau)r_0(\mathcal{M}) - 1.27r_0(\mathcal{M})^2\tau}{\|x - y\|_2\Big(r_0(\mathcal{M})^2 - \|x - \mathrm{proj}_{\mathcal{M}}\, x\|_2^2 \cos^2(\theta)\Big)}
\tag{71}
$$

Now we will show that the numerator is necessarily larger than zero, allowing us to further bound $z$ by upper bounding the denominator. Note that the numerator is greater than zero if and only if

$$
(r_0(\mathcal{M}) - \tau)(s_0(\mathcal{M}) - \tau)r_0(\mathcal{M}) > 1.27r_0(\mathcal{M})^2\tau
$$

$$
\iff \frac{(r_0(\mathcal{M}) - \tau)(s_0(\mathcal{M}) - \tau)}{r_0(\mathcal{M})\tau} > 1.27
$$

Note that $(r_0(\mathcal{M}) - \tau)/r_0(\mathcal{M}) > 2/3$ since $r_0(\mathcal{M}) > 3\tau$, and $(s_0(\mathcal{M}) - \tau)/\tau > 2$ since $s_0(\mathcal{M}) > 3\tau$. Thus, the left-hand side of the equation above is larger than $4/3$ which is larger than $1.27$,

rendering the statement true and the numerator positive. Now we can continue to bound $z$ by upper bounding the denominator,

$$z \geq \frac{(r_0(\mathcal{M}) - \tau)(s_0(\mathcal{M}) - \tau)r_0(\mathcal{M}) - 1.27 r_0(\mathcal{M})^2 \tau}{\|x - y\|_2 \left( r_0(\mathcal{M})^2 - \|x - \text{proj}_{\mathcal{M}} x\|_2^2 \cos^2(\theta) \right)} \tag{72}$$

$$> \frac{(r_0(\mathcal{M}) - \tau)(s_0(\mathcal{M}) - \tau)r_0(\mathcal{M}) - 1.27 r_0(\mathcal{M})^2 \tau}{r_0(\mathcal{M})^2 \sqrt{(s_0(\mathcal{M}) - \tau)^2 - \tau^2}} \tag{73}$$

$$= \frac{(r_0(\mathcal{M}) - \tau)(s_0(\mathcal{M}) - \tau) - 1.27 r_0(\mathcal{M})\tau}{r_0(\mathcal{M}) \sqrt{(s_0(\mathcal{M}) - \tau)^2 - \tau^2}}. \tag{74}$$

Finally, we can derive a bound on $\phi'$ from 72 since $z = \arccos(\phi')$,

$$\phi' < \arccos\left( \frac{(r_0(\mathcal{M}) - \tau)(s_0(\mathcal{M}) - \tau)r_0(\mathcal{M}) - 1.27 r_0(\mathcal{M})^2 \tau}{r_0(\mathcal{M})^2 \sqrt{(s_0(\mathcal{M}) - \tau)^2 - \tau^2}} \right). \tag{75}$$

Note that the fact that the numerator and denominator are positive implies the right-hand side of 75 is necessarily $< \pi/2$. For all $\phi > \phi'$ we have $d < s_0(\mathcal{M}) - \tau$. Thus it must also be true that for all $\phi$ larger than the right-hand side of 75, $d < s_0(\mathcal{M}) - \tau$. however, it remains to show that this is in fact a violation of branch separation. Since the nearest point to $y$ on $C$ might not be (and likely is not) $\text{proj}_{\mathcal{M}} x$, we need to show that the manifold distance between $y$ and this point is large. For ease of notation, let $C(t_y)$ be the nearest point to $y$ on $C$.

First we need to bound the distance $d_{\mathcal{M}}(\text{proj}_{\mathcal{M}} x, C(t_y))$. Since $C$ has radius of curvature $r_0(\mathcal{M})$, $d_{\mathcal{M}}(\text{proj}_{\mathcal{M}} x, C(t_y))$ can be expressed as

$$d_{\mathcal{M}}(\text{proj}_{\mathcal{M}} x, C(t_y)) = r_0(\mathcal{M}) \arctan\left( \frac{\|x - y\|_2 \sin(\phi)}{\|\text{proj}_{\mathcal{M}} x - y''\|_2 + r_0(\mathcal{M})} \right). \tag{76}$$

This follows from Figure 15, as we are approximating the geodesic arc $\gamma$ with $C$. The distance along $C$ from $\text{proj}_{\mathcal{M}} x$ to $C(t_y)$ can be written as the angle swept out times the radius of curvature. Note that $\arctan(x)$ is an increasing function of $x$, and $\arctan(x) \leq x$ for $x \geq 0$. Since $\phi \in [0, \pi/2)$ the argument of $\arctan$ in 76 is necessarily non-negative, allowing us to bound it as follows

$$d_{\mathcal{M}}(\text{proj}_{\mathcal{M}} x, C(t_y)) = r_0(\mathcal{M}) \arctan\left( \frac{\|x - y\|_2 \sin(\phi)}{\|\text{proj}_{\mathcal{M}} x - y''\|_2 + r_0(\mathcal{M})} \right) \tag{77}$$

$$\leq r_0(\mathcal{M}) \arctan\left( \frac{\|x - y\|_2 \sin(\phi)}{r_0(\mathcal{M})} \right) \tag{78}$$

$$\leq r_0(\mathcal{M}) \cdot \frac{\|x - y\|_2 \sin(\phi)}{r_0(\mathcal{M})} \tag{79}$$

$$= \|x - y\|_2 \sin(\phi). \tag{80}$$

Now we can apply assumptions on $\epsilon$ to bound 80 as a function of manifold embedding parameters.

$$d_{\mathcal{M}}(\text{proj}_{\mathcal{M}} x, C(t_y)) \leq \|x - y\|_2 \sin(\phi) \tag{81}$$

$$\leq \epsilon \tag{82}$$

$$< r_0(\mathcal{M}). \tag{83}$$

Now we can apply the triangle inequality,

$$d_{\mathcal{M}}(C(t_y), \text{proj}_{\mathcal{M}} y) \geq d_{\mathcal{M}}(\text{proj}_{\mathcal{M}} x, \text{proj}_{\mathcal{M}} y) - d_{\mathcal{M}}(C(t_y), \text{proj}_{\mathcal{M}} x) \tag{84}$$

$$> (\pi + 1)r_0(\mathcal{M}) - r_0(\mathcal{M}) \tag{85}$$

$$= \pi r_0(\mathcal{M}). \tag{86}$$

As before, we will apply the definition of *minimum branch separation*, which states that if $d_{\mathcal{M}}(C(t_y), \text{proj}_{\mathcal{M}} y)$ is larger than $\pi r_0(\mathcal{M})$, then $d = \|C(t_y) - \text{proj}_{\mathcal{M}} y\|_2 \geq s_0(\mathcal{M})$. Applying the triangle inequality one more time leads to the conclusion that $\|C(t_y) - y\|_2 \geq s_0(\mathcal{M}) - \tau$. However, we see that when $\phi$ does not satisfy 75, $d < s_0(\mathcal{M}) - \tau$ and we have a contradiction.

$\square$

Lemma A.3 formalizes and proves the fact that shortcut edges are necessarily oriented close to a normal direction to $\mathcal{M}$ from the perspective of each endpoint. To leverage this result, we then need to be able to say that this orientation implies a concentration of measure away from the midpoint of all shortcuts. Figure 11 provides a visualization of this phenomenon, establishing intuition for why this is true.

Concretely, Lemma A.4 derives the measure of the sets $U_x$ and $B_\epsilon(x)$ detailed in Figure 12 in the limit of vanishing noise. The expressions for the measures of these sets are obtained by integrating the probability density function $\rho$ (detailed in eq. (1)) over each respective set. What results is an expression that is a function of $\phi$, the angle between the edge $(x, y)$ and the closest normal direction of $\mathcal{M}$ at $x$.

Intuitively, the result matches what we would expect to see in the no noise ($\sigma = 0$) scenario. The $m$-dimensional measure of $U_x$ when the center of $B_\epsilon(x)$ is displaced at a distance $\|x - \mathrm{proj}_\mathcal{M} x\|_2$ from a $m$-dimensional manifold $\mathcal{M}$ would simply be (in the locally flat case) proportional to the volume of the intersection, which is a portion of a $m$ dimensional ball of radius $\sqrt{\epsilon^2 - \|x - \mathrm{proj}_\mathcal{M} x\|_2^2}$. The conditional probability of a point being in $U_x$ given $B_\epsilon(x)$ would therefore be the ratio of the volume of this portion of a $m$ dimensional ball to the full volume of the $m$ dimensional ball. Figure 16 provides a visualization for a 2-dimensional manifold embedded in $\mathbb{R}^3$.

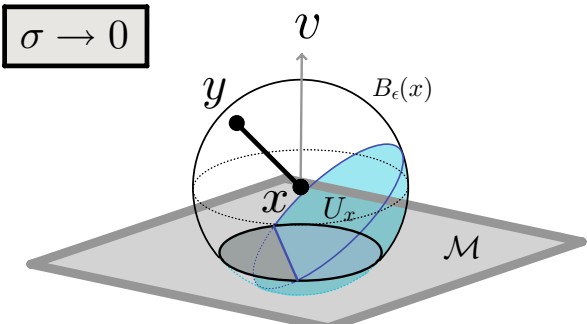

Figure 16: A visualization of $v \in N_{\mathrm{proj}_\mathcal{M} x}$, $U_x$ and the edge $(x, y)$ for a 2-dimensional manifold embedded in $\mathbb{R}^3$. Observe that the measure of $B_\epsilon(x) \setminus U_x$ approaches the volume of the intersection of $B_\epsilon(x) \setminus U_x$ and $\mathcal{M}$ as $\sigma \to 0$. Similarly, the measure of $U_x$ approaches the volume of the intersection of $U_x$ and $\mathcal{M}$ as $\sigma \to 0$. Both of these regions are shown as the 2-dimensional dark grey and matte blue shaded regions respectively.

**Lemma A.4.** *Suppose $(x, y)$ is an edge in a nearest neighbor graph built from data consisting of samples from the probability density function defined by 1. Define $v$ to be the vector in the normal space of $\mathcal{M}$ at $\mathrm{proj}_\mathcal{M} x$ such that the angle $\phi$ between $v$ and $(y - x)$ is minimized. Also define*

$$f(\phi) = \cot(\phi)\left(\sec(\phi)\left(\frac{\epsilon - \|x - y\|_2}{2}\right) - d_v\right)$$

*and,*

$$R = \sqrt{\epsilon^2 - \|x - \mathrm{proj}_\mathcal{M} x\|_2^2}$$

*and finally,*

$$r(z) = \begin{cases} \sqrt{\epsilon^2 - \|x - \mathrm{proj}_\mathcal{M} x\|_2^2 - z^2} & |z| \leq \sqrt{\epsilon^2 - \|x - \mathrm{proj}_\mathcal{M} x\|_2^2} \\ 0 & otherwise \end{cases}. \tag{87}$$

*Then*

$$\lim_{\sigma \to 0^+} \mathbb{P}[\, a \in U_x \mid a \in B_\epsilon(x)\,] \geq \frac{\int_{f(\phi)}^{\sqrt{\epsilon^2 - \|x - \mathrm{proj}_\mathcal{M} x\|_2^2}} Vol(B_{r(z)}^{m-1}(0))dz}{Vol(B_R^m(0))} \tag{88}$$

> *where $m$ is the dimension of $\mathcal{M}$.*

*Proof.* Rewriting $\mathbb{P}[\, a \in U_x \,|\, a \in B_\epsilon(x) \,]$,

$$\mathbb{P}[\, a \in U_x \,|\, a \in B_\epsilon(x) \,] = \frac{\mathbb{P}[\, a \in U_x \,]}{\mathbb{P}[\, a \in B_\epsilon(x) \,]} \tag{89}$$

$$= \frac{\mu(U_x)}{\mu(B_\epsilon(x))} \tag{90}$$

Since the probability density in $B_\epsilon(x)$ is nonuniform, we need to integrate $\rho$ over the regions we are interested in. To do so, we will integrate along $v \in N_{\mathrm{proj}_{\mathcal{M}} x}\mathcal{M}$ and use a locally flat approximation of $\mathcal{M}$. Observe that $(y - x)$ can be written as $\alpha v_T + \beta v$, where $v_T \in T_{\mathrm{proj}_{\mathcal{M}} x}$, and $v$ is the vector in $N_{\mathrm{proj}_{\mathcal{M}} x}\mathcal{M}$ that forms the smallest angle $\phi$ with the vector $(y - x)$. Also note that the vector $(x - \mathrm{proj}_{\mathcal{M}} x)$ is oriented normal to $\mathcal{M}$ at $\mathrm{proj}_{\mathcal{M}} x$ as well. It can therefore be written as $d_v v + d_x v_N$, where $v$ is the vector defined previously and $v_N$ is some other vector in the normal space. It follows that $d_v = \|x - \mathrm{proj}_{\mathcal{M}} x\|_2 \cos(\theta)$ and $d_x = \|x - \mathrm{proj}_{\mathcal{M}} x\|_2 \sin(\theta)$, where $\theta$ is the angular displacement between $(x - \mathrm{proj}_{\mathcal{M}} x)$ and $v$. A visualization of these quantities and vectors is provided in Figure 17.

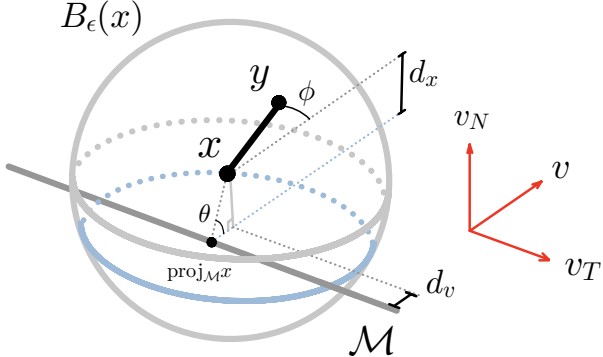

Figure 17: A visualization of quantities defined and used for Lemma A.4.

All coordinates mentioned from here onwards specify displacements from the point $\mathrm{proj}_{\mathcal{M}} x$. Note that $m$ orthogonal ambient directions are oriented tangent to $\mathcal{M}$ at $\mathrm{proj}_{\mathcal{M}} x$, while $D - m$ directions are oriented normal to $\mathcal{M}$ at $\mathrm{proj}_{\mathcal{M}} x$. Since the probability density is constant in tangential directions, we will begin by evaluating the measure of the subspace spanned by the normal directions directly; thereafter we will integrate along the tangent directions.

Let $u_1, u_2, u_{m+3}, \ldots, u_D$ denote coordinates in an orthonormal basis of $N_{\mathrm{proj}_{\mathcal{M}} x}\mathcal{M}$, where $u_1$ corresponds with direction $v$ and $u_2$ corresponds with direction $v_N$. Now let $z_3, \ldots, z_{m+2}$ denote coordinates in the $m$ tangential directions. We choose to define these coordinates with these indices as it reflects the order in which we will integrate. This choice will become clear deeper into the proof.

Observe that $x$'s only nonzero coordinates (as we have just defined them) are $u_1$ and $u_2$. Therefore the probability density $\rho$ is symmetric about $u_i = 0$ for $i \in \{m+3, \ldots, D\}$. This arises from the fact that the $u$ coordinates are the only coordinates in which the probability density varies - $z$ coordinates have no effect (due to our locally flat approximation). Therefore, we will begin by evaluating the measure of the subspace of $B_\epsilon(x)$ (centered at fixed coordinates $u_1, u_2, z_3, \ldots, z_{m+2}$) spanned by normal directions $u_{m+3}, \ldots, u_D$. The probability density at any point in this subspace is simply a

function of $u$ coordinates,

$$\rho(u_1, u_2, z_3, \ldots, z_{m+2}, u_{m+3}, \ldots, u_D) = \frac{1}{Z} \exp\left(-\frac{u_1^2 + u_2^2 + \sum_{i=d+3}^{D} u_i^2}{2\sigma^2}\right)$$

$$= \frac{1}{Z} \exp\left(-\frac{u_1^2 + u_2^2}{2\sigma^2}\right) \exp\left(-\frac{\sum_{i=d+3}^{D} u_i^2}{2\sigma^2}\right).$$

Observe that the point $(u_1, u_2, z_3, \ldots, z_{m+2}, 0, \ldots, 0)$ is at the same distance from the boundary of the tubular neighborhood in any direction spanned by $u_i$ for $i \in \{m+3, \ldots, D\}$. While we do not yet have an exact expression for that distance, we can represent it as some function of $(u_1, u_2, z_3, \ldots, z_{m+2})$, denoted $r(u_1, u_2, z_3, \ldots, z_{m+2}) = r$. Put another way, if we consider the intersection of $\text{Tub}_\tau \mathcal{M}$, $B_\epsilon(x)$ and the subspace spanning $u_{m+3}, \ldots, u_D$ centered at $(u_1, u_2, z_3, \ldots, z_{m+2}, 0, \ldots, 0)$, it looks like a $D - m - 2$ dimensional ball of radius $r$.

Thus, we can evaluate the measure of this subspace of $B_\epsilon(x)$ centered at the point with coordinates $(u_1, u_2, z_3, \ldots, z_{m+2}, 0, \ldots, 0)$ spanned by normal directions $u_{m+3}, \ldots, u_D$ as

$$\frac{1}{Z} \exp\left(-\frac{u_1^2 + u_2^2}{2\sigma^2}\right) \int_{\mathbf{u}_{m+3:D} \in B_r(0)} \exp\left(-\frac{\sum_{i=d+3}^{D} u_i^2}{2\sigma^2}\right) dV.$$

Now we will manipulate the term in the integral so that it becomes equivalent to the CDF of a $\chi^2$ distribution with $D - m - 2$ degrees of freedom,

$$\frac{1}{Z} \exp\left(-\frac{u_1^2 + u_2^2}{2\sigma^2}\right) \int_{\mathbf{u}_{m+3:D} \in B_r(0)} \exp\left(-\frac{\sum_{i=d+3}^{D} u_i^2}{2\sigma^2}\right) dV$$

$$= \frac{(2\pi)^{\frac{D-m-2}{2}}}{Z} \exp\left(-\frac{u_1^2 + u_2^2}{2\sigma^2}\right) \int_{\mathbf{u}_{m+3:D} \in B_r(0)} \frac{1}{(2\pi)^{\frac{D-m-2}{2}}} \exp\left(-\frac{\sum_{i=d+3}^{D} u_i^2}{2\sigma^2}\right) dV$$

$$= \underbrace{\frac{\sigma^{D-m-2}(2\pi)^{\frac{D-m-2}{2}}}{Z}}_{:= C} \exp\left(-\frac{u_1^2 + u_2^2}{2\sigma^2}\right)$$

$$\cdot \underbrace{\int_{\mathbf{w}_{m+3:D} \in B_{r/\sigma}(0)} \frac{1}{(2\pi)^{\frac{D-m-2}{2}}} \exp\left(-\frac{\sum_{i=d+3}^{D} w_i^2}{2}\right) dV}_{F_{\chi^2}\left(\frac{r^2}{\sigma^2}\right)}$$

$$= C \exp\left(-\frac{u_1^2 + u_2^2}{2\sigma^2}\right) F_{\chi^2}\left(\frac{r^2}{\sigma^2}\right)$$

where the third to last line was obtained by applying the change of variables $w_i = u_i / \sigma$ for $i \in \{m+3, \ldots, D\}$. Now we would like to integrate this expression over all remaining directions $z_3, \ldots, z_{m+2}$ and $u_1, u_2$. To obtain appropriate limits of integration, its helpful to explicitly write out an expression for relevant parts of the sets $B_\epsilon(x)$ and the local region of $\text{Tub}_\tau \mathcal{M}$ in terms of the defined coordinates. The sets can be described as

$$B_\epsilon(x) = \left\{ (u_1, u_2, z_3, \ldots, z_{m+2}, u_{m+3}, \ldots, u_D) \,\middle|\, (u_1 - d_v)^2 + (u_2 - d_x)^2 \right.$$

$$\left. + \sum_{i=3}^{m+2} z_i^2 + \sum_{i=d+3}^{D} u_i^2 \leq \epsilon^2 \right\} \quad (91)$$

and

$$S = \left\{ (u_1, u_2, z_3, \ldots, z_{m+2}, u_{m+3}, \ldots, u_D) \,\middle|\, \sum_{i \in \{1,2,d+3,\ldots,D\}} u_i^2 \leq \tau^2 \right\}$$

respectively, where $S$ describes the local region of the tubular neighborhood (when $\mathcal{M}$ is approximated as flat). Naturally, we are interested in integrating over the intersection of these two regions.

We will start with $z_{m+2}$. Observe that the limits for $z_{m+2}$ are necessarily symmetric - this follows from the fact that the point $(u_1, u_2, z_3, \ldots, z_{m+1}, 0, \ldots, 0)$ is equidistant along $z_{m+2}$ from the boundary of $B_\epsilon(x)$ in both the positive and negative directions. This distance is precisely

$$\sqrt{\epsilon^2 - (u_1 - d_v)^2 - (u_2 - d_x)^2 - \sum_{i=3}^{m+1} z_i^2}. \tag{92}$$

We do not concern ourselves with the boundary of $\text{Tub}_\tau \mathcal{M}$ in the tangential directions because we are employing a locally flat approximation. Now note that, for a point with coordinates $(u_1, u_2, z_3, \ldots, z_{m+1}, z_{m+2}, 0, \ldots, 0)$ the distance to the boundary of $B_\epsilon(x)$ in any direction spanned by $u_{m+3}, \ldots, u_D$ is a slight modification to 92

$$\sqrt{\epsilon^2 - (u_1 - d_v)^2 - (u_2 - d_x)^2 - \sum_{i=3}^{m+2} z_i^2}. \tag{93}$$

Observe that this gives us part of our expression for $r(u_1, u_2, z_3, \ldots, z_{m+2})$. In the normal directions, however, we are *also* concerned with the boundary of $\text{Tub}_\tau \mathcal{M}$, which is necessarily at a distance

$$\sqrt{\tau^2 - u_1^2 - u_2^2} \tag{94}$$

in any direction spanned by $u_{m+3}, \ldots, u_D$. Thus, the limits of integration for $z_{m+2}$ directions must range from $-1$ times 92 to $+1$ times 92. The radius in the argument of $F_{\chi^2}$ must then be the minimum of eq. (93) and eq. (94). Now integrating out $z_{m+2}$ yields

$$\mu\left[B_\epsilon(x) \cap S \Big|_{(u_1, u_2, z_3, \ldots, z_{m+1})}\right] = C \exp\left(-\frac{u_1^2 + u_2^2}{2\sigma^2}\right) \int_{-\sqrt{\epsilon^2 - (u_1 - d_v)^2 - (u_2 - d_x)^2 - \sum_{i=3}^{m+1} z_i^2}}^{\sqrt{\epsilon^2 - (u_1 - d_v)^2 - (u_2 - d_x)^2 - \sum_{i=3}^{m+1} z_i^2}}$$
$$F_{\chi^2}\left(\frac{\min\{\epsilon^2 - (u_1 - d_v)^2 - (u_2 - d_x)^2 - \sum_{i=3}^{m+2} z_i^2, \tau^2 - u_1^2 - u_2^2\}}{\sigma^2}\right) dz_{m+2} \tag{95}$$

where $B_\epsilon(x) \cap S \Big|_{(u_1, u_2, z_3, \ldots, z_{m+1})}$ denotes the subset of $B_\epsilon(x) \cap S$ when we fix all listed coordinates and allow the others to vary. Now we can apply the same argument to integrate out $z_{m+1}$. A point with coordinates $(u_1, u_2, z_3, \ldots, z_d, 0, 0, \ldots, 0)$ is at distance

$$\sqrt{\epsilon^2 - (u_1 - d_v)^2 - (u_2 - d_x)^2 - \sum_{i=3}^{m} z_i^2} \tag{96}$$

from the boundary of $B_\epsilon(x)$ in the $\pm z_{m+1}$ directions. Thus,

$$\mu\left[B_\epsilon(x) \cap S \Big|_{(u_1, u_2, z_3, \ldots, z_d)}\right] =$$
$$C \exp\left(-\frac{u_1^2 + u_2^2}{2\sigma^2}\right) \int_{-\sqrt{\epsilon^2 - (u_1 - d_v)^2 - (u_2 - d_x)^2 - \sum_{i=3}^{m} z_i^2}}^{\sqrt{\epsilon^2 - (u_1 - d_v)^2 - (u_2 - d_x)^2 - \sum_{i=3}^{m} z_i^2}} \int_{-\sqrt{\epsilon^2 - (u_1 - d_v)^2 - (u_2 - d_x)^2 - \sum_{i=3}^{m+1} z_i^2}}^{\sqrt{\epsilon^2 - (u_1 - d_v)^2 - (u_2 - d_x)^2 - \sum_{i=3}^{m+1} z_i^2}}$$
$$F_{\chi^2}\left(\frac{\min\{\epsilon^2 - (u_1 - d_v)^2 - (u_2 - d_x)^2 - \sum_{i=3}^{m+2} z_i^2, \tau^2 - u_1^2 - u_2^2\}}{\sigma^2}\right) dz_{m+2} dz_{m+1}. \tag{97}$$

Repeating this process for all remaining $z_i$'s yields

$$\mu\left[B_\epsilon(x) \cap S \Big|_{(u_1, u_2)}\right] = C \exp\left(-\frac{u_1^2 + u_2^2}{2\sigma^2}\right) \int_{-\sqrt{\epsilon^2 - (u_1 - d_v)^2 - (u_2 - d_x)^2}}^{\sqrt{\epsilon^2 - (u_1 - d_v)^2 - (u_2 - d_x)^2}}$$
$$\cdots \int_{-\sqrt{\epsilon^2 - (u_1 - d_v)^2 - (u_2 - d_x)^2 - \sum_{i=3}^{m} z_i^2}}^{\sqrt{\epsilon^2 - (u_1 - d_v)^2 - (u_2 - d_x)^2 - \sum_{i=3}^{m} z_i^2}} \int_{-\sqrt{\epsilon^2 - (u_1 - d_v)^2 - (u_2 - d_x)^2 - \sum_{i=3}^{m+1} z_i^2}}^{\sqrt{\epsilon^2 - (u_1 - d_v)^2 - (u_2 - d_x)^2 - \sum_{i=3}^{m+1} z_i^2}}$$
$$F_{\chi^2}\left(\frac{\min\{\epsilon^2 - (u_1 - d_v)^2 - (u_2 - d_x)^2 - \sum_{i=3}^{m+2} z_i^2, \tau^2 - u_1^2 - u_2^2\}}{\sigma^2}\right)$$
$$dz_{m+2} dz_{m+1} \ldots dz_3. \tag{98}$$

A point with coordinates $(u_1, 0, \ldots, 0)$ is at distance

$$\sqrt{\epsilon^2 - (u_1 - d_v)^2} - d_x \tag{99}$$

from the boundary of $B_\epsilon(x)$ in the $-u_2$ direction, while it is at distance

$$\sqrt{\epsilon^2 - (u_1 - d_v)^2} + d_x \tag{100}$$

in the $+u_2$ direction. This can be verified by inspecting Figure 17. Since $u_2$ spans a normal direction, we must also consider the boundary of $\text{Tub}_\tau \mathcal{M}$. Namely, the point $(u_1, 0, \ldots, 0)$ is at distance

$$\sqrt{\tau^2 - u_1^2} \tag{101}$$

from the boundary of $\text{Tub}_\tau \mathcal{M}$ in the $\pm u_2$ direction. Therefore,

$$
\mu\Big[B_\epsilon(x) \cap S\Big|_{(u_1)}\Big] =
$$
$$
C \int_{\max\left\{-\sqrt{\tau^2-u_1^2},\,-\sqrt{\epsilon^2-(u_1-d_v)^2}+d_x\right\}}^{\min\left\{\sqrt{\tau^2-u_1^2},\,\sqrt{\epsilon^2-(u_1-d_v)^2}+d_x\right\}} \exp\left(-\frac{u_1^2 + u_2^2}{2\sigma^2}\right) \int_{-\sqrt{\epsilon^2-(u_1-d_v)^2-(u_2-d_x)^2}}^{\sqrt{\epsilon^2-(u_1-d_v)^2-(u_2-d_x)^2}}
$$
$$
\cdots \int_{-\sqrt{\epsilon^2-(u_1-d_v)^2-(u_2-d_x)^2-\sum_{i=3}^m z_i^2}}^{\sqrt{\epsilon^2-(u_1-d_v)^2-(u_2-d_x)^2-\sum_{i=3}^m z_i^2}} \int_{-\sqrt{\epsilon^2-(u_1-d_v)^2-(u_2-d_x)^2-\sum_{i=3}^{m+1} z_i^2}}^{\sqrt{\epsilon^2-(u_1-d_v)^2-(u_2-d_x)^2-\sum_{i=3}^{m+1} z_i^2}}
$$
$$
F_{\chi^2}\left(\frac{\min\{\epsilon^2 - (u_1 - d_v)^2 - (u_2 - d_x)^2 - \sum_{i=3}^{m+2} z_i^2,\, \tau^2 - u_1^2 - u_2^2\}}{\sigma^2}\right)
$$
$$
dz_{m+2} dz_{m+1} \ldots dz_3 du_2. \tag{102}
$$

And finally since $2\tau < \epsilon$ we have,

$$
\mu\Big[B_\epsilon(x) \cap S\Big] =
$$
$$
C \int_{-\tau}^{\tau} \int_{\max\left\{-\sqrt{\tau^2-u_1^2},\,-\sqrt{\epsilon^2-(u_1-d_v)^2}+d_x\right\}}^{\min\left\{\sqrt{\tau^2-u_1^2},\,\sqrt{\epsilon^2-(u_1-d_v)^2}+d_x\right\}} \exp\left(-\frac{u_1^2 + u_2^2}{2\sigma^2}\right) \int_{-\sqrt{\epsilon^2-(u_1-d_v)^2-(u_2-d_x)^2}}^{\sqrt{\epsilon^2-(u_1-d_v)^2-(u_2-d_x)^2}}
$$
$$
\cdots \int_{-\sqrt{\epsilon^2-(u_1-d_v)^2-(u_2-d_x)^2-\sum_{i=3}^m z_i^2}}^{\sqrt{\epsilon^2-(u_1-d_v)^2-(u_2-d_x)^2-\sum_{i=3}^m z_i^2}} \int_{-\sqrt{\epsilon^2-(u_1-d_v)^2-(u_2-d_x)^2-\sum_{i=3}^{m+1} z_i^2}}^{\sqrt{\epsilon^2-(u_1-d_v)^2-(u_2-d_x)^2-\sum_{i=3}^{m+1} z_i^2}}
$$
$$
F_{\chi^2}\left(\frac{\min\{\epsilon^2 - (u_1 - d_v)^2 - (u_2 - d_x)^2 - \sum_{i=3}^{m+2} z_i^2,\, \tau^2 - u_1^2 - u_2^2\}}{\sigma^2}\right)
$$
$$
dz_{m+2} dz_{m+1} \ldots dz_3 du_2 du_1. \tag{103}
$$

To obtain an expression for $\mu[B_\epsilon(x)]$ we need one more component. Visual inspection of Figure 11 reveals that some of the measure of $B_\epsilon(x)$ can come from a region of the manifold that is at a far intrinsic distance, but small Euclidean distance. This is a phenomenon we have not yet accounted for. To do so, we will provide a large overestimate of the measure it contributes (which in the end, will not affect the conclusion of the proof). Since it is hard to quantify the volume of this region in the general case, we know its volume is necessarily bounded by $\text{Vol}(B_\epsilon(x))$. It follows that its measure is upper bounded by the maximum probability density in the region times $\text{Vol}(B_\epsilon(x))$.

To find the maximum probability density in this region, we need to reason about the minimum distance any point in this region can have to the manifold. Consider a point $p$ in this region, and its corresponding nearest point on $\mathcal{M}$, $\text{proj}_\mathcal{M} p$. Since we have stipulated $r_0(\mathcal{M}) > \epsilon$, any manifold geodesic path through $\text{proj}_\mathcal{M} x$ could not have left $B_\epsilon(x)$ and re-entered it without travelling more than $\pi r_0(\mathcal{M})$ distance. Our locally flat approximation ensures that our expression for $\mu[B_\epsilon(x)]$ accounts for the volume traced out by all geodesic paths that pass through $\text{proj}_\mathcal{M} x$ before they leave $B_\epsilon(x)$; thus the only measure we have not accounted for is associated with areas of $\mathcal{M}$ where a geodesic path through $\text{proj}_\mathcal{M} x$ left $B_\epsilon(x)$ and later returned.

Since we know the manifold distance from $\text{proj}_{\mathcal{M}} x$ to $\text{proj}_{\mathcal{M}} p$ exceeds $\pi r_0(\mathcal{M})$, we know $\| \text{proj}_{\mathcal{M}} x - \text{proj}_{\mathcal{M}} p \|_2 \geq s_0(\mathcal{M})$ from the definition of minimum branch separation. Using the triangle inequality and applying assumptions to bound $\epsilon$ and $\tau$ we can say

$$\| p - \text{proj}_{\mathcal{M}} p \|_2 \geq \| \text{proj}_{\mathcal{M}} x - \text{proj}_{\mathcal{M}} p \|_2 - \| x - \text{proj}_{\mathcal{M}} x \|_2 - \| x - p \|_2$$
$$> s_0(\mathcal{M}) - \tau - \sqrt{(s_0(\mathcal{M}) - \tau)^2 - \tau^2}$$
$$:= \delta$$
$$> 0.$$

Thus, the measure of this distant region of $B_\epsilon(x)$ is necessarily bounded above by

$$\text{Vol}\Big(B_\epsilon(x)\Big) \frac{1}{Z} \exp\Big(-\frac{\delta^2}{2\sigma^2}\Big).$$

Thus, we can safely conclude that

$$\mu\Big[B_\epsilon(x)\Big] \leq$$
$$C \int_{-\tau}^{\tau} \int_{\max\{-\sqrt{\tau^2-u_1^2}, -\sqrt{\epsilon^2-(u_1-d_v)^2}+d_x\}}^{\min\{\sqrt{\tau^2-u_1^2}, \sqrt{\epsilon^2-(u_1-d_v)^2}+d_x\}} \exp\Big(-\frac{u_1^2+u_2^2}{2\sigma^2}\Big) \int_{-\sqrt{\epsilon^2-(u_1-d_v)^2-(u_2-d_x)^2}}^{\sqrt{\epsilon^2-(u_1-d_v)^2-(u_2-d_x)^2}}$$
$$\cdots \int_{-\sqrt{\epsilon^2-(u_1-d_v)^2-(u_2-d_x)^2-\sum_{i=3}^m z_i^2}}^{\sqrt{\epsilon^2-(u_1-d_v)^2-(u_2-d_x)^2-\sum_{i=3}^m z_i^2}} \int_{-\sqrt{\epsilon^2-(u_1-d_v)^2-(u_2-d_x)^2-\sum_{i=3}^{m+1} z_i^2}}^{\sqrt{\epsilon^2-(u_1-d_v)^2-(u_2-d_x)^2-\sum_{i=3}^{m+1} z_i^2}}$$
$$F_{\chi^2}\left(\frac{\min\{\epsilon^2 - (u_1-d_v)^2 - (u_2-d_x)^2 - \sum_{i=3}^{m+2} z_i^2, \tau^2 - u_1^2 - u_2^2\}}{\sigma^2}\right)$$
$$dz_{m+2}dz_{m+1}\ldots dz_3 du_2 du_1 + \text{Vol}\Big(B_\epsilon(x)\Big) \frac{1}{Z} \exp\Big(-\frac{\delta^2}{2\sigma^2}\Big). \quad (104)$$

Computing $\mu[U_x]$ involves changing the lower limit of integration for $z_3$. In the $u_1$, $z_3$ plane the boundary of $U_x$ is simply a line with slope $\tan(\phi)$; the $u_1$ intercept can be obtained with simple geometry, and is $d_v - \sec(\phi)(\frac{\epsilon - \|x-y\|_2}{2})$. It follows that the expression for the boundary can be rearranged as,

$$z_3 = \cot(\phi)\bigg(u_1 - d_v + \sec(\phi)\Big(\frac{\epsilon - \|x-y\|_2}{2}\Big)\bigg).$$

To understand this step we encourage the reader to refer back to Figure 12 to recall the definition of $U_x$. Moving forward, for the sake of simplicity we can also discard the second term in eq. (104) to obtain a lower bound on $\mu[U_x]$. Therefore,

$$\mu\Big[U_x\Big] \leq$$
$$C \int_{-\tau}^{\tau} \int_{\max\{-\sqrt{\tau^2-u_1^2}, -\sqrt{\epsilon^2-(u_1-d_v)^2}+d_x\}}^{\min\{\sqrt{\tau^2-u_1^2}, \sqrt{\epsilon^2-(u_1-d_v)^2}+d_x\}} \exp\Big(-\frac{u_1^2+u_2^2}{2\sigma^2}\Big) \int_{\cot(\phi)\big(u_1-d_v+\sec(\phi)\big(\frac{\epsilon-\|x-y\|_2}{2}\big)\big)}^{\sqrt{\epsilon^2-(u_1-d_v)^2-(u_2-d_x)^2}}$$
$$\cdots \int_{-\sqrt{\epsilon^2-(u_1-d_v)^2-(u_2-d_x)^2-\sum_{i=3}^m z_i^2}}^{\sqrt{\epsilon^2-(u_1-d_v)^2-(u_2-d_x)^2-\sum_{i=3}^m z_i^2}} \int_{-\sqrt{\epsilon^2-(u_1-d_v)^2-(u_2-d_x)^2-\sum_{i=3}^{m+1} z_i^2}}^{\sqrt{\epsilon^2-(u_1-d_v)^2-(u_2-d_x)^2-\sum_{i=3}^{m+1} z_i^2}}$$
$$F_{\chi^2}\left(\frac{\min\{\epsilon^2 - (u_1-d_v)^2 - (u_2-d_x)^2 - \sum_{i=3}^{m+2} z_i^2, \tau^2 - u_1^2 - u_2^2\}}{\sigma^2}\right)$$
$$dz_{m+2}dz_{m+1}\ldots dz_3 du_2 du_1. \quad (105)$$

We are interested in evaluating the ratio of 105 and 104 in the limit of $\sigma \to 0^+$. Since 104 is the measure of a finite volume subset of $\mathbb{R}^D$ with a nonempty intersection with the support of $\rho$, for all $\sigma > 0$ the quantity is nonzero. To make sure the limits converge to a nonzero value, we will multiply

the top and bottom by $\frac{1}{2\pi\sigma^2 C}$. As we will also show, the ensures that the limit of the denominator converges to a nonzero value. Thus we can pull the limit into the fraction to say,

$$\lim_{\sigma\to 0^+} \frac{\frac{1}{2\pi\sigma^2 C}\mu[U_x]}{\frac{1}{2\pi\sigma^2 C}\mu[B_\epsilon(x)]} = \frac{\lim_{\sigma\to 0^+} \frac{1}{2\pi\sigma^2 C}\mu[U_x]}{\lim_{\sigma\to 0^+} \frac{1}{2\pi\sigma^2 C}\mu[B_\epsilon(x)]}. \tag{106}$$

The denominator can be bounded with

$$\frac{1}{2\pi\sigma^2 C}\mu\Big[B_\epsilon(x)\Big] \leq$$

$$\int_{-\tau}^{\tau}\int_{\max\left\{-\sqrt{\tau^2-u_1^2},-\sqrt{\epsilon^2-(u_1-d_v)^2}+d_x\right\}}^{\min\left\{\sqrt{\tau^2-u_1^2},\sqrt{\epsilon^2-(u_1-d_v)^2}+d_x\right\}} \frac{1}{2\pi\sigma^2}\exp\left(-\frac{u_1^2+u_2^2}{2\sigma^2}\right)\Bigg[\int_{-\sqrt{\epsilon^2-(u_1-d_v)^2-(u_2-d_x)^2}}^{\sqrt{\epsilon^2-(u_1-d_v)^2-(u_2-d_x)^2}}$$

$$\cdots \int_{-\sqrt{\epsilon^2-(u_1-d_v)^2-(u_2-d_x)^2-\sum_{i=3}^{m}z_i^2}}^{\sqrt{\epsilon^2-(u_1-d_v)^2-(u_2-d_x)^2-\sum_{i=3}^{m}z_i^2}} \int_{-\sqrt{\epsilon^2-(u_1-d_v)^2-(u_2-d_x)^2-\sum_{i=3}^{m+1}z_i^2}}^{\sqrt{\epsilon^2-(u_1-d_v)^2-(u_2-d_x)^2-\sum_{i=3}^{m+1}z_i^2}}$$

$$F_{\chi^2}\left(\frac{\min\{\epsilon^2-(u_1-d_v)^2-(u_2-d_x)^2-\sum_{i=3}^{m+2}z_i^2,\tau^2-u_1^2-u_2^2\}}{\sigma^2}\right)$$

$$dz_{m+2}dz_{m+1}\ldots dz_3\Bigg]du_2du_1 + \frac{1}{2\pi\sigma^2}\mathrm{Vol}\Big(B_\epsilon(x)\Big)\frac{1}{CZ}\exp\left(-\frac{\delta^2}{2\sigma^2}\right). \tag{107}$$

Defining everything in the bracket as $g(u_1,u_2)$ and taking the limit, we get

$$\lim_{\sigma\to 0^+}\frac{1}{2\pi\sigma^2 C}\mu\Big[B_\epsilon(x)\Big]$$

$$\leq \lim_{\sigma\to 0^+}\int_{-\tau}^{\tau}\int_{\max\left\{-\sqrt{\tau^2-u_1^2},-\sqrt{\epsilon^2-(u_1-d_v)^2}+d_x\right\}}^{\min\left\{\sqrt{\tau^2-u_1^2},\sqrt{\epsilon^2-(u_1-d_v)^2}+d_x\right\}}\frac{1}{2\pi\sigma^2}\exp\left(-\frac{u_1^2+u_2^2}{2\sigma^2}\right)g(u_1,u_2)du_2du_1$$

$$+\lim_{\sigma\to 0^+}\frac{1}{2\pi\sigma^2}\mathrm{Vol}\Big(B_\epsilon(x)\Big)\frac{1}{CZ}\exp\left(-\frac{\delta^2}{2\sigma^2}\right). \tag{108}$$

The second term can be rewritten as,

$$\lim_{\sigma\to 0^+}\frac{\mathrm{Vol}\big(B_\epsilon(x)\big)}{(2\pi)^{(D-m+2)/2}\cdot\sigma^{D-m}}\exp\left(-\frac{\delta^2}{2\sigma^2}\right).$$

Since $\delta$ is a constant, one can use repeated applications of L'Hopital's rule to show that this term converges to 0 in the limit. Now we can focus on the first term of eq. (108),

$$\lim_{\sigma\to 0^+}\frac{1}{2\pi\sigma^2 C}\mu\Big[B_\epsilon(x)\Big]$$

$$\leq \lim_{\sigma\to 0^+}\int_{-\tau}^{\tau}\int_{\max\left\{-\sqrt{\tau^2-u_1^2},-\sqrt{\epsilon^2-(u_1-d_v)^2}+d_x\right\}}^{\min\left\{\sqrt{\tau^2-u_1^2},\sqrt{\epsilon^2-(u_1-d_v)^2}+d_x\right\}}\frac{1}{2\pi\sigma^2}\exp\left(-\frac{u_1^2+u_2^2}{2\sigma^2}\right)g(u_1,u_2)du_2du_1. \tag{109}$$

Define the limits of integration for $u_2$ as $L_{\mathrm{low}}(u_1)$ and $L_{\mathrm{up}}(u_1)$ respectively. Now we will rewrite the equation above, replacing the limits of integration with indicators

$$\lim_{\sigma\to 0^+}\frac{1}{2\pi\sigma^2 C}\mu\Big[B_\epsilon(x)\Big] \leq \lim_{\sigma\to 0^+}\int_{-\infty}^{\infty}\int_{-\infty}^{\infty}\frac{1}{2\pi\sigma^2}\exp\left(-\frac{u_1^2+u_2^2}{2\sigma^2}\right)$$

$$\cdot \mathbb{1}\Big[u_2\in[L_{\mathrm{low}}(u_1),L_{\mathrm{up}}(u_1)],u_1\in[-\tau,\tau]\Big]g(u_1,u_2)du_2du_1. \tag{110}$$

Applying the change of variables $w_1=u_1/\sigma$ and $w_2=u_2/\sigma$ yields,

$$\lim_{\sigma\to 0^+}\frac{1}{2\pi\sigma^2 C}\mu\Big[B_\epsilon(x)\Big] \leq \lim_{\sigma\to 0^+}\int_{-\infty}^{\infty}\int_{-\infty}^{\infty}\frac{1}{2\pi}\exp\left(-\frac{w_1^2+w_2^2}{2}\right)$$

$$\cdot \mathbb{1}\Big[w_2\in\Big[\frac{L_{\mathrm{low}}(\sigma w_1)}{\sigma},\frac{L_{\mathrm{up}}(\sigma w_1)}{\sigma}\Big],w_1\in[-\tau/\sigma,\tau/\sigma]\Big]g(\sigma w_1,\sigma w_2)dw_2dw_1. \tag{111}$$

Observe that $g$ as defined is simply an integral of a function bounded by 1 over a region bounded by $B_\epsilon(x)$. Therefore, $g$ must be no larger than $\text{Vol}(B_\epsilon(x))$. Clearly the function

$$\frac{1}{2\pi} \exp\left(-\frac{w_1^2 + w_2^2}{2}\right) \text{Vol}(B_\epsilon(x))$$

dominates the integrand of $g$; furthermore, this function is integrable as the double integral over $w_1$ and $w_2$ evaluates to $\text{Vol}(B_\epsilon(x))$. Therefore we can use it as our dominating function to invoke the dominated convergence theorem, allowing us to pull the limit into the integral,

$$\lim_{\sigma \to 0^+} \frac{1}{2\pi\sigma^2 C} \mu\Big[B_\epsilon(x)\Big] \leq \int_{-\infty}^{\infty} \int_{-\infty}^{\infty} \frac{1}{2\pi} \exp\left(-\frac{w_1^2 + w_2^2}{2}\right)$$
$$\cdot \lim_{\sigma \to 0^+} \mathbb{1}\left[w_2 \in \left[\frac{L_{\text{low}}(\sigma w_1)}{\sigma}, \frac{L_{\text{up}}(\sigma w_1)}{\sigma}\right], w_1 \in [-\tau/\sigma, \tau/\sigma]\right] \lim_{\sigma \to 0^+} g(\sigma w_1, \sigma w_2) dw_2 dw_1.$$
$$(112)$$

Note that $L_{\text{low}}(0)$ is necessarily negative, as $\sqrt{\epsilon^2 - d_v^2} \geq \sqrt{\epsilon^2 - \tau^2} \geq \sqrt{4\tau^2 - \tau^2} = \tau\sqrt{3} > d_x$. Therefore, the indicator function in the integrand converges pointwise to 1. Writing out the second term,

$$\lim_{\sigma \to 0^+} g(\sigma w_1, \sigma w_2) = \lim_{\sigma \to 0^+} \int_{-\sqrt{\epsilon^2 - (\sigma w_1 - d_v)^2 - (\sigma w_2 - d_x)^2}}^{\sqrt{\epsilon^2 - (\sigma w_1 - d_v)^2 - (\sigma w_2 - d_x)^2}}$$
$$\cdots \int_{-\sqrt{\epsilon^2 - (\sigma w_1 - d_v)^2 - (\sigma w_2 - d_x)^2 - \sum_{i=3}^{m} z_i^2}}^{\sqrt{\epsilon^2 - (\sigma w_1 - d_v)^2 - (\sigma w_2 - d_x)^2 - \sum_{i=3}^{m} z_i^2}} \int_{-\sqrt{\epsilon^2 - (\sigma w_1 - d_v)^2 - (\sigma w_2 - d_x)^2 - \sum_{i=3}^{m+1} z_i^2}}^{\sqrt{\epsilon^2 - (\sigma w_1 - d_v)^2 - (\sigma w_2 - d_x)^2 - \sum_{i=3}^{m+1} z_i^2}}$$
$$F_{\chi^2}\left(\frac{\min\{\epsilon^2 - (\sigma w_1 - d_v)^2 - (\sigma w_2 - d_x)^2 - \sum_{i=3}^{m+2} z_i^2, \tau^2 - \sigma w_1^2 - \sigma w_2^2\}}{\sigma^2}\right)$$
$$dz_{m+2} dz_{m+1} \ldots dz_3. \quad (113)$$

We can pull all limits of integration into an indicator function, but for the sake of brevity we will not write it out. Note that this step is simply taken to illustrate the fact that we can invoke the dominated convergence theorem again. Observe then that for all $\sigma > 0$ such that the terms inside of the radicals are nonnegative, we are integrating a Chi-squared CDF (which is bounded above by 1) over $\mathbb{R}^m$. However, the aforementioned indicator function is always bounded by an indicator over the set $B_\epsilon^m(0)$. Thus, we can choose $\mathbb{1}[\mathbf{z}_{3:d+2} \in B_\epsilon^m(0)]$ as our integrable dominating function, allowing us to pull the limit into the integrals.

We can evaluate the indicator function in the limit, and pull the indicators back into the limits of integration. Also observe that the integrand (the CDF of the Chi-squared distribution) converges to 1 as its argment converges to $+\infty$. Thus,

$$\lim_{\sigma \to 0^+} g(\sigma w_1, \sigma w_2) = \int_{-\sqrt{\epsilon^2 - d_v^2 - d_x^2}}^{\sqrt{\epsilon^2 - d_v^2 - d_x^2}} \cdots \int_{-\sqrt{\epsilon^2 - d_v^2 - d_x^2 - \sum_{i=3}^{m} z_i^2}}^{\sqrt{\epsilon^2 - d_v^2 - d_x^2 - \sum_{i=3}^{m} z_i^2}}$$
$$\int_{-\sqrt{\epsilon^2 - d_v^2 - d_x^2 - \sum_{i=3}^{m+1} z_i^2}}^{\sqrt{\epsilon^2 - d_v^2 - d_x^2 - \sum_{i=3}^{m+1} z_i^2}} dz_{m+2} dz_{m+1} \ldots dz_3$$
$$(114)$$

$$= \text{Vol}\Big(B_R^m(0)\Big) \quad (115)$$

where $R = \sqrt{\epsilon^2 - \|x - \text{proj}_{\mathcal{M}} x\|_2^2}$ since $d_v^2 + d_x^2 = \|x - \text{proj}_{\mathcal{M}} x\|_2^2$. Note $B_R^m(0)$ denotes an $m$-dimensional Euclidean ball of radius $R$. Plugging this into eq. (112),

$$\lim_{\sigma \to 0^+} \frac{1}{2\pi\sigma^2 C} \mu\Big[B_\epsilon(x)\Big] \le \int_{-\infty}^{\infty} \int_{-\infty}^{\infty} \frac{1}{2\pi} \exp\Big(-\frac{w_1^2 + w_2^2}{2}\Big) \cdot \text{Vol}\Big(B_r^m(0)\Big) dw_2 dw_1 \qquad (116)$$

$$= \text{Vol}\Big(B_r^m(0)\Big) \int_{-\infty}^{\infty} \int_{-\infty}^{\infty} \frac{1}{2\pi} \exp\Big(-\frac{w_1^2 + w_2^2}{2}\Big) dw_2 dw_1 \qquad (117)$$

$$= \text{Vol}\Big(B_r^m(0)\Big). \qquad (118)$$

Now we want to evaluate $\frac{1}{2\pi\sigma^2 C} \mu[U_x]$ in the limit. Recall,

$$\frac{1}{2\pi\sigma^2 C} \mu\Big[U_x\Big] \ge \int_{-\tau}^{\tau} \int_{\max\{-\sqrt{\tau^2-u_1^2}, -\sqrt{\epsilon^2-(u_1-d_v)^2}+d_x\}}^{\min\{\sqrt{\tau^2-u_1^2}, \sqrt{\epsilon^2-(u_1-d_v)^2}+d_x\}} \frac{1}{2\pi\sigma^2} \exp\Big(-\frac{u_1^2 + u_2^2}{2\sigma^2}\Big)$$

$$\int_{\cot(\phi)\left(u_1-d_v+\sec(\phi)\left(\frac{\epsilon-\|x-y\|_2}{2}\right)\right)}^{\sqrt{\epsilon^2-(u_1-d_v)^2-(u_2-d_x)^2}} \left[ \cdots \int_{-\sqrt{\epsilon^2-(u_1-d_v)^2-(u_2-d_x)^2-\sum_{i=3}^m z_i^2}}^{\sqrt{\epsilon^2-(u_1-d_v)^2-(u_2-d_x)^2-\sum_{i=3}^m z_i^2}} \right.$$

$$\int_{-\sqrt{\epsilon^2-(u_1-d_v)^2-(u_2-d_x)^2-\sum_{i=3}^{m+1} z_i^2}}^{\sqrt{\epsilon^2-(u_1-d_v)^2-(u_2-d_x)^2-\sum_{i=3}^{m+1} z_i^2}}$$

$$F_{\chi^2}\left(\frac{\min\{\epsilon^2 - (u_1 - d_v)^2 - (u_2 - d_x)^2 - \sum_{i=3}^{m+2} z_i^2, \tau^2 - u_1^2 - u_2^2\}}{\sigma^2}\right)$$

$$\left. dz_{m+2} dz_{m+1} \cdots \right] dz_3 du_2 du_1. \qquad (119)$$

Define everything in the bracket to be $g'(u_1, u_2, z_3)$. Rewriting we have,

$$\frac{1}{2\pi\sigma^2 C} \mu\Big[U_x\Big] \ge \int_{-\tau}^{\tau} \int_{\max\{-\sqrt{\tau^2-u_1^2}, -\sqrt{\epsilon^2-(u_1-d_v)^2}+d_x\}}^{\min\{\sqrt{\tau^2-u_1^2}, \sqrt{\epsilon^2-(u_1-d_v)^2}+d_x\}} \frac{1}{2\pi\sigma^2} \exp\Big(-\frac{u_1^2 + u_2^2}{2\sigma^2}\Big)$$

$$\int_{\cot(\phi)\left(u_1-d_v+\sec(\phi)\left(\frac{\epsilon-\|x-y\|_2}{2}\right)\right)}^{\sqrt{\epsilon^2-(u_1-d_v)^2-(u_2-d_x)^2}} g'(u_1, u_2, z_3) dz_3 du_2 du_1. \qquad (120)$$

Taking the limit on both sides,

$$\lim_{\sigma \to 0^+} \frac{1}{2\pi\sigma^2 C} \mu\Big[U_x\Big] \ge$$

$$\lim_{\sigma \to 0^+} \int_{-\tau}^{\tau} \int_{\max\{-\sqrt{\tau^2-u_1^2}, -\sqrt{\epsilon^2-(u_1-d_v)^2}+d_x\}}^{\min\{\sqrt{\tau^2-u_1^2}, \sqrt{\epsilon^2-(u_1-d_v)^2}+d_x\}} \frac{1}{2\pi\sigma^2} \exp\Big(-\frac{u_1^2 + u_2^2}{2\sigma^2}\Big)$$

$$\cdot \int_{\cot(\phi)\left(u_1-d_v+\sec(\phi)\left(\frac{\epsilon-\|x-y\|_2}{2}\right)\right)}^{\sqrt{\epsilon^2-(u_1-d_v)^2-(u_2-d_x)^2}} g'(u_1, u_2, z_3) dz_3 du_2 du_1. \qquad (121)$$

We will use the same definitions for the limits of integration of $u_2$ from 110, and further define

$$L_{\text{low},z_3}(u_1, \phi) = \cot(\phi)\Big(u_1 - d_v + \sec(\phi)\Big(\frac{\epsilon - \|x - y\|_2}{2}\Big)\Big)$$

and

$$L_{\text{up},z_3}(u_1, u_2) = \sqrt{\epsilon^2 - (u_1 - d_v)^2 - (u_2 - d_x)^2}.$$

Converting the limits of integration to indicators,

$$\lim_{\sigma \to 0^+} \frac{1}{2\pi\sigma^2 C} \mu\Big[U_x\Big] \geq \lim_{\sigma \to 0^+} \int_{-\infty}^{\infty} \int_{-\infty}^{\infty} \int_{-\infty}^{\infty} \frac{1}{2\pi\sigma^2} \exp\Big(-\frac{u_1^2 + u_2^2}{2\sigma^2}\Big) g'(u_1, u_2, z_3)$$

$$\mathbb{1}\bigg[z_3 \in \Big[L_{\text{low},z_3}(u_1, \phi), L_{\text{up},z_3}(u_1, u_2)\Big],$$

$$u_2 \in \Big[L_{\text{low}}(u_1), L_{\text{up}}(u_1)\Big], u_1 \in \Big[-\tau, \tau\Big]\bigg]$$

$$dz_3 du_2 du_1. \quad (122)$$

Now let's apply the same change of variables $w_1 = u_1/\sigma$ and $w_2 = u_2/\sigma$ to get

$$\lim_{\sigma \to 0^+} \frac{1}{2\pi\sigma^2 C} \mu\Big[U_x\Big] \geq \lim_{\sigma \to 0^+} \int_{-\infty}^{\infty} \int_{-\infty}^{\infty} \int_{-\infty}^{\infty} \frac{1}{2\pi} \exp\Big(-\frac{w_1^2 + w_2^2}{2}\Big) g'(\sigma w_1, \sigma w_2, z_3)$$

$$\cdot \mathbb{1}\bigg[z_3 \in \Big[L_{\text{low},z_3}(\sigma w_1, \phi), L_{\text{up},z_3}(\sigma w_1, \sigma w_2)\Big],$$

$$w_2 \in \Big[\frac{L_{\text{low}}(\sigma w_1)}{\sigma}, \frac{L_{\text{up}}(\sigma w_1)}{\sigma}\Big], w_1 \in \Big[-\tau/\sigma, \tau/\sigma\Big]\bigg]$$

$$dz_3 dw_2 dw_1. \quad (123)$$

Through a similar argument from 112, we can choose the function

$$\frac{1}{2\pi} \exp\Big(-\frac{w_1^2 + w_2^2}{2}\Big) \text{Vol}(B_\epsilon^{m-1}(0))$$

as our dominating function for the integrand. Now we can pull the limit inside the integrals,

$$\lim_{\sigma \to 0^+} \frac{1}{2\pi\sigma^2 C} \mu\Big[U_x\Big] \geq \int_{-\infty}^{\infty} \int_{-\infty}^{\infty} \int_{-\infty}^{\infty} \frac{1}{2\pi} \exp\Big(-\frac{w_1^2 + w_2^2}{2}\Big) \lim_{\sigma \to 0^+} g'(\sigma w_1, \sigma w_2, z_3)$$

$$\lim_{\sigma \to 0^+} \mathbb{1}\bigg[z_3 \in \Big[L_{\text{low},z_3}(\sigma w_1, \phi), L_{\text{up},z_3}(\sigma w_1, \sigma w_2)\Big],$$

$$w_2 \in \Big[\frac{L_{\text{low}}(\sigma w_1)}{\sigma}, \frac{L_{\text{up}}(\sigma w_1)}{\sigma}\Big], w_1 \in \Big[-\tau/\sigma, \tau/\sigma\Big]\bigg]$$

$$dz_3 dw_2 dw_1. \quad (124)$$

Observe that the indicator converges to

$$\mathbb{1}\bigg[z_3 \in \Big[\cot(\phi)\Big(\sec(\phi)\Big(\frac{\epsilon - \|x - y\|_2}{2}\Big) - d_v\Big), \sqrt{\epsilon^2 - \|x - \text{proj}_{\mathcal{M}} x\|_2^2}\Big]\bigg].$$

Now we want to evaluate $g'$ in the limit,

$$\lim_{\sigma \to 0^+} g'(\sigma w_1, \sigma w_2, z_3) =$$

$$\lim_{\sigma \to 0^+} \int_{-\sqrt{\epsilon^2 - (\sigma w_1 - d_v)^2 - (\sigma w_2 - d_x)^2 - z_3^2}}^{\sqrt{\epsilon^2 - (\sigma w_1 - d_v)^2 - (\sigma w_2 - d_x)^2 - z_3^2}} \cdots \int_{-\sqrt{\epsilon^2 - (\sigma w_1 - d_v)^2 - (\sigma w_2 - d_x)^2 - \sum_{i=3}^{m+1} z_i^2}}^{\sqrt{\epsilon^2 - (\sigma w_1 - d_v)^2 - (\sigma w_2 - d_x)^2 - \sum_{i=3}^{m+1} z_i^2}}$$

$$F_{\chi^2}\Big(\frac{\min\{\epsilon^2 - (\sigma w_1 - d_v)^2 - (\sigma w_2 - d_x)^2 - \sum_{i=3}^{m+2} z_i^2, \tau^2 - \sigma w_1^2 - \sigma w_2^2\}}{\sigma^2}\Big)$$

$$dz_{m+2} \ldots dz_4. \quad (125)$$

Again we can pull all limits of integration into an indicator function, but for the sake of brevity we will not write it out. Note that this step is simply taken to illustrate the fact that we can invoke the dominated convergence theorem again. Observe then that for all $\sigma > 0$ such that the terms inside of the radicals are nonnegative, we are integrating a Chi-squared CDF (which is bounded above by 1) over $\mathbb{R}^{m-1}$. However, the aforementioned indicator function is always bounded by an indicator over the set $B_\epsilon^{m-1}(0)$. Thus, we can choose $\mathbb{1}[\mathbf{z}_{3:d+2} \in B_\epsilon^{m-1}(0)]$ as our dominating function, which is clearly integrable. This allows us to apply the dominated convergence theorem to pull the limit into the integral.

We can evaluate the indicator function in the limit, and pull the indicators back into the limits of integration. Also observe that the integrand (the CDF of the Chi-squared distribution) converges to 1 as its argment converges to $+\infty$. Thus,

$$\lim_{\sigma \to 0^+} g'(\sigma w_1, \sigma w_2, z_3) = \int_{-\sqrt{\epsilon^2 - d_v^2 - d_x^2 - z_3^2}}^{\sqrt{\epsilon^2 - d_v^2 - d_x^2 - z_3^2}} \cdots \int_{-\sqrt{\epsilon^2 - d_v^2 - d_x^2 - \sum_{i=3}^{m+1} z_i^2}}^{\sqrt{\epsilon^2 - d_v^2 - d_x^2 - \sum_{i=3}^{m+1} z_i^2}} dz_{m+2} \ldots dz_4 \quad (126)$$

$$= \mathrm{Vol}\big(B_{r(z_3)}^{m-1}(0)\big) \quad (127)$$

where

$$r(z_3) = \begin{cases} \sqrt{\epsilon^2 - \|x - \mathrm{proj}_{\mathcal{M}} x\|_2^2 - z_3^2} & |z_3| \leq \sqrt{\epsilon^2 - \|x - \mathrm{proj}_{\mathcal{M}} x\|_2^2} \\ 0 & \text{otherwise} \end{cases}. \quad (128)$$

Plugging into 124, we have

$$\lim_{\sigma \to 0^+} \frac{1}{2\pi\sigma^2 C} \mu\big[U_x\big] \geq \int_{-\infty}^{\infty} \int_{-\infty}^{\infty} \int_{-\infty}^{\infty} \frac{1}{2\pi} \exp\left(-\frac{w_1^2 + w_2^2}{2}\right) \mathrm{Vol}\big(B_{r(z_3)}^{m-1}(0)\big)$$

$$\mathbb{1}\left[z_3 \in \left[\cot(\phi)\left(\sec(\phi)\left(\frac{\epsilon - \|x - y\|_2}{2}\right) - d_v\right), \sqrt{\epsilon^2 - \|x - \mathrm{proj}_{\mathcal{M}} x\|_2^2}\right]\right]$$

$$dz_3 dw_2 dw_1. \quad (129)$$

Pulling the indicator back into the limits of integration and rearranging, we get

$$\lim_{\sigma \to 0^+} \frac{1}{2\pi\sigma^2 C} \mu\big[U_x\big] \geq \int_{-\infty}^{\infty} \int_{-\infty}^{\infty} \int_{\cot(\phi)\left(\sec(\phi)\left(\frac{\epsilon - \|x - y\|_2}{2}\right) - d_v\right)}^{\sqrt{\epsilon^2 - \|x - \mathrm{proj}_{\mathcal{M}} x\|_2^2}}$$

$$\frac{1}{2\pi} \exp\left(-\frac{w_1^2 + w_2^2}{2}\right) \mathrm{Vol}\big(B_{r(z_3)}^{m-1}(0)\big) dz_3 dw_2 dw_1$$

$$(130)$$

$$= \underbrace{\int_{-\infty}^{\infty} \int_{-\infty}^{\infty} \frac{1}{2\pi} \exp\left(-\frac{w_1^2 + w_2^2}{2}\right) dw_2 dw_1}_{=1}$$

$$\cdot \int_{\cot(\phi)\left(\sec(\phi)\left(\frac{\epsilon - \|x - y\|_2}{2}\right) - d_v\right)}^{\sqrt{\epsilon^2 - \|x - \mathrm{proj}_{\mathcal{M}} x\|_2^2}} \mathrm{Vol}\big(B_{r(z_3)}^{m-1}(0)\big) dz_3$$

$$(131)$$

$$= \int_{\cot(\phi)\left(\sec(\phi)\left(\frac{\epsilon - \|x - y\|_2}{2}\right) - d_v\right)}^{\sqrt{\epsilon^2 - \|x - \mathrm{proj}_{\mathcal{M}} x\|_2^2}} \mathrm{Vol}\big(B_{r(z_3)}^{m-1}(0)\big) dz_3. \quad (132)$$

Let

$$f(\phi) = \cot(\phi)\left(\sec(\phi)\left(\frac{\epsilon - \|x - y\|_2}{2}\right) - d_v\right)$$

and redefine $z_3$ to $z$. Now we can combine 132 and 118 to obtain the bound on 106, and thus bound 90:

$$\lim_{\sigma \to 0^+} \mathbb{P}[\, a \in U_x \,|\, a \in B_\epsilon(x) \,] \geq \frac{\int_{f(\phi)}^{\sqrt{\epsilon^2 - \|x - \text{proj}_{\mathcal{M}} x\|_2^2}} \text{Vol}(B_{r(z)}^{m-1}(0)) dz}{\text{Vol}(B_R^m(0))}. \tag{133}$$

where $R = \sqrt{\epsilon^2 - \|x - \text{proj}_{\mathcal{M}} x\|_2^2}$ and $r(z) = \sqrt{\epsilon^2 - \|x - \text{proj}_{\mathcal{M}} x\|_2^2 - z^2}$. $\qquad\square$

### A.4.3 PROPOSITIONS

While the lemmata represent a majority of the theoretical results of this work, the theorems that stitch them together require a few more propositions. We will start by showing that the minimum radius of curvature of geodesic paths through the tubular neighborhood of a manifold $\mathcal{M}$ have a minimum radius of curvature $r_0(\mathcal{M}) - \tau$, where $r_0(\mathcal{M})$ is the minimum radius of curvature of geodesics in $\mathcal{M}$.

> **Proposition 1.** *Suppose $\mathcal{M}$ is a compact submanifold of $\mathbb{R}^D$ without boundary and with a minimum radius of curvature $r_0(\mathcal{M})$. Then $\text{Tub}_\tau \mathcal{M}$ has a minimum radius of curvature of $r_0(\mathcal{M}) - \tau$.*

*Proof.* Note that the ambient curvature $k$ of a path $\gamma$ through $\mathbb{R}^D$ can be written as $k^2 = k_g^2 + k_n^2$, where $k_g$ denotes the geodesic curvature and $k_n$ denotes the normal curvature. Paths that are geodesic in some submanifold $\mathcal{M}$ necessarily have 0 geodesic curvature with respect to $\mathcal{M}$; thus $k = k_n$ for such curves.

Now consider a length-parameterized geodesic path $\gamma_{\text{Tub}} : [a, b] \to \text{Tub}_\tau \mathcal{M}$. Since $\gamma_{\text{Tub}}$ is a geodesic of $\text{Tub}_\tau \mathcal{M}$, we know that $k(t) = k_n(t)$ for all $t \in [a, b]$. Now let's consider two cases:

1. $\gamma_{\text{Tub}}(t)$ is at distance less than $\tau$ from $\mathcal{M}$. In this case, the $\gamma_{\text{Tub}}(t)$ does not intersect with the boundary of $\text{Tub}_\tau \mathcal{M}$; it follows that no directions normal to $\text{Tub}_\tau \mathcal{M}$ exist at this point, and the normal acceleration $k_n(t)$ must be 0. Therefore, $k(t) = 0$. This formalizes an intuitive concept - geodesics through $\text{Tub}_\tau \mathcal{M}$ are necessarily straight lines if they do not lie on the boundary of $\text{Tub}_\tau \mathcal{M}$.

2. $\gamma_{\text{Tub}}(t)$ is at distance exactly equal to $\tau$ from $\mathcal{M}$. In this case, there exists a single normal direction to $\text{Tub}_\tau \mathcal{M}$ at $\gamma_{\text{Tub}}(t)$. It is simply the direction from $\gamma_{\text{Tub}}(t)$ to the nearest point on $\mathcal{M}$ (up to sign). Therefore, the only situation in which $\gamma_{\text{Tub}}$ has nonzero curvature is when it lies exactly on the boundary of $\text{Tub}_\tau \mathcal{M}$; the curvature is simply the norm of $\ddot{\gamma}_{\text{Tub}}(t)$, which is some vector necessarily normal to the boundary of $\text{Tub}_\tau \mathcal{M}$ at $\gamma_{\text{Tub}}(t)$.

It follows that bounding the radius of curvature of geodesics through $\text{Tub}_\tau \mathcal{M}$ can be accomplished by considering geodesic paths restricted to the boundary of $\text{Tub}_\tau \mathcal{M}$, as all other geodesics through $\text{Tub}_\tau \mathcal{M}$ have no curvature (and thus infinite radius of curvature).

Now redefine $\gamma_{\text{Tub}}$ to be some geodesic segment that lies entirely on the surface of $\text{Tub}_\tau \mathcal{M}$ with nonzero curvature; suppose this segment starts at $\gamma_{\text{Tub}}(t_1)$ and ends at $\gamma_{\text{Tub}}(t_2)$. Choose $n$ intermediate and *overlapping* time intervals $\{I_i\}_{i=1}^n$, $I_i = [a_i^1, a_i^2]$ where $a_i^1 < a_i^2$, $a_{i+1}^1 < a_i^2 < a_{i+1}^2$ and $a_0^1 = t_1$, $a_n^2 = t_2$. Observe that $\bigcup_{i=1}^n I_i = [t_1, t_2]$, so the set of interval spans $[t_1, t_2]$. Note that we choose for these intervals to be overlapping so we can avoid the case where curvature at the endpoints of sub-intervals is unaccounted for.

Now we will analyze a single sub-segment of $\gamma_{\text{Tub}}$ defined by the image of $I_j = [a_j^1, a_j^2]$ under the map $\gamma_{\text{Tub}}$. We encourage the reader to refer to Figure 18 for the remainder of the proof. As previously established, $\ddot{\gamma}_{\text{Tub}}(t)$ is necessarily parallel to the vector from $\text{proj}_{\mathcal{M}} \gamma_{\text{Tub}}(t)$ to $\gamma_{\text{Tub}}(t)$. Consider now the plane spanned by the vectors $\ddot{\gamma}_{\text{Tub}}(t)$ and $\dot{\gamma}_{\text{Tub}}(t)$. Now consider the osculating circle $O$ approximation of $\gamma_{\text{Tub}}$ near $\gamma_{\text{Tub}}(t)$ for some $t \in I_j$. Since $I_j$ can be made arbitrarily small, $O$ is an arbitrarily good approximation of $\gamma_{\text{Tub}}(t)$ for $t \in I_j$. Suppose $O$ has some radius $r$.

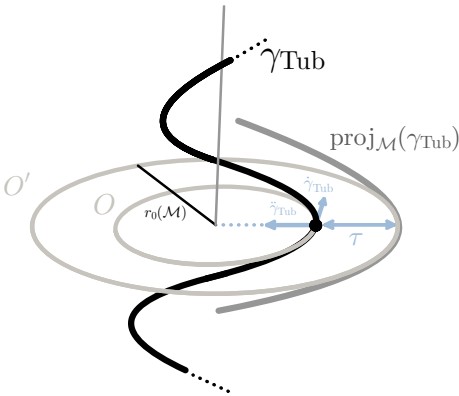

Figure 18: A diagram depicting $\gamma_{\text{Tub}}$, $\text{proj}_{\mathcal{M}} \gamma_{\text{Tub}}$, $O$ and $O'$.

Finally, consider the path in $\mathcal{M}$ defined by the image of $\gamma_{\text{Tub}}(t)$ for $t \in I_j$ under the map $\text{proj}_{\mathcal{M}} \gamma_{\text{Tub}}(t)$; colloquially this path is the projection of the subsegment of $\gamma_{\text{Tub}}$ onto $\mathcal{M}$. Given that the osculating plane contains the vector $\ddot{\gamma}_{\text{Tub}}(t)$ locally (for $t \in I_j$), and we have already established that $\ddot{\gamma}_{\text{Tub}}(t)$ is parallel to the residual of the projection $(\text{proj}_{\mathcal{M}} \gamma_{\text{Tub}}(t) - \gamma_{\text{Tub}}(t))$ then this projected path must also lie in the plane spanned by the vectors $\ddot{\gamma}_{\text{Tub}}(t)$ and $\dot{\gamma}_{\text{Tub}}(t)$ for $t \in I_j$. Since we are considering the case where $\gamma_{\text{Tub}}(t)$ is at distance exactly equal to $\tau$ from $\mathcal{M}$ for all $t$, this projected path is then approximated with its own osculating circle $O'$ lying in this plane with radius $r + \tau$, sharing the same center as $O$.

Since the projected path is approximated by $O'$, its acceleration vector in a neighborhood near $t$ must be parallel to $\ddot{\gamma}_{\text{Tub}}(t)$; this implies that it is entirely orthogonal to $\mathcal{M}$, making the path geodesic in $\mathcal{M}$. Since it is geodesic in $\mathcal{M}$, the radius of $O'$ for all segments of all possible geodesics $\gamma_{\text{Tub}}$ is no smaller than $r_0(\mathcal{M})$. Solving for the minimum possible radius of $O$ yields $r_0(\mathcal{M}) - \tau$.

We will quickly show that there always exists a geodesic segment in the tubular neighborhood such that $O'$ as defined above *achieves* this minimum radius $r_0(\mathcal{M})$. Pick some geodesic segment in $\mathcal{M}$ called $\gamma_{\mathcal{M}}$ with a constant nonzero acceleration $\|\ddot{\gamma}_{\mathcal{M}}\|_2 = 1/r_0(\mathcal{M})$. Note that $\gamma_{\mathcal{M}}$ can be made infinitesimally small such that the constant acceleration assumption is arbitrarily accurate. Also note that $\ddot{\gamma}_{\mathcal{M}}$ is necessarily oriented *normal* to $\mathcal{M}$, as $\gamma_{\mathcal{M}}$ is a geodesic of $\mathcal{M}$. Define another segment as follows,

$$\gamma_{\text{Tub}}(t) = \gamma_{\mathcal{M}}(t) + \tau \frac{\ddot{\gamma}_{\mathcal{M}}(t)}{\|\ddot{\gamma}_{\mathcal{M}}(t)\|_2} \tag{134}$$

where the domain of $\gamma_{\text{Tub}}(t)$ is the domain of $\gamma_{\mathcal{M}}(t)$. Observe that this segment necessarily lies on the boundary of the tubular neighborhood as it was displaced exactly $\tau$ from $\mathcal{M}$ in the direction $\ddot{\gamma}_{\mathcal{M}}(t)$, which lies normal to $\mathcal{M}$. Also observe that we can approximate $\gamma_{\mathcal{M}}(t)$ with its osculating circle $O'$ arbitrarily well, where $O'$ has a radius $r_0(\mathcal{M})$. Since $\ddot{\gamma}_{\mathcal{M}}(t)$ and $\gamma_{\mathcal{M}}(t)$ lie in the osculating plane, $\gamma_{\mathcal{M}}(t)$ must as well (as we have defined it to be a linear combination of these two vectors). In fact, $\gamma_{\text{Tub}}(t)$ is approximated arbitrarily well by $O$, its osculating circle centered at the center of $O'$. This very much mirrors the scenario that we posed earlier in the proof, visualized in Figure 18.

Since $O$ and $O'$ lie in the same plane centered about the same point, $\ddot{\gamma}_{\mathcal{M}}(t)$ and $\ddot{\gamma}_{\text{Tub}}(t)$ must be parallel. Since $\ddot{\gamma}_{\mathcal{M}}(t)$ lies normal to $\mathcal{M}$, $\ddot{\gamma}_{\text{Tub}}(t)$ must as well. By construction of $\gamma_{\text{Tub}}(t)$, $\ddot{\gamma}_{\mathcal{M}}(t)$ must be parallel to the residual of the projection of $\gamma_{\text{Tub}}(t)$ onto $\mathcal{M}$; this follows from the uniqueness of $\text{proj}_{\mathcal{M}}$ stemming from the assumptions that $\tau < r_0(\mathcal{M})$ and $2\tau < s_0(\mathcal{M})$ (detailed in assumptions 2 and 1). This residual is the unique direction normal to the tubular neighborhood at $\gamma_{\text{Tub}}(t)$, and we have just established that $\ddot{\gamma}_{\mathcal{M}}(t)$ (and thus $\ddot{\gamma}_{\text{Tub}}(t)$) is parallel to it. It follows that $\gamma_{\text{Tub}}(t)$ is a geodesic segment of $\text{Tub}_\tau \mathcal{M}$, as its acceleration vector lies completely normal to $\text{Tub}_\tau \mathcal{M}$. Its radius of curvature, $r_0(\mathcal{M}) - \tau$, can be deduced from the difference in radii of $O$ and $O'$.

We can conclude that $r$, the radius of curvature of any geodesic path in the tubular neighborhood, always achieves a minimum at $r = r_0(\mathcal{M}) - \tau$.

$\square$

Now we will move on to showing that the ORC (as we have defined it in Section 2.3) of any edge in an unweighted graph is necessarily restricted to the finite interval $[-2, 1]$.

**Proposition 2.** *For any edge $(x, y)$ in an unweighted graph, $-2 \leq \kappa(x, y) \leq 1$.*

*Proof.* Observe that, under the current construction, the unweighted graph distance between a node $a \in \mathcal{N}(x)$ and $b \in \mathcal{N}(y)$ is bounded above and below,

$$0 \leq d_G(a, b) \leq 3.$$

This implies the same bound on the Wasserstein distance between the measures $\mu_x$ and $\mu_y$. This is clear when we rewrite the 1-Wasserstein distance as follows,

$$W(\mu_x, \mu_y) = \inf_{\gamma \in \Pi(\mu_x, \mu_y)} \mathbb{E}_{a,b \sim \gamma}[d_G(a, b)]$$

where $\Pi(\mu_x, \mu_y)$ denotes the set of all measures on $V \times V$ with marginals $\mu_x$ and $\mu_y$. Thus,

$$0 \leq W(\mu_x, \mu_y) \leq 3\,,$$

and because $\kappa(x, y) = 1 - W(\mu_x, \mu_y)/1$, we have

$$-2 \leq \kappa(x, y) \leq 1.$$

$\square$

Now we will prove a bound on geodesic distances through the tubular neighborhood as a function of manifold geodesic distances. This proposition is used in Theorem 3.3 to bound graph distances of shortcut edges in ORC thresholded graphs.

**Proposition 3.** *Let $d_{Tub}(x, y)$ be the geodesic distance induced by the Euclidean metric from $x \in Tub_\tau(\mathcal{M})$ to $y \in Tub_\tau(\mathcal{M})$. Then,*

$$\frac{d_{Tub}(x, y)}{d_{\mathcal{M}}(\mathrm{proj}_{\mathcal{M}} x, \mathrm{proj}_{\mathcal{M}} y)} \geq \frac{r_0(\mathcal{M}) - \tau}{r_0(\mathcal{M})}.$$

*Proof.* Let $\gamma_{\text{Tub}}$ be the geodesic path through $\text{Tub}_\tau \mathcal{M}$ from $x$ to $y$. Select $P$ points $\{p_i\}_{i=1}^{P}$ (a method for which will be described soon) along $\gamma_{\text{Tub}}$ and join the projection of successive points $p_i$, $p_{i+1}$ onto $\mathcal{M}$ by a segment $\gamma_{\text{proj}}^{i,i+1}$ with curvature $r_0(\mathcal{M})$. Let's denote the segment of $\gamma_{\text{Tub}}$ between $p_i$, $p_{i+1}$ as $\gamma_{\text{Tub}}^{i,i+1}$. Also suppose $P$ is large enough such that the geodesic segment through $\mathcal{M}$ connecting the projections of $p_i$ and $p_{i+1}$ can be arbitrarily well approximated by its osculating circle $O_{i,i+1}$. Observe that since $\mathcal{M}$ is smooth and we have required $\tau < s_0(\mathcal{M})/2$ and $\tau < r_0(\mathcal{M})$ in assumptions 1, we have a unique projection. It follows that $\mathrm{proj}_{\mathcal{M}}(x) \; \forall x \in \text{Tub}_\tau \mathcal{M}$ is continuous.

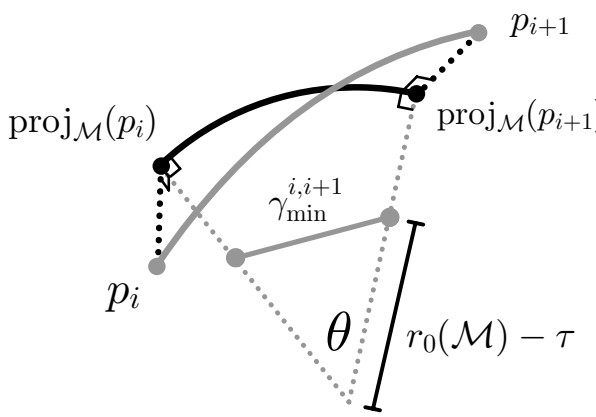

Figure 19: A diagram depicting $p_i$, $p_{i+1}$ and various related quantities.

Note that

$$d_{\text{Tub}}(x, y) = L(\gamma_{\text{Tub}}) = \sum_{i=1}^{P-1} L(\gamma_{\text{Tub}}^{i,i+1})$$

and

$$d_{\mathcal{M}}(\text{proj}_{\mathcal{M}} x, \text{proj}_{\mathcal{M}} y) \leq \sum_{i=1}^{P-1} d_{\mathcal{M}}(\text{proj}_{\mathcal{M}} p_i, \text{proj}_{\mathcal{M}} p_{i+1}) \leq \sum_{i=1}^{P-1} L(\gamma_{\text{proj}}^{i,i+1}).$$

The second inequality in the second statement requires explanation; observe that, since $\text{proj}_{\mathcal{M}}(\cdot)$ is continuous one can choose a $T$ such that for all $\Delta t < T$

$$\| \text{proj}_{\mathcal{M}} \gamma_{\text{Tub}}(t + \Delta t) - \text{proj}_{\mathcal{M}} \gamma_{\text{Tub}}(t) \|_2 < s_0(\mathcal{M})$$

for all $t \in \text{dom}(\gamma_{\text{Tub}})$. This gives us a principled way of choosing $\{p_i\}_{i=1}^P$, where $p_i = \gamma_{\text{Tub}}(t_i)$, $t_i = \sum_{j=1}^{i-1}(\Delta t)_j$. From the definition of minimum branch separation, we know that $\| \text{proj}_{\mathcal{M}} p_{i+1} - \text{proj}_{\mathcal{M}} p_i \|_2 < s_0(\mathcal{M}) \implies d_{\mathcal{M}}(\text{proj}_{\mathcal{M}} p_i, \text{proj}_{\mathcal{M}} p_{i+1}) \leq \pi r_0(\mathcal{M})$. Now we will apply Lemma 3 from Bernstein et al. (2001) to say

$$d_{\mathcal{M}}(\text{proj}_{\mathcal{M}} p_i, \text{proj}_{\mathcal{M}} p_{i+1}) \leq 2 r_0(\mathcal{M}) \arcsin \left( \frac{\| \text{proj}_{\mathcal{M}} p_{i+1} - \text{proj}_{\mathcal{M}} p_i \|_2}{2 r_0(\mathcal{M})} \right) \qquad (135)$$

$$= L(\gamma_{\text{proj}}^{i,i+1}). \qquad (136)$$

The second equality follows from the way we have defined $L(\gamma_{\text{proj}}^{i,i+1})$: an arc with a constant radius of curvature $r_0(\mathcal{M})$ from $\text{proj}_{\mathcal{M}} p_i$ to $\text{proj}_{\mathcal{M}} p_{i+1}$. Application of some trigonometry yields the arc length.

Now consider a single segment defined by $p_i$ and $p_{i+1}$. Since we assume $\gamma_{\text{proj}}^{i,i+1}$ has curvature $r_0(\mathcal{M})$, it is approximated exactly by its osculating circle with radius $r_0(\mathcal{M})$. Let $\gamma_{\text{min}}^{i,i+1}$ be the straight-line path between the points at distance $r_0(\mathcal{M}) - \tau$ from the center of this osculating circle in the directions of $\text{proj}_{\mathcal{M}} p_i$ and $\text{proj}_{\mathcal{M}} p_{i+1}$ respectively. A diagram is shown in Figure 19 for clarity.

Observe that the length $l_{\text{min}}^{i,i+1} = L(\gamma_{\text{min}}^{i,i+1})$ must be no larger than the length of any straight-line path connecting any $a \in \text{Tub}_\tau \mathcal{M}$ and $b \in \text{Tub}_\tau \mathcal{M}$ such that they project to $\text{proj}_{\mathcal{M}} p_i$ and $\text{proj}_{\mathcal{M}} p_{i+1}$ respectively. In understanding this, it helps to note that the set of points $a$ could lie in is a $D - m$ (where $m$ is the dimension of $\mathcal{M}$) dimensional space created by intersecting the tubular neighborhood with the hyperplane that contains $p_i$ and the center of the osculating circle $O_{i,i+1}$ (defined earlier in the proof) and spans all manifold normal directions at $p_i$. The set of points $b$ could lie in has a similar form; just replace $p_i$ with $p_{i+1}$. The only directions from $p_i$ and $p_{i+1}$ (restricted to the described subspaces) that bring the points closer is the direction towards the center of the osculating circle $O_{i,i+1}$. Starting at $p_i$ and $p_{i+1}$ and travelling in these respective directions until you hit the boundary of the tubular neighborhood will minimize the distance. This minimum distance bounds $L(\gamma_{\text{Tub}}^{i,i+1})$, and we can compute it as follows,

$$L(\gamma_{\text{Tub}}^{i,i+1}) \geq \| a - b \|_2 \qquad (137)$$

$$= \frac{r - \tau}{r} \| \text{proj}_{\mathcal{M}} p_i - \text{proj}_{\mathcal{M}} p_{i+1} \|_2 \qquad (138)$$

where $r$ is the radius of $O_{i,i+1}$. Since $(x - a)/x$ is increasing for positive $x$ and $a$, we have

$$L(\gamma_{\text{Tub}}^{i,i+1}) \geq \frac{r_0(\mathcal{M}) - \tau}{r_0(\mathcal{M})} \| \text{proj}_{\mathcal{M}} p_i - \text{proj}_{\mathcal{M}} p_{i+1} \|_2 \qquad (139)$$

$$= l_{\text{min}}^{i,i+1}. \qquad (140)$$

since $r \leq r_0(\mathcal{M})$. Thus, we know that $L(\gamma_{\text{Tub}}^{i,i+1}) \geq l_{\text{min}}^{i,i+1}$. Also note that $l_{\text{min}}^{i,i+1}$ can be written as a function of $L(\gamma_{\text{proj}}^{i,i+1})$ as follows: compute $\theta$, the angle swept out by $\gamma_{\text{proj}}^{i,i+1}$, $\theta = L(\gamma_{\text{proj}}^{i,i+1})/r_0(\mathcal{M})$. Then observe that $l_{\text{min}}^{i,i+1} = 2(r_0(\mathcal{M}) - \tau)\sin(\theta/2) = 2(r_0(\mathcal{M}) - \tau)\sin(L(\gamma_{\text{proj}}^{i,i+1})/2 r_0(\mathcal{M}))$.

Finally, note that $\sin(x) \approx x$ for small $x$ which is a reasonable approximation for large $P$. Note that $P$ can be made arbitrarily large as segments can be divided in half without shattering the requirement that $\|\text{proj}_{\mathcal{M}} p_{i+1} - \text{proj}_{\mathcal{M}} p_i\|_2 < s_0(\mathcal{M})$. Thus, $l_{\min}^{i,i+1} \approx 2(r_0(\mathcal{M}) - \tau)L(\gamma_{\text{proj}}^{i,i+1})/2r_0(\mathcal{M}) = \frac{r_0(\mathcal{M}) - \tau}{r_0(\mathcal{M})}L(\gamma_{\text{proj}}^{i,i+1})$. Putting it all together,

$$
\begin{aligned}
d_{\text{Tub}}(x, y) &= \sum_{i=1}^{P-1} L(\gamma_{\text{Tub}}^{i,i+1}) \\
&\geq \sum_{i=1}^{P-1} l_{\min}^{i,i+1} \\
&\approx \frac{r_0(\mathcal{M}) - \tau}{r_0(\mathcal{M})} \sum_{i=1}^{P-1} L(\gamma_{\text{proj}}^{i,i+1}) \\
&\geq \frac{r_0(\mathcal{M}) - \tau}{r_0(\mathcal{M})} \sum_{i=1}^{P-1} d_{\mathcal{M}}(\text{proj}_{\mathcal{M}} p_i, \text{proj}_{\mathcal{M}} p_{i+1}) \\
&\geq \frac{r_0(\mathcal{M}) - \tau}{r_0(\mathcal{M})} d_{\mathcal{M}}(\text{proj}_{\mathcal{M}} x, \text{proj}_{\mathcal{M}} y).
\end{aligned}
$$

$\square$

Now we will bound the measure of an arbitrary $D$-dimensional ball centered at a point in the tubular neighborhood, a result which will be used in Theorem 3.2.

**Proposition 4.** *Suppose $\rho$ is the probability density function defined in 1 for a manifold $\mathcal{M}$ embedded in $\mathbb{R}^D$. Then for $r_0(\mathcal{M}) \gg \delta$*

$$
\mathbb{P}_{z \sim \rho}\Big[\|z - x\|_2 \leq \delta\Big] \geq \frac{\delta^D \text{Vol}(B_1(0))}{2Z} \cdot e^{-\frac{\tau^2}{2\sigma^2}}. \tag{141}
$$

*Proof.* We can rewrite the left-hand side of 141 as

$$
\begin{aligned}
\mathbb{P}_{z \sim \rho}\Big[\|z - x\|_2 \leq \delta\Big] &= \mu\Big[\{z \mid \|z - x\|_2 \leq \delta\}\Big] \\
&\geq \frac{1}{2}\text{Vol}(B_\delta(0)) \cdot \min_{\|z-x\|_2 \leq \delta,\, z \in \text{Tub}_\tau \mathcal{M}} \rho(z) \\
&= \frac{1}{2}\delta^D \text{Vol}(B_1(0)) \cdot \min_{\|z-x\|_2 \leq \delta,\, z \in \text{Tub}_\tau \mathcal{M}} \rho(z)
\end{aligned}
$$

The second line follows from the fact that a $\delta$-ball centered exactly on the surface of the tubular neighborhood $\text{Tub}_\tau \mathcal{M}$ is approximately cut it half; while curvature of $\mathcal{M}$ may slightly reduce this volume, considering $r_0(\mathcal{M}) \gg \delta$ makes this a reasonable approximation. Now it remains to bound the minimum probability density. We can do this by simply choosing the minimum value $\rho$ takes on in the tubular neighborhood,

$$
\min_{z \in \text{Tub}_\tau \mathcal{M}} \rho(z) = \frac{1}{Z}e^{-\frac{\tau^2}{2\sigma^2}} \tag{142}
$$

Combining everything,

$$
\mathbb{P}_{z \sim \rho}\Big[\|z - x\|_2 \leq \delta\Big] \geq \frac{\delta^D \text{Vol}(B_1(0))}{2Z} \cdot e^{-\frac{\tau^2}{2\sigma^2}}. \tag{143}
$$

$\square$

Finally, we will show that the probability that the neighborhoods of two points overlap completly approaches 1 as the points converge to each other. This proposition is also used in Theorem 3.2.

**Proposition 5.** *Suppose $\{x_i\}_{i=1}^{\infty}$ and $\{y_i\}_{i=1}^{\infty}$ are sequences of points in a series of point clouds sampled i.i.d. according to the probability density function $\rho$ defined by 1. Also suppose that $\lim_{i \to \infty} \|x_i - y_i\|_2 = 0$. Then*

$$\lim_{i \to \infty} \mathbb{P}_{a \sim \rho}\Big[a \in B_\epsilon(x_i) \cap B_\epsilon(y_i) \,\Big|\, a \in B_\epsilon(x_i) \cup B_\epsilon(y_i)\Big] = 1.$$

*Proof.* Our proof will use a similar argument to that of Lemma A.4. First we will rearrange to put the term of interest in a friendlier form,

$$\mathbb{P}_{a \sim \rho}\Big[a \in B_\epsilon(x) \cap B_\epsilon(y) \,\Big|\, a \in B_\epsilon(x) \cup B_\epsilon(y)\Big] \tag{144}$$

$$= \frac{\mu\Big(B_\epsilon(x) \cap B_\epsilon(y)\Big)}{\mu\Big(B_\epsilon(x) \cup B_\epsilon(y)\Big)} \tag{145}$$

$$= \frac{\mu\Big(B_\epsilon(x) \cap B_\epsilon(y)\Big)}{\mu\Big(B_\epsilon(x)\Big) + \mu\Big(B_\epsilon(y)\Big) - \mu\Big(B_\epsilon(x) \cap B_\epsilon(y)\Big)}. \tag{146}$$

Computing each of these terms involves integrating $\rho$ (defined by 1) over the set of interest. We define $S_x = B_\epsilon(x)$, $S_y = B_\epsilon(y)$ and $S_{x,y} = B_\epsilon(x) \cap B_\epsilon(y)$. Evaluating the measures, we have

$$\mu\Big(B_\epsilon(x)\Big) = \int_{z \in S_x} \rho(z) dV \tag{147}$$

$$= \int_{\mathbb{R}^D} \rho(z) \cdot \chi_{S_x}(z) dV \tag{148}$$

and

$$\mu\Big(B_\epsilon(y)\Big) = \int_{z \in S_y} \rho(z) dV \tag{149}$$

$$= \int_{\mathbb{R}^D} \rho(z) \cdot \chi_{S_y}(z) dV \tag{150}$$

where $\chi_A(z)$ is an indicator function, defined as

$$\chi_A(z) = \begin{cases} 1 & \text{if } z \in A \\ 0 & \text{if } z \notin A \end{cases}.$$

The measure of the intersection of the two epsilon balls can also be described with

$$\mu\Big(B_\epsilon(x) \cap B_\epsilon(y)\Big) = \int_{z \in S_{x,y}} \rho(z) dV \tag{151}$$

$$= \int_{\mathbb{R}^D} \rho(z) \cdot \chi_{S_{x,y}}(z) dV. \tag{152}$$

Now suppose we have two sequences, $\{x_i\}_{i=1}^{\infty}$ and $\{y_i\}_{i=1}^{\infty}$ where $\lim_{i \to \infty} x_i = x$, $\lim_{i \to \infty} y_i = y$, and $\lim_{i \to \infty} \|x_i - y_i\|_2 = 0$. Now define $S_{x,i} = B_\epsilon(x_i)$, $S_{y,i} = B_\epsilon(y_i)$ and $S_i = B_\epsilon(x_i) \cap B_\epsilon(y_i)$.

We will show that $\lim_{i \to \infty} \chi_{S_{x,i}}(z) = \lim_{i \to \infty} \chi_{S_{y,i}}(z)$ pointwise, from which it will follow that $\chi_{S_i}$ will converge to the same function pointwise. Define $\chi(z) := \lim_{i \to \infty} \chi_{S_{y,i}}(z)$. Now we'll show that $\lim_{i \to \infty} \chi_{S_{x,i}}(z) = \chi(z)$. This involves showing that for all $a \in \mathbb{R}^D$ we have $\lim_{i \to \infty} \chi_{S_{x,i}}(a) = \chi(a)$. Consider two cases:

- $\chi(a) = 1$. Since $x_i \to x$ and $\|x_i - y_i\|_2 \to 0$, we have $x_i \to y$. Since $\chi(a) = 1$, we know that for sufficiently large $i$, $\|a - y_i\|_2 \leq \epsilon$, so it follows that $\|a - y\|_2 \leq \epsilon$. Since $x_i \to y$, it must be true that $\|a - x_i\|_2 \leq \epsilon$ for sufficiently large $i$. Thus for sufficiently large $i$, $\chi_{S_{x,i}}(a) = \chi(a) = 1$.

- $\chi(a) = 0$. This implies $\|a - y_i\|_2 > \epsilon$ for sufficiently large $i$. Since $\|x_i - y_i\|_2 \to 0$ we must also have that $\|a - x_i\|_2 > \epsilon$ for sufficiently large $i$ as well. Thus for sufficiently large $i$ we have $\chi_{S_{x,i}}(a) = \chi(a) = 0$.

Therefore $\lim_{i \to \infty} \chi_{S_{x,i}}(z) = \lim_{i \to \infty} \chi_{S_{y,i}}(z) = \chi(z)$. It then follows that the indicator function on intersection of the two epsilon balls (denoted $S_{x_i, y_i}$) must also converge pointwise to $\chi(z)$. Namely, $\lim_{i \to \infty} \chi_{S_{x_i, y_i}}(z) = \chi(z)$.

Now we want to use these results to evaluate 146 in the limit. As we will show, the denominator converges to a nonzero value, and thus the limit can be taken into the numerator and denominator as follows,

$$\lim_{i \to \infty} \mathbb{P}_{a \sim \rho}\Big[a \in B_\epsilon(x_i) \cap B_\epsilon(y_i) \,\Big|\, a \in B_\epsilon(x_i) \cup B_\epsilon(y_i)\Big] \tag{153}$$

$$= \lim_{i \to \infty} \frac{\mu\big(B_\epsilon(x_i) \cap B_\epsilon(y_i)\big)}{\mu\big(B_\epsilon(x_i)\big) + \mu\big(B_\epsilon(y_i)\big) - \mu\big(B_\epsilon(x_i) \cap B_\epsilon(y_i)\big)} \tag{154}$$

$$= \frac{\lim_{i \to \infty} \mu\big(B_\epsilon(x_i) \cap B_\epsilon(y_i)\big)}{\lim_{i \to \infty} \Big(\mu\big(B_\epsilon(x_i)\big) + \mu\big(B_\epsilon(y_i)\big) - \mu\big(B_\epsilon(x_i) \cap B_\epsilon(y_i)\big)\Big)}. \tag{155}$$

Let's consider the numerator first. Rewriting using 152, we have

$$\lim_{i \to \infty} \mu\big(B_\epsilon(x_i) \cap B_\epsilon(y_i)\big) = \lim_{i \to \infty} \int_{\mathbb{R}^D} \rho(z) \cdot \chi_{S_{x_i, y_i}}(z) dV.$$

Observe that $|\rho(z)\chi_{S_{x_i, y_i}}(z)| \le \rho(z)$ for all $i$; $\rho(z)$ is integrable as it represents a probability density function over $\mathbb{R}^D$. Thus, we can invoke the dominated convergence theorem to pull the limit inside of the integral to obtain,

$$\lim_{i \to \infty} \mu\big(B_\epsilon(x_i) \cap B_\epsilon(y_i)\big) = \int_{\mathbb{R}^D} \lim_{i \to \infty} \rho(z) \cdot \chi_{S_{x_i, y_i}}(z) dV$$

$$= \int_{\mathbb{R}^D} \rho(z) \cdot \chi(z) dV.$$

Now let's evaluate $\mu\big(B_\epsilon(x_i)\big)$ in the limit. We can invoke the dominated convergence theorem again for both to pull the limit inside of the integral as before,

$$\lim_{i \to \infty} \mu\big(B_\epsilon(x_i)\big) = \lim_{i \to \infty} \int_{\mathbb{R}^D} \rho(z) \cdot \chi_{S_{x_i}}(z) dV$$

$$= \int_{\mathbb{R}^D} \lim_{i \to \infty} \rho(z) \cdot \chi_{S_{x_i}}(z) dV$$

$$= \int_{\mathbb{R}^D} \rho(z) \cdot \chi(z) dV.$$

And finally the same steps can be taken to find that $\lim_{i\to\infty} \mu\big(B_\epsilon(y_i)\big) = \int_{\mathbb{R}^D} \rho(z) \cdot \chi(z)dV$. Stitching it all together, we have

$$\lim_{i\to\infty} \mathbb{P}_{a\sim\rho}\Big[a \in B_\epsilon(x_i) \cap B_\epsilon(y_i) \,\Big|\, a \in B_\epsilon(x_i) \cup B_\epsilon(y_i)\Big]$$

$$= \frac{\displaystyle\lim_{i\to\infty} \mu\big(B_\epsilon(x_i) \cap B_\epsilon(y_i)\big)}{\displaystyle\lim_{i\to\infty}\Big(\mu\big(B_\epsilon(x_i)\big) + \mu\big(B_\epsilon(y_i)\big) - \mu\big(B_\epsilon(x_i) \cap B_\epsilon(y_i)\big)\Big)}$$

$$= \frac{\displaystyle\int_{\mathbb{R}^D} \rho(z) \cdot \chi(z)dV}{\displaystyle\int_{\mathbb{R}^D} \rho(z) \cdot \chi(z)dV + \int_{\mathbb{R}^D} \rho(z) \cdot \chi(z)dV - \int_{\mathbb{R}^D} \rho(z) \cdot \chi(z)dV}$$

$$= \frac{\displaystyle\int_{\mathbb{R}^D} \rho(z) \cdot \chi(z)dV}{\displaystyle\int_{\mathbb{R}^D} \rho(z) \cdot \chi(z)dV}$$

$$= 1.$$

$\square$

## A.5 ADDITIONAL EXPERIMENTS

### A.5.1 PRUNING

Table 4: Pruning performance of our method vs. baselines described in Appendix A.3.1. For each entry, the top row indicates the percentage of "good" edges removed, while the bottom row indicates the percentage of shortcut edges removed.

| | Concentric Circles | Mixture of Gaussians | Moons | S | Cassini |
|---|---|---|---|---|---|
| ORC-MANL (ours) | **0.0** ± 0.0 **100.0** ± 0.0 | **0.0** ± 0.0 **100.0** ± 0.0 | **0.0** ± 0.0 **100.0** ± 0.0 | **0.0** ± 0.0 **100.0** ± 0.0 | **0.0** ± 0.0 **100.0** ± 0.0 |
| ORC ONLY | 12.6 ± 0.2 **100.0** ± 0.0 | 13.7 ± 0.2 **100.0** ± 0.0 | 12.3 ± 0.2 **100.0** ± 0.0 | 13.4 ± 0.2 **100.0** ± 0.0 | 10.9 ± 0.1 **100.0** ± 0.0 |
| BISECTION (Xia et al., 2008) | 2.6 ± 0.2 45.8 ± 21.5 | 2.0 ± 0.1 45.3 ± 18.5 | 1.7 ± 0.1 **100.0** ± 0.0 | 1.9 ± 0.1 82.0 ± 33.7 | 2.5 ± 0.1 40.0 ± 38.1 |
| MST (Zemel & Carreira-Perpiñán, 2004; Chao et al., 2007) | 0.3 ± 0.2 59.6 ± 8.8 | 1.3 ± 0.2 9.9 ± 11.2 | 0.3 ± 0.1 4.0 ± 12.0 | 2.8 ± 0.4 **100.0** ± 0.0 | 0.0 ± 0.1 70.0 ± 45.8 |
| DENSITY (Chao et al., 2006) | 2.1 ± 0.4 81.9 ± 16.7 | 0.3 ± 0.0 **100.0** ± 0.0 | 5.7 ± 0.3 88.9 ± 23.4 | 3.1 ± 0.3 **100.0** ± 0.0 | 2.1 ± 0.1 0.0 ± 0.0 |
| DISTANCE | 0.8 ± 0.1 80.3 ± 12.0 | 2.8 ± 0.1 **100.0** ± 0.0 | 0.1 ± 0.1 98.6 ± 4.3 | 8.8 ± 0.2 **100.0** ± 0.0 | 1.3 ± 0.1 0.0 ± 0.0 |

### A.5.2 MANIFOLD LEARNING: T-SNE EMBEDDINGS

Here we show additional runs of t-SNE (Van der Maaten & Hinton, 2008) on noisy samples from the swiss roll and the swiss hole. These experiments supplement those that are presented in Figure 5. We find that t-SNE embeddings are inconsistent between runs for this particular task, so to ensure full transparency we show results on three different samples in Figure 20.

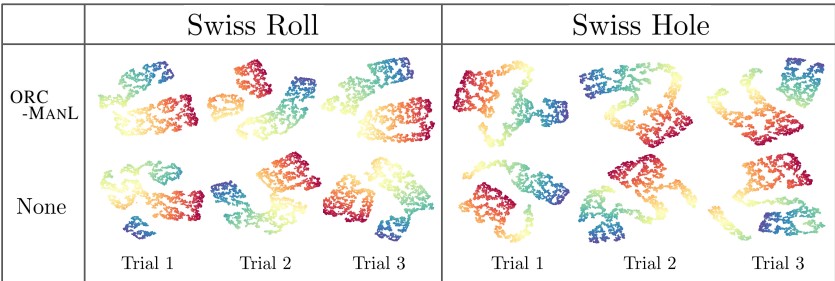

Figure 20: Embeddings produced by t-SNE (Van der Maaten & Hinton, 2008) on three different sets of noisy samples from the swiss roll (left) and swiss hole (right).

### A.5.3 REAL DATA: MNIST AND KMNIST

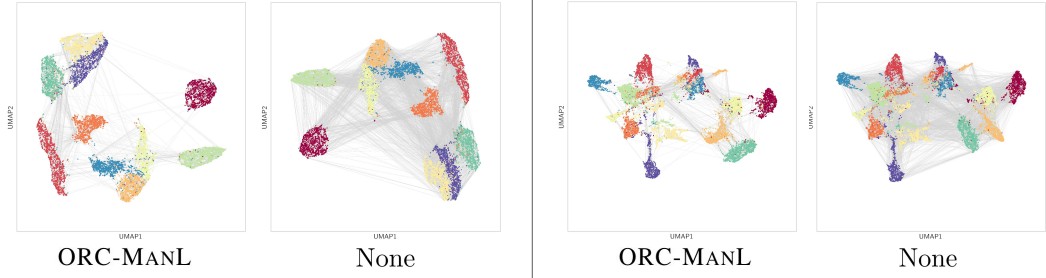

Figure 21: UMAP embeddings of 10,000 MNIST (left) and KMNIST (right) datapoints using nearest neighbor graphs with and without ORC-MANL preprocessing. Ground truth classes are annotated and edges of the pruned and unpruned graphs are visualized in grey.

In this section we include experiments that demonstrate the efficacy of ORC-MANL on canonical datasets in the machine learning literature. Specifically, we evaluate on nearest neighbor graphs built from the MNIST (Deng, 2012) and KMNIST datasets (Clanuwat et al., 2018), with results visualized in Figure 21 (where UMAP is used to visualize pruned and unpruned graphs). Unsurprisingly we find that ORC-MANL pruning removes a significant number of inter-class edges while also preserving intra-class structure. Specifically, we find that ORC-MANL removes 14.92% of inter-class edges and 7.90% of intra-class edges for the MNIST dataset, while it removes 14.70% of inter-class edges and 5.39% of intra-class edges for KMNIST. We emphasize, however, that edges that connect data points in different classes may *not* necessarily be shortcut edges. Similarly, edges that connect points in the same classes can satisfy the definition of shortcut edges. Thus interpretation of inter versus intra-class edge removal results should be approached with caution.

### A.5.4 CLUSTERING

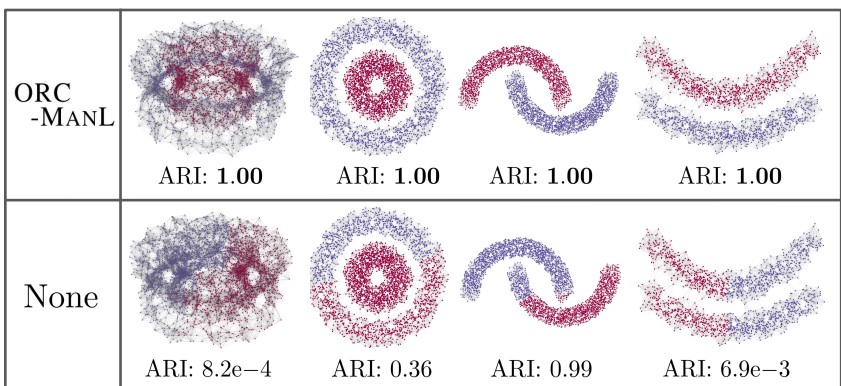

Figure 22: Spectral clustering applied to several synthetic manifolds, with adjusted Rand index (ARI) shown. Nearest neighbor graphs in the top row were pruned by ORC-MANL, while the nearest neighbor graphs on the bottom row were not.

We further evaluate ORC-MANL pruned graphs on spectral clustering, a canonical algorithm for finding communities of arbitrary shape (Bach & Jordan, 2003). The algorithm embeds the data using the eigenvectors of the graph Laplacian, and proceeds by running the $k$-means algorithm on the embedding (Hartigan & Wong, 1979). On this task we expect ORC-MANL to help very much: if the underlying manifold exhibits several connected components, ORC-MANL pruning should reveal them, resulting in an easy task for spectral clustering. Unsurprisingly, we find that holds true, as ORC-MANL improves spectral clustering as measured by the adjusted Rand index (ARI) (Hubert & Arabie, 1985). A higher ARI indicates a better alignment with base truth clustering, with the best possible score being 1. We test on noisy manifolds exhibiting two underlying connected components, where visualization of the results is shown in Figure 22. We find that ORC-MANL ensures perfect ARI across examples, while the unpruned graphs result in a trivial clustering that is not representative of the underlying structure of the data. We do not include t-SNE or UMAP embeddings as they require finite pairwise distances, which does not hold in the case where pruning (correctly) results in more than one connected component.

### A.5.5 PARAMETER ABLATIONS

To analyze the sensitivity of the ORC-MANL algorithm to key parameters, we include the results of pruning experiments with varying parameter settings. Specifically, we analyze the nearest-neighbor graph parameter $k$, the ORC threshold parameter $\delta$, and the thresholded graph distance parameter $\lambda$. The pruning results on four synthetic noisy manifolds with 4000 points across 10 seeds are visualized in Table 5. The top row of each entry depicts the mean and standard deviation of the percentage of *good* edges removed, while the bottom row of each entry depicts the mean and standard deviation of the percentage of *shortcutting* edges removed. Parameters are set to $k = 20$, $\delta = 0.8$, and $\lambda = 0.01$ unless the said parameter is being varied.

Table 5: Ablation experiments for parameters $k$, $\delta$ and $\lambda$ respectively.

| | $k = 10$ | $k = 15$ | $k = 20$ | $k = 25$ | $k = 30$ |
|---|---|---|---|---|---|
| Concentric Circles | $0.4 \pm 0.1$ $100.0 \pm 0.0$ | $0.0 \pm 0.0$ $100.0 \pm 0.0$ | $0.0 \pm 0.0$ $100.0 \pm 0.0$ | $0.0 \pm 0.0$ $100.0 \pm 0.0$ | $0.0 \pm 0.0$ $100.0 \pm 0.0$ |
| Mixture of Gaussians | $0.7 \pm 0.2$ $100.0 \pm 0.0$ | $2.0\text{e-}3 \pm 4.5\text{e}{-}3$ $100.0 \pm 0.0$ | $0.0 \pm 0.0$ $100.0 \pm 0.0$ | $0.0 \pm 0.0$ $100.0 \pm 0.0$ | $0.0 \pm 0.0$ $100.0 \pm 0.0$ |
| Chained Tori | $0.7 \pm 0.3$ $97.9 \pm 6.4$ | $2.3\text{e-}3 \pm 7.0\text{e}{-}3$ $100.0 \pm 0.0$ | $0.0 \pm 0.0$ $100.0 \pm 0.0$ | $0.0 \pm 0.0$ $91.6 \pm 20.0$ | $0.0 \pm 0.0$ $100.0 \pm 0.0$ |
| Concentric Hyperboloids | $0.5 \pm 0.2$ $100.0 \pm 0.0$ | $2.9\text{e-}4 \pm 8.6\text{e}{-}4$ $100.0 \pm 0.0$ | $0.0 \pm 0.0$ $100.0 \pm 0.0$ | $0.0 \pm 0.0$ $97.6 \pm 7.2$ | $0.0 \pm 0.0$ $100.0 \pm 0.0$ |

| | $\delta = 0.70$ | $\delta = 0.75$ | $\delta = 0.80$ | $\delta = 0.85$ | $\delta = 0.90$ |
|---|---|---|---|---|---|
| Concentric Circles | $0.1 \pm 0.1$ $96.2 \pm 1.1$ | $0.0 \pm 0.0$ $100.0 \pm 0.0$ | $0.0 \pm 0.0$ $100.0 \pm 0.0$ | $0.0 \pm 0.0$ $100.0 \pm 0.0$ | $0.0 \pm 0.0$ $84.6 \pm 31.3$ |
| Mixture of Gaussians | $0.3 \pm 0.3$ $100.0 \pm 0.0$ | $0.0 \pm 0.0$ $100.0 \pm 0.0$ | $0.0 \pm 0.0$ $100.0 \pm 0.0$ | $0.0 \pm 0.0$ $100.0 \pm 0.0$ | $0.0 \pm 0.0$ $100.0 \pm 0.0$ |
| Chained Tori | $0.4 \pm 0.2$ $96.0 \pm 5.0$ | $0.0 \pm 0.0$ $100.0 \pm 0.0$ | $0.0 \pm 0.0$ $100.0 \pm 0.0$ | $0.0 \pm 0.0$ $100.0 \pm 0.0$ | $0.0 \pm 0.0$ $97.6 \pm 7.2$ |
| Concentric Hyperboloids | $0.4 \pm 0.2$ $94.5 \pm 5.3$ | $1.3\text{e-}3 \pm 3.9\text{e}{-}3$ $100.0 \pm 0.0$ | $0.0 \pm 0.0$ $100.0 \pm 0.0$ | $0.0 \pm 0.0$ $100.0 \pm 0.0$ | $0.0 \pm 0.0$ $91.2 \pm 13.8$ |

| | $\lambda = 1\text{e}{-}3$ | $\lambda = 1\text{e}{-}2$ | $\lambda = 0.1$ | $\lambda = 0.2$ | $\lambda = 0.5$ |
|---|---|---|---|---|---|
| Concentric Circles | $0.0 \pm 0.0$ $100.0 \pm 0.0$ | $0.0 \pm 0.0$ $100.0 \pm 0.0$ | $2.7\text{e}{-}2 \pm 2.3\text{e}{-}2$ $100.0 \pm 0.0$ | $0.3 \pm 5.0\text{e}{-}2$ $100.0 \pm 0.0$ | $12.7 \pm 0.1$ $100.0 \pm 100.0$ |
| Mixture of Gaussians | $0.0 \pm 0.0$ $100.0 \pm 0.0$ | $0.0 \pm 0.0$ $100.0 \pm 0.0$ | $4.6\text{e}{-}2 \pm 1.9\text{e}{-}2$ $100.0 \pm 0.0$ | $0.5 \pm 4.3\text{e}{-}2$ $100.0 \pm 0.0$ | $13.8 \pm 0.2$ $100.0 \pm 0.0$ |
| Chained Tori | $0.0 \pm 0.0$ $89.2 \pm 29.8$ | $0.0 \pm 0.0$ $100.0 \pm 0.0$ | $0.1 \pm 4.4\text{e}{-}2$ $100.0 \pm 0.0$ | $12.9 \pm 0.2$ $100.0 \pm 0.0$ | $15.7 \pm 0.2$ $97.4 \pm 7.7$ |
| Concentric Hyperboloids | $0.0 \pm 0.0$ $98.8 \pm 3.0$ | $0.0 \pm 0.0$ $98.5 \pm 3.0$ | $8.8\text{e}{-}2 \pm 5.2\text{e}{-}2$ $100.0 \pm 0.0$ | $1.0 \pm 0.1$ $100.0 \pm 0.0$ | $13.0 \pm 0.2$ $99.8 \pm 0.5$ |

### A.5.6 SCRNASEQ REAL DATA: UMAP AND tSNE

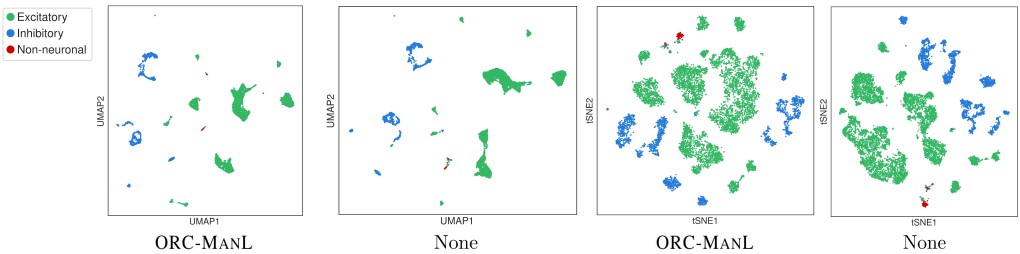

Figure 23: UMAP (McInnes et al., 2018a) and t-SNE (Van der Maaten & Hinton, 2008) embeddings of scRNAseq data from 10000 Anterolateral Motor Cortex (ALM) brain cells in mice with base truth cell-type annotation with and without ORC-MANL pruning. Data available from Abdelaal et al. (2019) and the Allen Brain Institute.

Here we include UMAP (McInnes et al., 2018a) and t-SNE (Van der Maaten & Hinton, 2008) embeddings the of scRNAseq data of brain cells in mice that was analyzed in Section 4.2.

Figure 23 shows embeddings of scRNAseq data from 10000 Anterolateral Motor Cortex (ALM) brain cells in mice. Unlike the embeddings produced by Isomap (Tenenbaum et al., 2000) shown in Figure 8, we find that UMAP and t-SNE do poorly with and without ORC-MANL pruning as measured by the extent to which neuronal (consisting of labels "Inhibitory" and "Excitatory") and non-neuronal cell communities are preserved.

### A.5.7 MANIFOLD CURVATURE

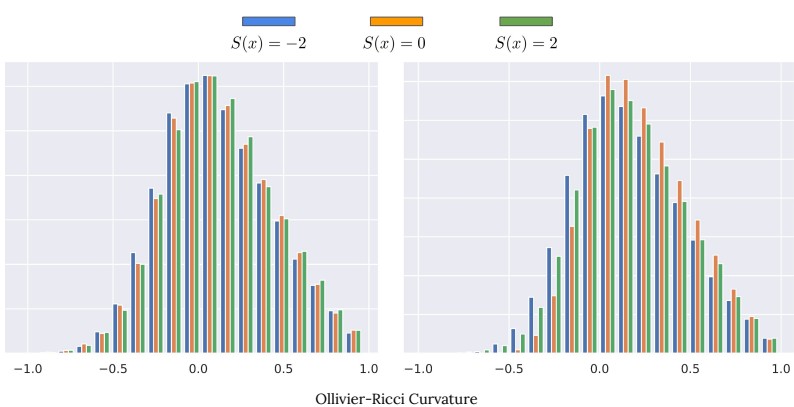

Figure 24: Empirical distribution of ORC for 2000 points sampled each from the Bolza surface, the flat torus and the unit 2-sphere using $k$-NN connectivity (left) and $\epsilon$-radius connectivity (right). The Bolza surface has scalar curvature $-2$, the flat torus has scalar curvature $0$ while the sphere has scalar curvature $+2$. Note that for 2-dimensional surfaces such as these, the scalar curvature is simply twice the Gaussian curvature.

Theoretical and empirical results from van der Hoorn et al. (2021) indicate that under noiseless and uniform sampling, a specific instantiation of ORC converges to the Ricci curvature of the underlying manifold from which the data was sampled (up to a constant of proportionality). This result could be a point of concern for ORC-MANL, as it would suggest variable performance as a function of manifold curvature. To quell any concern, in Figure 24 we plot the distribution of ORC values (computed with the formulation described in Section 2.3) for nearest neighbor graphs where points were sampled noiselessly from three different manifolds of varying curvature: the Bolza surface (scalar curvature $-2$), the flat torus (scalar curvature $0$) and the unit sphere (scalar curvature $+2$). Note that for surfaces such as these, the scalar curvature is simply twice the Ricci curvature.

In this experiment, we find minimal variation in the distribution of ORC values for the three manifolds. While this does not suggest invariance to curvature in general, we consider this to be an indication that for the regimes we expect to encounter in practice, underlying manifold curvature should not be a significant cause for concern.

### A.5.8 EMPIRICAL CONVERGENCE: THE SWISS ROLL

Figure 25 plots the ORC versus thresholded graph distance for all edges in a nearest neighbor graph built from noisy samples from the swiss roll. The figure also plots the thresholds indicated from the theoretical results detailed in Section 3. In this example, we see that all shortcut edges have ORC below $-1 + 4(1 - \delta)$, though some non-shortcut edges fall under this threshold too. But the figure also illustrates that *all* shortcut edges exceed the thresholded graph distance, while *no* non-shortcut edges do.

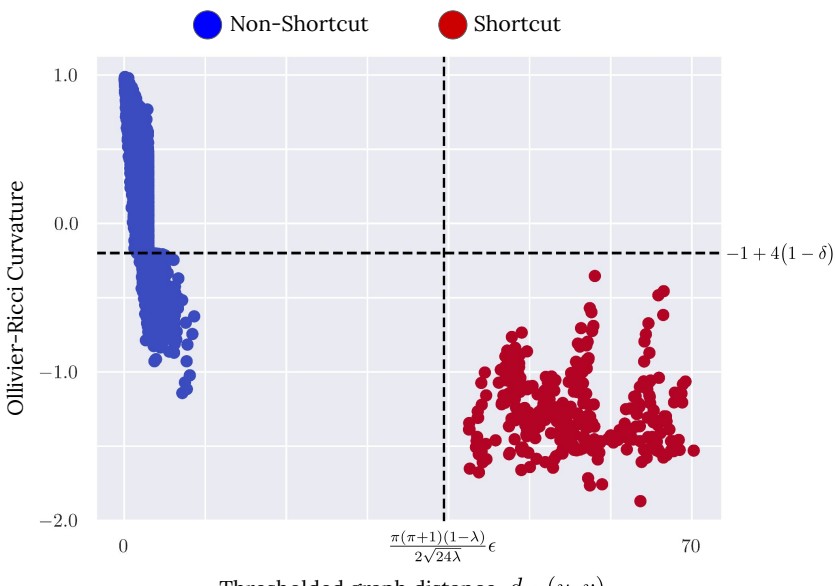

Figure 25: Plot of ORC versus *thresholded* graph distance $d_{G'}$ for all edges in an unpruned nearest neighbor graph of the noisy 3D swiss roll. Dotted lines indicate theory-derived thresholds on ORC and thresholded-graph distance respectively. Note that we use $\delta = 0.8$, $\lambda = 0.01$ and $\epsilon = 3$.

