# OpenReview forum: "Recovering Manifold Structure Using Ollivier Ricci Curvature"
_ICLR.cc/2025/Conference — ICLR 2025 Spotlight_

### Official Review · Reviewer_rkDt · 2024-10-25

**Soundness:** 3
**Presentation:** 2
**Contribution:** 3
**Rating:** 8
**Confidence:** 3

**Summary:**

In this paper, the authors propose using the Ollivier-Ricci curvature, calculated for each edge of graph, to prune graphs in machine learning methods that construct local graphs from data points. They develop an algorithm that removes edges that, due to noise, create shortcuts connecting the low-dimensional manifold formed by the data. Using synthetic data where the shortcuts are known, they demonstrate the ability to identify these shortcuts. Furthermore, they experimentally show that this approach is useful for tasks such as preprocessing for manifold learning methods like ISOMAP and UMAP, computing persistent homology, and intrinsic dimension estimation.

**Strengths:**

Providing a method that can accurately prune shortcut edges—which significantly affect the performance of downstream tasks—using a relatively simple algorithm will have a major impact on many machine learning algorithms based on local graphs.

**Weaknesses:**

The presentation of the paper has considerable room for improvement.

Given the nature of the research, it's somewhat understandable that most of the content is described in the Appendix. However, it is undesirable that figures providing intuitive explanations of geometric concepts and constants (thresholds) that play important roles in the algorithm cannot be interpreted without consulting the Appendix. For example, rather than including numerous detailed experimental results, adding figures that intuitively explain the definition and properties of the minimum branch separation would contribute more to understanding after making rooms by moving one of two experimental results to the Appendix. Additionally, the fact that the basis for comparing with ORC, '-1 + 4(1 - δ)', is provided in Lemma A.1 hinders the understanding of the proposed algorithm.

The (weighted) shortest-path metrics \( d_G \) and \( d_{\mathcal{M}} \) are not formally defined. These are very important quantities, and the discussion should proceed after giving precise definitions of them.

In Figures 4, 5, and 6, some words (upper left part of Figures) are displayed overlapping, making it impossible to discern what is written.

It has been pointed out that intrinsic dimension estimation using the MLE by Levina & Bickel has bias.
https://www.inference.org.uk/mackay/dimension/
If the claim is that the proposed ORC-MANL preprocessing contributes to correct intrinsic dimension estimation, then a bias-reduced estimator should be used and discuss how their results might change with a different estimator.

**Questions:**

Major question:
Since the proposed ORC-based method is based on the idea of removing edges with highly negative curvature, wouldn't it perform poorly when the graph constructed using the epsilon-radius rule has an almost tree-like topology (i.e., graphs with very few cycles or triangles)? In other words, in graphs that are close to tree structures, a majority of edges may have negative curvature, and as a result, wouldn't the ORC-based method over-prune the graph?


Minor questions:
What does ORC-MANL stand for? While it's clear that ORC is Ollivier-Ricci curvature, there's no explanation of MANL. Since it is the proposed algorithm in this paper and an important term, it should be spelled out first before abbreviating.

Why is only Definition 2.1 enclosed in a box and emphasized in color? Other Propositions and Theorems are not treated this way; what is the reason for doing this only for this definition?

---

> ### Author Response · Authors · 2024-11-20
> **Rebuttal (Part 1)**
>
> We would like to thank the reviewer for leaving thoughtful comments and suggestions regarding our work!
>
> Weaknesses:
>
> 1. *Given the nature of the research, it's somewhat understandable that most of the content is described in the Appendix. However, it is undesirable that figures providing intuitive explanations of geometric concepts and constants (thresholds) that play important roles in the algorithm cannot be interpreted without consulting the Appendix. For example, rather than including numerous detailed experimental results, adding figures that intuitively explain the definition and properties of the minimum branch separation would contribute more to understanding after making rooms by moving one of two experimental results to the Appendix. Additionally, the fact that the basis for comparing with ORC, '-1 + 4(1 - δ)', is provided in Lemma A.1 hinders the understanding of the proposed algorithm.*
>
> We apologize for any lack of clarity due to the presentation.  To improve the presentation, we have added a figure to the body of the paper that depicts the two key manifold embedding terms that we use: minimum branch separation and minimum radius of curvature. As for the exact expressions for the curvature and distance thresholds in the ORC-ManL algorithm, we understand the confusion and we have adjusted the text to try to be clearer. We have added a brief discussion of the ORC threshold on line 206, and a discussion of the distance threshold on line 219. To make room for these adjustments, we have moved one of the two tables depicting pruning results to Appendix A.5.1.
>
>
>
> 2. *The (weighted) shortest-path metrics ( d_G ) and ( d_{\mathcal{M}} ) are not formally defined. These are very important quantities, and the discussion should proceed after giving precise definitions of them.*
>
> This is an excellent point, as these metrics should have been clearly defined and may not be clear from context. We have added exact definitions and a brief discussion for said metrics.
>
> 3. *In Figures 4, 5, and 6, some words (upper left part of Figures) are displayed overlapping, making it impossible to discern what is written.*
>
> Unfortunately this seems to be an artifact of pdf rendering on mobile devices (tablets and smartphones) due to the use of certain pdf rendering software. We sincerely apologize for this, and we have addressed it with updated figures made with different software that does not exhibit the problem in our tests.
>
>
>
> 4. *It has been pointed out that intrinsic dimension estimation using the MLE by Levina & Bickel has bias. https://www.inference.org.uk/mackay/dimension/ If the claim is that the proposed ORC-MANL preprocessing contributes to correct intrinsic dimension estimation, then a bias-reduced estimator should be used and discuss how their results might change with a different estimator.*
>
> This is a great question.  Although the global estimate produced by averaging local estimates in Levina-Bickel is indeed biased, the local intrinsic dimension estimators themselves are not biased. In our experiment, we simply look at the MSE of pointwise estimates from the ground truth value, which involves only the local (unbiased) estimators.

---

> > ### Comment · Reviewer_rkDt · 2024-11-22
> > **Thanks for the clarification**
> >
> > Thank you for your response. I'm basically satisfied with your response and the revised manuscript. I would suggest adding a sentence to say that Levina&Bickel's estimator is used for local intrinsic dimension estimation and is unbiased.

---

> > > ### Author Response · Authors · 2024-11-22
> > > **Response to reviewer comment**
> > >
> > > Thank you for the suggestion! We have added a sentence to the appendix acknowledging that we only use Levina&Bickel's local estimator.

---

> > > > ### Comment · Reviewer_rkDt · 2024-11-24
> > > >
> > > > Thank you for addressing all of my concerns. I have increased my score accordingly.

---

> ### Author Response · Authors · 2024-11-20
> **Rebuttal (Part 2)**
>
> Questions:
> 1. *Major question: Since the proposed ORC-based method is based on the idea of removing edges with highly negative curvature, wouldn't it perform poorly when the graph constructed using the epsilon-radius rule has an almost tree-like topology (i.e., graphs with very few cycles or triangles)? In other words, in graphs that are close to tree structures, a majority of edges may have negative curvature, and as a result, wouldn't the ORC-based method over-prune the graph?*
>
> This is an excellent observation by the reviewer. We would like to emphasize the fact that using unweighted graph distances to compute ORC (as we do) results in ORC values that reflect local connectivity and deviate from expectations from classical notions of curvature.  For empirical support of this idea, we would also like to direct the reviewer to Section A.5.7 where we include an experiment that illustrates the empirical invariance of the distribution of ORC values to the underlying curvature of the manifold. As a separate point, we emphasize the fact that Theorem 3.2 indicates that if one has many samples, then there exist many positive curvature edges incident to any datapoint with probability 1. Critically, this result is independent of the underlying curvature of the submanifold. Here is some intuition as to why this is true: if we are given many noisy samples from a submanifold of Euclidean space, we would expect that the nearest neighbor graph has some clique-like (positive ORC) connectivity regardless of how negative the curvature of the submanifold. Therefore if we sample enough points we would not in general expect to see graphs with tree-like connectivity, regardless of how negative the curvature of the underlying manifold. In practice, this implies that if we have enough samples, we should expect the graph with the candidate edge set C removed  (where C is the set of edges with curvature more nergative than the threshold) to effectively scaffold the underlying manifold–this would mean ORCManL would be able to detect any shortcutting edges without pruning too many good ones. If the graph had tree-like connectivity, then ORCManL may prune many edges (depending on the chosen ORC threshold and the number of children each node has). However, we emphasize that this is not a scenario we expect to encounter when dealing with nearest-neighbor graphs; given enough points, many neighbors of any two points connected by an edge are likely to be shared. Connectivity of this nature is not exhibited by trees.
>
>
>
> 2. *Minor questions: What does ORC-MANL stand for? While it's clear that ORC is Ollivier-Ricci curvature, there's no explanation of MANL. Since it is the proposed algorithm in this paper and an important term, it should be spelled out first before abbreviating.*
>
> We are very sorry for any confusion. We have added the following sentence to the introduction of the manuscript: “Guided by these results, we describe an algorithm, Ollivier-Ricci Curvature-based Manifold Learning and recovery (ORC-ManL), to detect and prune shortcut edges.”
>
>
> 3. *Why is only Definition 2.1 enclosed in a box and emphasized in color? Other Propositions and Theorems are not treated this way; what is the reason for doing this only for this definition?*
>
> We thank the reviewer for pointing out this formatting inconsistency. We have changed it so that all definitions, propositions, remarks, lemmas, and theorems use the same formatting.

---

### Official Review · Reviewer_Dtcv · 2024-11-03

**Soundness:** 3
**Presentation:** 3
**Contribution:** 3
**Rating:** 6
**Confidence:** 2

**Summary:**

The paper presents ORC-MANL, an algorithm aimed at refining the structure of nearest neighbor graphs by eliminating spurious "shortcut" edges that disrupt accurate manifold representations. This method uses Ollivier-Ricci curvature (ORC) to detect and prune edges that likely bypass the true manifold structure. By removing these shortcuts, ORC-MANL improves manifold accuracy, enhancing performance in downstream tasks such as clustering, manifold learning, and persistent homology.

**Strengths:**

- ORC-MANL effectively removes shortcut edges, which often misrepresent relationships in the data by connecting distant points across the manifold. By pruning these edges, the algorithm preserves the true geometric and topological structure of the data, yielding a more accurate manifold representation.

- It is supported by a solid theoretical basis showing how Ollivier-Ricci curvature identifies shortcut edges. Empirical results on synthetic and real datasets confirm that ORC-MANL effectively prunes unnecessary edges while preserving important connections, making it a robust choice across a range of data types and applications.

- It operates with fixed parameters for curvature thresholds across different datasets, reducing the need for dataset-specific tuning. This consistency is an advantage over other methods that require careful parameter adjustment to perform optimally.

- ORC-MANL demonstrates versatility across both synthetic and real-world datasets, including complex biological data like high-dimensional single-cell RNA sequencing. This adaptability makes it suitable for diverse data shapes and manifold structures.

**Weaknesses:**

- Calculating ORC involves complex optimal transport calculations and distance checks, which can be computationally intensive. This added complexity may limit ORC-MANL's scalability for large graphs or high-dimensional data, making it less suitable for applications requiring quick processing.

- It seems to me that ORC-MANL’s performance depends on how the initial nearest neighbor graph is constructed, including choices like the number of neighbors k in k-NN graphs or the distance threshold ϵ in ϵ-radius graphs. Poorly chosen parameters can produce an inaccurate initial graph, which may reduce the effectiveness of ORC-MANL’s pruning. This sensitivity may require users to fine-tune parameters for each dataset to achieve the best results.

- While ORC-MANL generally operates effectively with fixed parameters, the initial nearest neighbor graph setup (e.g., choosing k for k-NN graphs) may still need adjustment across different data types. This reliance on specific curvature thresholds and distance validations can impose extra processing time, especially in cases where the default settings don’t align well with the data structure.

**Questions:**

See weakness.

---

> ### Author Response · Authors · 2024-11-20
> **Rebuttal**
>
> We would like to thank the reviewer for taking the time to read and analyze our work!
>
> Weaknesses/Questions:
>
> 1. The reviewer brings up an excellent point, and we have added Section A.2 to the appendix to discuss ORCManL’s time complexity. The ORCManL algorithm has a time complexity of O(|V|k^4) + O(k|V|^2log|V|) when using k-NN graphs and is therefore sub-cubic in the dataset size (since k is typically dependent on the dimensionality of the data manifold and independent of the number of samples). This arises in part from the fact that the optimal transport calculations use only very local information  (measures over one-hop neighborhoods) and therefore can be done quickly.   Given that other geometric data analysis algorithms (e.g., PCA or Isomap) require solving large eigenvector problems that scale as O(|V|^3), we consider this runtime reasonable. Finally, to provide empirical evidence about the time complexity, in Section A.2 we include an experiment that shows that ORC-ManL runs significantly faster than UMAP over a range of dataset sizes.
>
> 2. This is an excellent point. To address this we added ablation experiments (Table 5 in Section A.5.5) that support the notion that our method is robust to the choice of k.  Over the entire range tested we find minimal variation in the performance of ORCManL. Furthermore, we would also like to emphasize that sensitivity to k is a ubiquitous challenge in problems that use nearest neighbor graphs, and general strategies for choosing k will apply to our algorithm.
>
> 3. To illustrate the degree to which ORCManL is sensitive to algorithm parameters delta and lambda, we include pruning results in Table 5 across different choices of these parameters. We find that ORCManL is minimally sensitive to lambda, and slightly more sensitive to delta. That being said, over the ranges tested we find that even in the worst case at least 84% of shortcutting edges are removed and no more than 16% of good edges are removed. We would also like to emphasize the fact that our use of the unweighted graph metric in computing ORC means the ORC threshold is invariant to scale and only depends on local connectivity. Similarly, the distance distortion threshold is a function of epsilon (or average neighbor distance if using k-NN connectivity), and is therefore also invariant to scale.  This means feature normalization is not required to use ORCManL. Furthermore, our use of consistent algorithm parameters across data modalities (synthetic 3D data, scRNAseq data, MNIST + KMNIST images) suggests that minimal adjustment is needed across datasets.

---

> > ### Comment · Reviewer_7aFm · 2024-11-26
> >
> > I thank the authors for the responses. I shall keep my score unchanged.

---

### Official Review · Reviewer_7aFm · 2024-11-03

**Soundness:** 3
**Presentation:** 4
**Contribution:** 3
**Rating:** 8
**Confidence:** 3

**Summary:**

The paper focuses on characterizing ``shortcut'' edges as edges that are essentially unfaithful to the geometric curvature and then designs a polynomial time algorithm to detect these edges. They first define discrete Ricci curvature corresponding to each edge, where a negative curvature implies shortcut edges. To further confirm that these edges are indeed shortcut edges, they ensure that the shortest distance between the corresponding vertices is sufficiently large if the shortcut edges are absent.

The algorithm is simple to understand, and the idea of negative Ricci curvature certainly shows promise in simulation settings when considering many different kinds of manifolds. The simulation results are quite rigorous, and the application to real-world data shows initial promise.

**Strengths:**

1) The algorithm aims to solve an important problem, pruning of ``bad edges'' in nearest-neighbor data embeddings with underlying manifold structures.

2) The theoretical results of the papers (especially Theorem 3.2 and 3.3) are both simple to understand and insightful.

3) The algorithm performs very well on large simulation datasets.

4) It also shows some promise in the real-world datasets (Improvement in the performance of ALM data for ISOMAP and reducing some inter-community edges in the PBMC dataset).

**Weaknesses:**

1) The primary weakness of the paper is that the application to real-world data is quite limited. The authors only consider $2$ real-world datasets. Experiments and verifications on more data (such as image datasets like Fashion-MNIST or K-MNIST) or embeddings of small document datasets would significantly increase confidence in the method.

2) The robustness of the tolerance parameters $\lambda$ and $\delta$ are not well explored. This is especially important for the real-world datasets.

3) The time complexity of the algorithm seems quite bad $\mathcal{O}(|V|^3)$. This is a minor weakness, given the fact that the paper primarily focuses on a theoretically established idea.

**Questions:**

1) The authors assume that the data comes from a smooth manifold $\mathcal{M}$. However, real-world experiments are conducted on data with underlying clusters, where it makes more sense to assume that different clusters come from different manifolds (leading to multi-manifold structures [1] ). How do the theoretical guarantees of the paper hold in such a setting?

2) For the PBMC dataset, while there are now fewer edges between different separated components of the UMAP/TSNE pictures, in my understanding, the overall UMAP/TSE embedding quality remains unchanged for both pruned and un-pruned nearest neighbor graph distances. Would this be a correct inference? Also, have the authors considered using datasets like Fashion-MNIST (which are quite well-studied in manifold learning literature) to better understand the usability of the method?

3) How did the authors choose the parameters for the baseline algorithms? How did the authors choose the $\lambda$ and $\delta$ parameters for the real-world experiments?

4) The authors have not commented on the run time complexity of the algorithm (I understand that it is a theoretical paper). I believe it should be $\mathcal{O}(|V|^3)$. Is that correct? If so, have the authors attempted to develop faster algorithms via sampling/approximation style techniques?

[1] Trillos, Nicolas Garcia, Pengfei He, and Chenghui Li. "Large sample spectral analysis of graph-based multi-manifold clustering." Journal of Machine Learning Research 24.143 (2023): 1-71.

---

> ### Author Response · Authors · 2024-11-20
> **Rebuttal**
>
> Thank you for carefully reading our submission and providing thoughtful feedback! We believe we can address all of the concerns and questions raised.
>
> Weaknesses:
> 1. To address this we have added experiments (Figure 21) that apply ORC-ManL to the KMNIST and MNIST datasets. We find that ORC-ManL removes many inter-class edges while preserving relevant structures in the data for both datasets. Specifically, we find that ORC-MANL removes 14.92% of inter-class edges and 7.90% of intra-class edges for the MNIST dataset, while it removes 14.70% of inter-class edges and 5.39% of intra-class edges for KMNIST. We emphasize, however, that edges that connect data points in different classes may not necessarily be shortcut edges. Similarly, edges that connect points in the same classes can satisfy the definition of shortcut edges. Thus interpretation of inter versus intra-class edge removal results should be approached with caution.
>
> 2. You are absolutely correct that for real-world applications understanding of the tolerance parameters is essential. To address this, we have added ablation experiments (Table 5) that explore the effect of varying lambda and delta. We find that the algorithm exhibits minimal sensitivity to lambda and marginally more sensitivity to delta. That being said, over the ranges tested we find that even in the worst case at least 84% of shortcutting edges are removed and no more than 16% of good edges are removed.
>
> 3. This is an excellent point, and we have added Section A.2 to the appendix to discuss ORCManL’s time complexity. The ORC-ManL algorithm has a time complexity of O(|V|k^4) + O(k|V|^2log|V|) when using k-NN graphs and is therefore sub-cubic in the dataset size (since k is typically dependent on the dimensionality of the data manifold and independent of the number of samples). This arises in part from the fact that the optimal transport calculations use only very local information  (measures over one-hop neighborhoods) and therefore can be done quickly.   Given that other geometric data analysis algorithms (e.g., PCA or Isomap) require solving large eigenvector problems that scale as O(|V|^3), we consider this runtime reasonable. Finally, to provide empirical evidence about the time complexity, in Section A.2 we include an experiment that shows that ORC-ManL runs much faster than UMAP over a range of dataset sizes.
>
> Questions:
> 1. Thank you for the insightful question.  When the underlying data is a disjoint union of manifolds of the same dimension, our theoretical analysis continues to apply; this only requires the assumption of pairwise disjointness of the clusters and consistent intrinsic dimensionality.  When the clusters are sampled from manifolds of different dimensions, we are quite confident that the algorithm still works with analogous theoretical justification, but carefully exploring the theory here is the subject of future work.
>
> 2. In this experiment UMAP was simply used to visualize the pruned and unpruned graphs, allowing us to see which edges the algorithm chose to remove.  Your inference is correct, as in general we see limited sensitivity of UMAP and tSNE to shortcut edges and so these algorithms do not benefit from ORC-ManL pruning.
>
> 3. For the baseline algorithms we compared against, parameters were tuned individually for each synthetic manifold to maximize performance to provide the most competitive possible comparison. In contrast, for our evaluation of ORC-ManL parameters lambda = 0.01 and delta = 0.8 were obtained by tuning for a single synthetic manifold (the swiss roll)  and the same parameter settings were then used across all synthetic manifolds in the evaluation set.
>
> 4. We believe that our response to the third weakness encompasses part of this question.  Thank you for the suggestion to consider subsampling — we have not explored subsampling or similar approximation methods in detail, but we believe that subsampling would accelerate the method with limited accuracy loss.  Theoretical verification of this is future work.

---

### Author Response · Authors · 2024-11-20
**Rebuttal**

We thank all reviewers for carefully reading our work and providing insightful reviews. We have responded to their comments individually and adjusted the manuscript accordingly, with new additions highlighted in blue for clarity. Here we summarize key changes:

1. Time Complexity: We have added a section to the appendix (A.2) that discusses the time complexity of the ORCManL algorithm and compares it to other geometric data analysis algorithms. In this section, we show that the time complexity of ORCManL is sub-cubic in dataset size when using k-NN graphs. We also provide a new figure (Figure 10) that documents its wall clock time relative to UMAP. We find that over a range of dataset sizes, ORCManL is significantly faster than UMAP. We find that ORCManL terminates significantly slower than PCA, despite its theoretical time complexity that beats the cubic time complexity of eigendecomposition approaches. We attribute this to SVD optimizations and speedups that can be achieved when the ambient data dimension d << n.

2. Real World Data: We have added experiments to the appendix (A.5.3) that illustrate ORCManL’s strong performance on MNIST and KMNIST.

3. Parameter Ablations: We have added ablation experiments to the appendix (A.5.5, Table 5) that detail ORCManL’s pruning performance on several synthetic datasets as a function of algorithm parameters. We find limited sensitivity and strong performance across the range tested.

4. Presentation: We have adjusted the presentation to make figures clearer and formatting consistent. A figure (Figure 1) has been added to visualize manifold embedding parameters, and previously undefined terms have been defined.

We hope these changes clarify any confusion and address concerns brought up by reviewers. We will be happy to provide further explanations or more experiments if any reviewer feels this would be beneficial. Thank you!

---

### Meta-Review · Area_Chair_N339 · 2024-12-16

**Metareview:**

The paper introduces ORC-MANL, an algorithm to refine nearest neighbor graphs by removing "shortcut" edges that misrepresent the true geometric structure of data manifolds. Using Ollivier-Ricci curvature, edges with negative curvature are identified as shortcuts and pruned if their removal significantly increases the shortest path between connected nodes. This approach improves graph fidelity and benefits tasks like manifold learning, clustering, persistent homology, and intrinsic dimension estimation. Simulation results on synthetic and real-world datasets demonstrate the algorithm’s effectiveness, highlighting its potential as a preprocessing step for manifold-based machine learning methods such as ISOMAP and UMAP.

The idea of graph pruning based on Ollivier-Ricci curvature is innovative, and the reviewers are satisfied with the authors' rebuttal. Therefore, I also recommend accepting the paper.

**Additional Comments On Reviewer Discussion:**

The reviewer raised concerns about the computational complexity of the proposed method and potential over-pruning issues when the graph is close to a tree (i.e., having negative curvature). The authors clearly addressed these concerns, and one of the reviewers raised their score to 8. Currently, the average score is 7.33, which is clearly above the borderline. I believe this is a clear case for acceptance.

---

### Decision · Program_Chairs · 2025-01-22

Accept (Spotlight)